# Deep Learning From Crowdsourced Labels: Coupled Cross-entropy Minimization, Identifiability, and Regularization

**Shahana Ibrahim, Tri Nguyen, and Xiao Fu** [*]
School of Electrical Engineering and Computer Science
Oregon State University
Corvallis, OR 97330, USA

## Abstract

Using noisy crowdsourced labels from multiple annotators, a deep learning-based end-to-end (E2E) system aims to learn the label correction mechanism and the neural classifier simultaneously. To this end, many E2E systems concatenate the neural classifier with multiple annotator-specific "label confusion" layers and co-train the two parts in a parameter-coupled manner. The formulated *coupled cross-entropy minimization* (CCEM)-type criteria are intuitive and work well in practice. Nonetheless, theoretical understanding of the CCEM criterion has been limited. The contribution of this work is twofold: First, performance guarantees of the CCEM criterion are presented. Our analysis reveals for the first time that the CCEM can indeed correctly identify the annotators' confusion characteristics and the desired "ground-truth" neural classifier under realistic conditions, e.g., when only incomplete annotator labeling and finite samples are available. Second, based on the insights learned from our analysis, two regularized variants of the CCEM are proposed. The regularization terms provably enhance the identifiability of the target model parameters in various more challenging cases. A series of synthetic and real data experiments are presented to showcase the effectiveness of our approach.

## 1 Introduction

The success of deep learning has escalated the demand for labeled data to an unprecedented level. Some learning tasks can easily consume millions of labeled data (Najafabadi et al., 2015; Goodfellow et al., 2016). However, acquiring data labels is a nontrivial task—it often requires a pool of annotators with sufficient domain expertise to manually label the data items. For example, the popular Microsoft COCO dataset contains 2.5 million images and around 20,000 work hours aggregated from multiple annotators were used for its category labeling (Lin et al., 2014).

*Crowdsourcing* is considered an important working paradigm for data labeling. In crowdsourcing platforms, e.g., Amazon Mechanical Turk (Buhrmester et al., 2011), Crowdflower (Wazny, 2017), and ClickWork (Vakharia & Lease, 2013), data items are dispatched and labeled by many annotators; the annotations are then integrated to produce reliable labels. A notable challenge is that annotator-output labels are sometimes considerably noisy. Training machine learning models using noisy labels could seriously degrade the system performance (Arpit et al., 2017; Zhang et al., 2016a). In addition, the labels provided by individual annotators are often largely incomplete, as a dataset is often divided and dispatched to different annotators.

Early crowdsourcing methods often treat annotation integration and downstream operations, e.g., classification, as separate tasks; see, (Dawid & Skene, 1979; Karger et al., 2011a; Whitehill et al., 2009; Snow et al., 2008; Welinder et al., 2010; Liu et al., 2012; Zhang et al., 2016b; Ibrahim et al., 2019; Ibrahim & Fu, 2021). This pipeline estimates the annotators' confusion parameters (e.g., the confusion matrices under the Dawid & Skene (DS) model (Dawid & Skene, 1979)) in the first stage.

---

[*]Contact Information: Shahana Ibrahim, Tri Nguyen, and Xiao Fu: {ibrahish, nguyetr9, xiao.fu}@oregonstate.edu

Then, the corrected and integrated labels along with the data are used for training the downstream tasks' classifiers. However, simultaneously learning the annotators' confusions and the classifier in an end-to-end (E2E) manner has shown substantially improved performance in practice. (Raykar et al., 2010; Khetan et al., 2018; Tanno et al., 2019; Rodrigues & Pereira, 2018; Chu et al., 2021; Guan et al., 2018; Cao et al., 2019; Li et al., 2020; Wei et al., 2022; Chen et al., 2020).

To achieve the goal of E2E-based learning under crowdsourced labels, a class of methods concatenate a "confusion layer" for each annotator to the output of a neural classifier, and jointly learn these parts via a parameter-coupled manner. This gives rise to a *coupled cross entropy minimization* (CCEM) criterion (Rodrigues & Pereira, 2018; Tanno et al., 2019; Chen et al., 2020; Chu et al., 2021; Wei et al., 2022). In essence, the CCEM criterion models the observed labels as annotators' confused outputs from the *"ground-truth" predictor* (GTP)—i.e., the classifier as if being trained using the noiseless labels and with perfect generalization—which is natural and intuitive. A notable advantage of the CCEM criteria is that they often lead to computationally convenient optimization problems, as the confusion characteristics are modeled as just additional structured layers added to the neural networks. Nonetheless, the seemingly simple CCEM criterion and its variants have shown promising performance. For example, the *crowdlayer* method is a typical CCEM approach, which has served as a widely used benchmark for E2E crowdsourcing since its proposal (Rodrigues & Pereira, 2018). In (Tanno et al., 2019), a similar CCEM-type criterion is employed, with a trace regularization added. More CCEM-like learning criteria are seen in (Chen et al., 2020; Chu et al., 2021; Wei et al., 2022), with some additional constraints and considerations.

**Challenges.** Apart from its empirical success, understanding of the CCEM criterion has been limited. Particularly, it is often unclear if CCEM can correctly identify the annotators' confusion characteristics (that are often modeled as "confusion matrices") and the GTP under reasonable settings (Rodrigues & Pereira, 2018; Chu et al., 2021; Chen et al., 2020; Wei et al., 2022), but identifiability stands as the key for ensured performance. The only existing identifiability result on CCEM was derived under restricted conditions, e.g., the availability of infinite samples and the assumption that there exists annotators with diagonally-dominant confusion matrices (Tanno et al., 2019). To our best knowledge, model identifiability of CCEM has not been established under realistic conditions, e.g., in the presence of incomplete annotator labeling and non-experts under finite samples.

We should note that a couple of non-CCEM approaches proposed identifiability-guaranteed solutions for E2E crowdsourcing. The work in (Khetan et al., 2018) showed identifiability under their *expectation maximization* (EM)-based learning approach, but the result is only applicable to binary classification. An information-theoretic loss-based learning approach proposed in (Cao et al., 2019) presents some identifiability results, but under the presence of a group of independent expert annotators and infinite samples. These conditions are hard to meet or verify. In addition, the computation of these methods are often more complex relative to the CCEM-based methods.

**Contributions.** Our contributions are as follows:

• **Identifiability Characterizations of CCEM-based E2E Crowdsourcing.** In this work, we show that the CCEM criterion can indeed provably identify the annotators' confusion matrices and the GTP up to inconsequential ambiguities under mild conditions. Specifically, we show that, if the number of annotator-labeled items is sufficiently large and some other reasonable assumptions hold, the two parts can be recovered with bounded errors. Our result is the first finite-sample identifiability result for CCEM-based crowdsourcing. Moreover, our analysis reveals that the success of CCEM does *not* rely on conditional independence among the annotators. This is favorable (as annotator independence is a stringent requirement) and surprising, since conditional independence is often used to derive the CCEM criterion; see, e.g., (Tanno et al., 2019; Rodrigues & Pereira, 2018; Chu et al., 2021).

• **Regularization Design for CCEM With Provably Enhanced Identfiability.** Based on the key insights revealed in our analysis, we propose two types of regularizations that can provide enhanced identifiability guarantees under challenging scenarios. To be specific, the first regularization term ensures that the confusion matrices and the GTP can be identified without having any expert annotators if one has sufficiently large amount of data. The second regularization term ensures identifiability when class specialists are present among annotators. These identifiability-enhanced approaches demonstrate promising label integration performance in our experiments.

**Notation.** The notations are summarized in the supplementary material.

## 2  BACKGROUND

Crowdsourcing is a core working paradigm for labeling data, as individual annotators are often not reliable enough to produce quality labels. Consider a set of $N$ items $\{\boldsymbol{x}_n\}_{n=1}^N$. Each item belongs to one of the $K$ classes. Here, $\boldsymbol{x}_n \in \mathbb{R}^D$ denote the feature vector of $n$th data item. Let $\{y_n\}_{n=1}^N$ denote the set of *ground-truth* labels, where $y_n \in [K]$, $\forall n$. The ground-truth labels $\{y_n\}_{n=1}^N$ are *unknown*. We ask $M$ annotators to provide their labels for their assigned items. Consequently, each item $\boldsymbol{x}_n$ is labeled by a subset of annotators indexed by $\mathcal{S}_n \subseteq [M]$. Let $\{\widehat{y}_n^{(m)}\}_{(m,n)\in\mathcal{S}}$ denote the set of labels provided by all annotators, where $\mathcal{S} = \{(m,n) \mid m \in \mathcal{S}_n, n \in [N]\}$ and $\widehat{y}_n^{(m)} \in [K]$. Note that $|\mathcal{S}| \ll NM$ usually holds and that the annotators may often *incorrectly* label the items, leading to an incomplete and noisy $\{\widehat{y}_n^{(m)}\}_{(m,n)\in\mathcal{S}}$. Here, "incomplete" means that $\mathcal{S}$ does not cover all possible combinations of $(m,n)$—i.e., not all annotators label all items. The goal of an E2E crowdsourcing system is to train a reliable machine learning model (often a classifier) using such noisy and incomplete annotations. This is normally done via blindly estimating the confusion characteristics of the annotators from the noisy labels.

**Classic Approaches.** Besides the naive majority voting method, one of the most influential crowd-sourcing model is by Dawid & Skene (Dawid & Skene, 1979). Dawid & Skene models the confusions of the annotators using the conditional probabilities $\Pr(\widehat{y}_n^{(m)}|y_n)$. Consequently, annotator $m$'s confusion is fully characterized by a matrix defined as follows: $\boldsymbol{A}_m(k, k') := \Pr(\widehat{y}_n^{(m)} = k | y_n = k'), \forall k, k' \in [K]$. Dawid & Skene also proposed an EM algorithm that estimate $\boldsymbol{A}_m$'s under a naive Bayes generative model. Many crowdsourcing methods follow Dawid & Skene's modeling ideas (Ghosh et al., 2011; Dalvi et al., 2013; Karger et al., 2013; Zhang et al., 2016b; Traganitis et al., 2018; Ibrahim et al., 2019; Ibrahim & Fu, 2021). Like (Dawid & Skene, 1979), many of these methods do not exploit the data features $\boldsymbol{x}_n$ while learning the confusion matrices of the annotators. They often treat confusion estimation and classifier training as two sequential tasks. Such two-stage approaches may suffer from error propagation.

**E2E Crowdsourcing and The Identifiability Challenge.** The E2E approaches in (Rodrigues & Pereira, 2018; Tanno et al., 2019; Khetan et al., 2018) model the probability of $m$th annotator's response to the data item $\boldsymbol{x}_n$ as follows:

$$\Pr(\widehat{y}_n^{(m)} = k | \boldsymbol{x}_n) = \sum_{k'=1}^{K} \Pr(\widehat{y}_n^{(m)} = k | y_n = k')\Pr(y_n = k' | \boldsymbol{x}_n), \ k \in [K], \tag{1}$$

where the assumption that the annotator confusion is data-independent has been used to derive the right-hand side. In the above, the distribution $\Pr(\widehat{y}_n^{(m)}|y_n)$ models the confusions in annotator responses given the true label $y_n$ and $\Pr(y_n|\boldsymbol{x}_n)$ denotes the true label distribution of the data item $\boldsymbol{x}_n$. The distribution $\Pr(y_n|\boldsymbol{x}_n)$ can be represented using a mapping $\boldsymbol{f} : \mathbb{R}^D \to [0,1]^K$, $[\boldsymbol{f}(\boldsymbol{x}_n)]_k = \Pr(y_n = k | \boldsymbol{x}_n)$. Define a probability vector $\boldsymbol{p}_n^{(m)} \in [0,1]^K$ for every $m$ and $n$, such that $[\boldsymbol{p}_n^{(m)}]_k = \Pr(\widehat{y}_n^{(m)} = k | \boldsymbol{x}_n)$. With these notations, the following model holds:

$$\boldsymbol{p}_n^{(m)} = \boldsymbol{A}_m \boldsymbol{f}(\boldsymbol{x}_n), \forall m, n. \tag{2}$$

In practice, one does not observe $\boldsymbol{p}_n^{(m)}$. Instead, the annotations $\widehat{y}_n^{(m)}$ are categorical realizations of the distribution $\boldsymbol{p}_n^{(m)}$; i.e.,

$$\widehat{y}_n^{(m)} \sim \mathsf{Cart}(\boldsymbol{p}_n^{(m)}). \tag{3}$$

The goal of E2E crowdsourcing is to identify $\boldsymbol{A}_m$ and $\boldsymbol{f}$ in (2) from noisy labels $\{\widehat{y}_n^{(m)}\}_{(m,n)\in\mathcal{S}}$ and data $\{\boldsymbol{x}_n\}_{n=1}^N$. Note that even if $\boldsymbol{p}_n^{(m)}$ in (2) is observed, $\boldsymbol{A}_m$ and $\boldsymbol{f}(\cdot)$ are in general not *identifiable*. The reason is that one can easily find nonsingular matrices $\boldsymbol{Q} \in \mathbb{R}^{K \times K}$ such that $\boldsymbol{p}_n^{(m)} = \widetilde{\boldsymbol{A}}_m \widetilde{\boldsymbol{f}}(\boldsymbol{x}_n)$, where $\widetilde{\boldsymbol{A}}_m = \boldsymbol{A}_m \boldsymbol{Q}$ and $\widetilde{\boldsymbol{f}}(\boldsymbol{x}_n) = \boldsymbol{Q}^{-1}\boldsymbol{f}(\boldsymbol{x}_n)$. If an E2E approach does not ensure the identifiability of $\boldsymbol{A}_m$ and $\boldsymbol{f}$, but outputs $\widetilde{\boldsymbol{f}}(\boldsymbol{x}_n) = \boldsymbol{Q}^{-1}\boldsymbol{f}(\boldsymbol{x}_n)$, then the learned predictor is not likely to work reasonably. Identifying the ground-truth $\boldsymbol{f}$ only identifies the true labels for the training data (data for which the annotations have been collected), but is useful to predict the true labels for unseen test data items. In addition, correctly identifying $\boldsymbol{A}_m$'s helps quantify how reliable each annotator is.

**CCEM-based Approaches and Challenges.** A number of existing E2E methods are inspired by the model in (2) to formulate CCEM-based learning losses.

• *Crowdlayer* (Rodrigues & Pereira, 2018): For each $(m, n) \in \mathcal{S}$, the generative model in (2) naturally leads to a cross-entropy based learning loss, i.e., $\mathsf{CE}(\boldsymbol{A}_m \boldsymbol{f}(\boldsymbol{x}_n), \widehat{y}_n^{(m)})$.

As some annotators co-label items, the components $\boldsymbol{f}(\boldsymbol{x}_n)$ associated with the co-labeled items are shared by these annotators. With such latent component coupling and putting the cross-entropy losses for all $(m, n) \in \mathcal{S}$ together, the work in (Rodrigues & Pereira, 2018) formulate the following CCEM loss:

$$\underset{\boldsymbol{f} \in \mathcal{F}, \{\boldsymbol{A}_m \in \mathbb{R}^{K \times K}\}}{\text{minimize}} \frac{1}{|\mathcal{S}|} \sum_{(m,n) \in \mathcal{S}} \mathsf{CE}(\mathsf{softmax}(\boldsymbol{A}_m \boldsymbol{f}(\boldsymbol{x}_n)), \widehat{y}_n^{(m)}), \tag{4}$$

where $\mathcal{F} \subseteq \{\boldsymbol{f}(\boldsymbol{x}) \in \mathbb{R}^K | \boldsymbol{f}(\boldsymbol{x}) \in \boldsymbol{\Delta}_K, \forall \boldsymbol{x}\}$ is a function class and the softmax operator is applied as the output of $\boldsymbol{A}_m \boldsymbol{f}(\boldsymbol{x}_n)$ is supposed to be a probability mass function (PMF). The method based on the formulation in (4) was called *crowdlayer* (Rodrigues & Pereira, 2018). The name comes from the fact that the confusion matrices of the "crowd" $\boldsymbol{A}_m$'s can be understood as additional layers if $\boldsymbol{f}$ is represented by a neural network. The loss function in (4) is relatively easy to optimize, as any off-the-shelf neural network training algorithms can be directly applied. This seemingly natural and simple CCEM approach demonstrated substantial performance improvement upon classic methods. However, there is no performance characterization for (4).

• *TraceReg* (Tanno et al., 2019): The work in (Tanno et al., 2019) proposed a similar loss function—with a regularization term in order to establish identifiability of the $\boldsymbol{A}_m$'s and $\boldsymbol{f}$:

$$\underset{\boldsymbol{f} \in \mathcal{F}, \{\boldsymbol{A}_m \in \mathcal{A}\}}{\text{minimize}} \frac{1}{|\mathcal{S}|} \sum_{(m,n) \in \mathcal{S}} \mathsf{CE}(\boldsymbol{A}_m \boldsymbol{f}(\boldsymbol{x}_n), \widehat{y}_n^{(m)}) + \lambda \sum_{m=1}^{M} \mathsf{trace}(\boldsymbol{A}_m), \tag{5}$$

where $\mathcal{A}$ is the constrained set $\{\boldsymbol{A} \in \mathbb{R}^{K \times K} | \boldsymbol{A} \geq \boldsymbol{0}, \boldsymbol{1}^\top \boldsymbol{A} = \boldsymbol{1}^\top\}$. It was shown that (i) if $\boldsymbol{p}_n^{(m)}$ instead of $\widehat{y}_n^{(m)}$ is observed for every $m$ and $n$ and (ii) if the mean of the $\boldsymbol{A}_m$'s are diagonally dominant, then solving (5) with $\mathsf{CE}(\boldsymbol{A}_m \boldsymbol{f}(\boldsymbol{x}_n), \widehat{y}_n^{(m)})$ replaced by enforcing $\boldsymbol{p}_n^{(m)} = \boldsymbol{A}_m \boldsymbol{f}(\boldsymbol{x}_n)$ ensures identifying $\boldsymbol{A}_m$ for all $m$, up to column permutations. Even though, the result provides some justification for using CCEM criterion under trace regularization, the result is clearly unsatisfactory, as it requires many stringent assumptions to establish identifiability.

• *SpeeLFC* (Chen et al., 2020), *Union-Net* (Wei et al., 2022), *CoNAL* (Chu et al., 2021): A number of other variants of the CCEM criterion are also proposed for E2E learning with crowdsourced labels. The work in (Chen et al., 2020) uses a similar criterion as in (4), but with the softmax operator applied on the columns of $\boldsymbol{A}_m$'s, instead on the output of $\boldsymbol{A}_m \boldsymbol{f}(\boldsymbol{x}_n)$. The work in (Wei et al., 2022) concatenates $\boldsymbol{A}_m$'s vertically and applies the softmax operator on the columns of such concatenated matrix, while formulating the CCEM criterion. The method in (Chu et al., 2021) models a common confusion matrix in addition to the annotator-specific confusion matrices $\boldsymbol{A}_m$'s and employs a CCEM-type criterion to learn both in a coupled fashion. Nonetheless, none of these approaches have successfully tackled the identifiability aspect. The work (Wei et al., 2022) claims theoretical support for their approach, but the proof is flawed. In fact, understanding to the success of CCEM-based methods under practical E2E learning settings is still unclear.

**Other Related Works.** A couple of non-CCEM approaches have also considered the identifiability aspects (Khetan et al., 2018; Cao et al., 2019). The work in (Khetan et al., 2018) presents an EM-inspired alternating optimization strategy. Such procedure involves training the neural classifier multiple times, making it a computationally intensive approach. In addition, its identifiability guarantees are under certain strict assumptions, e.g., binary classification, identical confusions and conditional independence for annotators—which are hard to satisfy in practice. The approach in (Cao et al., 2019) employs an information-theoretic loss function to jointly train a predictor network and annotation aggregation network. The identifiability guarantees are again under restrictive assumptions, e.g., infinite data, no missing annotations and existence of mutually independent expert annotators.

We should also note that there exist works considering data-dependent annotator confusions as well, e.g., (Zhang et al., 2020) (multiple annotator case) and (Cheng et al., 2021; Xia et al., 2020; Zhu et al., 2022) (single annotator case)—leveraging more complex models and learning criteria. In our work, we focus on the criterion designed using a model in (1) as it is shown to be effective in practice.

## 3 IDENTIFIABILITY OF CCEM-BASED E2E CROWDSOURCING

In this section, we offer identifiability analysis of the CCEM learning loss under realistic settings. We consider the following re-expressed objective function of CCEM:

$$
\underset{\boldsymbol{f}, \{\boldsymbol{A}_m\}}{\text{minimize}} \quad -\frac{1}{|\mathcal{S}|} \sum_{(m,n) \in \mathcal{S}} \sum_{k=1}^{K} \mathbb{I}[\widehat{y}_n^{(m)} = k] \log[\boldsymbol{A}_m \boldsymbol{f}(\boldsymbol{x}_n)]_k \tag{6a}
$$

$$
\text{subject to} \quad \boldsymbol{f} \in \mathcal{F}, \ \boldsymbol{A}_m \in \mathcal{A}, \ \forall m, \tag{6b}
$$

where $\mathcal{F} \subseteq \{\boldsymbol{f}(\boldsymbol{x}) \in \mathbb{R}^K \mid \boldsymbol{f}(\boldsymbol{x}) \in \boldsymbol{\Delta}_K, \ \forall \boldsymbol{x}\}$ is a function class and $\mathcal{A}$ is the constrained set $\{\boldsymbol{A} \in \mathbb{R}^{K \times K} \mid \boldsymbol{A} \geq \boldsymbol{0}, \boldsymbol{1}^\top \boldsymbol{A} = \boldsymbol{1}^\top\}$, since each column of $\boldsymbol{A}_m$'s are conditional probability distributions.

The rationale is that the confusion matrices $\boldsymbol{A}_m$'s act as correction terms for annotator's labeling noise to output a true label classifier $\boldsymbol{f}$. Intuitively, the objective in (6a) encourages the output estimates $\widehat{\boldsymbol{f}}, \{\widehat{\boldsymbol{A}}_m\}$ to satisfy the relation $\boldsymbol{p}_n^{(m)} = \widehat{\boldsymbol{A}}_m \widehat{\boldsymbol{f}}(\boldsymbol{x}_n), \ \forall m, n$. One can easily note that the identifiability guarantees for $\boldsymbol{A}_m$ and $\boldsymbol{f}$ ( i.e., when does the ground-truth $\boldsymbol{A}_m$ and the ground-truth $\boldsymbol{f}$ can be identified) are not so straightforward since there exists an infinite number of nonsingular matrices $\boldsymbol{Q} \in \mathbb{R}^{K \times K}$ such that $\boldsymbol{p}_n^{(m)} = (\widehat{\boldsymbol{A}}_m \boldsymbol{Q})(\boldsymbol{Q}^{-1} \widehat{\boldsymbol{f}}(\boldsymbol{x}_n))$.

To proceed, we make the following assumptions:

**Assumption 1** *Each data item $\boldsymbol{x}_n$, $n \in [N]$ is drawn from a distribution $\mathcal{D}$ independently at random and has a bounded $\ell_2$ norm. The observed index pairs $(m,n)$ are included in $\mathcal{S}$ uniformly at random.*

**Assumption 2** *The neural network function class $\mathcal{F}$ has a complexity measure denoted as $\mathscr{R}_{\mathcal{F}}$.*

Here, we use the *sensitive complexity* parameter introduced in (Lin & Zhang, 2019) as the complexity measure $\mathscr{R}_{\mathcal{F}}$. In essence, $\mathscr{R}_{\mathcal{F}}$ gets larger as the neural network function class $\mathcal{F}$ gets deeper and wider—also see supplementary material Sec. K for more discussion.

**Assumption 3** *There exists a GTP $\boldsymbol{f}^{\natural} : \mathbb{R}^D \to [0,1]^K$, such that $[\boldsymbol{f}^{\natural}(\boldsymbol{x}_n)]_k = \Pr(y_n = k \mid \boldsymbol{x}_n)$, where $y_n$ denotes the true label of the data item $\boldsymbol{x}_n$. In addition, there exists $\widetilde{\boldsymbol{f}} \in \mathcal{F}$ such that $\|\widetilde{\boldsymbol{f}}(\boldsymbol{x}_n) - \boldsymbol{f}^{\natural}(\boldsymbol{x}_n)\|_2 \leq \nu$ for all $\boldsymbol{x}_n \sim \mathcal{D}$, where $0 \leq \nu < \infty$.*

**Assumption 4** *(Near-Class Specialist Assumption) Let $\boldsymbol{A}_m^{\natural}$ denote the ground-truth confusion matrix for annotator $m$. For each class $k$, there exists a near-class specialist, indexed by $m_k$, such that $\|\boldsymbol{A}_{m_k}^{\natural}(k,:) - \boldsymbol{e}_k^\top\|_2 \leq \xi_1$, where $0 \leq \xi_1 < \infty$.*

**Assumption 5** *(Near-Anchor Point Assumption) For each class $k$, there exists a near-anchor point, $\boldsymbol{x}_{n_k}$, such that $\|\boldsymbol{f}^{\natural}(\boldsymbol{x}_{n_k}) - \boldsymbol{e}_k\|_2 \leq \xi_2$), where $0 \leq \xi_2 < \infty$.*

Assumptions 1-3 are standard conditions for analyzing performance of neural models under finite sample settings. The Near-Class Specialist Assumption (NCSA) implies that there exist annotators for each class $k \in [K]$ who can correctly distinguish items belonging to class $k$ from those belonging to other classes. It is important to note that NCSA assumption is much milder compared to the commonly used assumptions such as the existence of all-class expert annotators (Cao et al., 2019) or diagonally dominant $\boldsymbol{A}_m^{\natural}$'s (Tanno et al., 2019). Instead, the NCSA only requires that for each class, there exists one annotator who is specialized for it (i.e., class-specialist)—but the annotator needs not be an all-class expert; see Fig. 1 in the supplementary material for illustration. The Near-Anchor Point Assumption (NAPA) is the relaxed version of the exact anchor point assumption commonly used in provable label noise learning methods (Xia et al., 2019; Li et al., 2021). Under Assumptions 1-5, we have the following result:

**Theorem 1** *Assume that each $[\boldsymbol{A}_m^{\natural} \boldsymbol{f}^{\natural}(\boldsymbol{x}_n)]_k$ and $[\boldsymbol{A}_m \boldsymbol{f}(\boldsymbol{x}_n)]_k$, $\forall \boldsymbol{A}_m \in \mathcal{A}, \forall \boldsymbol{f} \in \mathcal{F}$ are at least $(1/\beta)$. Also assume that $\sigma_{\max}(\boldsymbol{A}_m^{\natural}) \leq \sigma$, $\forall m$, for a certain $\sigma > 0$. Then, for any $\alpha > 0$, with*

*probability greater than* $1 - K/N^\alpha$, *any optimal solution* $\widehat{A}_m$ *and* $\widehat{f}$ *of the problem* (6) *satisfies the following relations:*

$$\min_{\boldsymbol{\Pi}} \|\widehat{\boldsymbol{A}}_m - \boldsymbol{A}_m^\natural \boldsymbol{\Pi}\|_{\mathrm{F}}^2 = K\sigma^2(\eta + \xi_1 + \xi_2), \ \forall m \in [M], \tag{7a}$$

$$\mathbb{E}_{\boldsymbol{x} \sim \mathcal{D}} \left[ \min_{\boldsymbol{\Pi}} \|\widehat{\boldsymbol{f}}(\boldsymbol{x}) - \boldsymbol{\Pi}^\top \boldsymbol{f}^\natural(\boldsymbol{x})\|_2^2 \right] = K(\eta + \xi_1 + \xi_2), \tag{7b}$$

*where* $\eta^2 = \mathcal{O}\left(\beta M N^\alpha/\sqrt{S} \left(\sqrt{M \log S} + (\|\boldsymbol{X}\|_{\mathrm{F}} \mathscr{R}_{\mathcal{F}})^{\frac{1}{4}}\right) + \beta\sqrt{K} M N^\alpha \nu\right)$, $\boldsymbol{\Pi} \in \{0, 1\}^K$ *is a permutation matrix, and* $\boldsymbol{X} = [\boldsymbol{x}_{n_1}, \ldots, \boldsymbol{x}_{n_S}]$, $(m_s, n_s) \in \mathcal{S}$, *if conditions* $\xi_1, \xi_2 \leq 1/K$, $\nu \leq 1/\beta K^2 M^2 N^\alpha$, *and* $S = |\mathcal{S}| = \Omega\left(\beta^2 M^2 N^{2\alpha} K^2 \max\left(M \log S, \sqrt{\|\boldsymbol{X}\|_{\mathrm{F}} \mathscr{R}_{\mathcal{F}}}\right)\right)$ *hold.*

The proof of Theorem 1 is relegated to the supplementary material in Sec. B. The takeaways are as follows: First, the CCEM can provably identify the $\boldsymbol{A}_m^\natural$'s and the GTP $\boldsymbol{f}^\natural$ under finite samples—and such finite-sample identifiability of CCEM was not established before. Second, some restrictive conditions (e.g., the existence of all-class experts (Tanno et al., 2019; Cao et al., 2019)) used in the literature for establishing CCEM's identifiability are actually not needed. Third, many works derived the CCEM criterion under the maximum likelihood principle via assuming that the annotators are conditionally independent; see (Rodrigues & Pereira, 2018; Chen et al., 2020; Tanno et al., 2019; Chu et al., 2021). Interestingly, as we revealed in our analysis, the identifiability of $\boldsymbol{A}_m^\natural$ and $\boldsymbol{f}^\natural$ under the CCEM criterion does *not* rely on the annotators' independence. These new findings support and explain the effectiveness of CCEM in a wide range of scenarios observed in the literature.

## 4 ENHANCING IDENTIFIABILITY VIA REGULARIZATION

Theorem 1 confirms that CCEM is a provably effective criterion for identifying $\boldsymbol{A}_m^\natural$'s and $\boldsymbol{f}^\natural$. The caveats lie in Assumptions 4 and 5. Essentially, these assumptions require that some rows of the collection $\{\boldsymbol{A}_m^\natural\}_{m=1}^M$ and some vectors of the collection $\{\boldsymbol{f}^\natural(\boldsymbol{x}_n)\}_{n=1}^N$ are close to the canonical vectors. These are reasonable assumptions, but satisfying them simultaneously may not always be possible. A natural question is if these assumptions can be relaxed, at least partially.

In this section, we propose variants of CCEM that admits enhanced identifiability. To this end, we use a condition that is often adopted in the nonnegative matrix factorization (NMF) literature:

**Definition 1** (SSC)(Fu et al., 2019; Gillis, 2020) *Consider the second-order cone* $\mathcal{C} = \{\boldsymbol{x} \in \mathbb{R}^K | \sqrt{K-1}\|\boldsymbol{x}\|_2 \leq \mathbf{1}^\top \boldsymbol{x}\}$. *A nonnegative matrix* $\boldsymbol{Z} \in \mathbb{R}_+^{L \times K}$ *satisfies the sufficiently scattered condition (SSC) if (i)* $\mathcal{C} \subseteq \mathrm{cone}(\boldsymbol{Z}^\top)$ *and (ii)* $\mathrm{cone}\{\boldsymbol{Z}^\top\} \subseteq \mathrm{cone}\{\boldsymbol{Q}\}$ *does not hold for any orthogonal* $\boldsymbol{Q} \in \mathbb{R}^{K \times K}$ *except for the permutation matrices.*

Geometrically, the matrix $\boldsymbol{Z}$ satisfying the SSC means that its rows span a large "area" within the nonnegative orthant, so that the conic hull spanned by the rows contains the second order cone $\mathcal{C}$ as a subset. Note that the SSC condition subsumes the the NCSA (Assumption (4) where $\xi_1 = 0$) and the NAPA (Assumption (5) where $\xi_1 = 0$) as special cases.

Using SSC, we first show that the CCEM criterion in (6) attains identifiability of $\boldsymbol{A}_m^\natural$'s and $\boldsymbol{f}^\natural$ without using the NCSA or the NAPA, when the problem size grows sufficiently large.

**Theorem 2** *Suppose that the incomplete labeling paradigm in Assumption 1 holds with* $S = \Omega(t)$ *and* $N = O(t^3)$, *for a certain* $t > 0$. *Also, assume that there exists* $\mathcal{Z} = \{n_1, \ldots, n_T\}$ *such that* $\boldsymbol{F}_{\mathcal{Z}}^\natural = [\boldsymbol{f}^\natural(\boldsymbol{x}_{n_1}), \ldots, \boldsymbol{f}^\natural(\boldsymbol{x}_{n_T})]$ *satisfies the SSC and* $\boldsymbol{W}^\natural = [\boldsymbol{A}_1^{\natural\top}, \ldots, \boldsymbol{A}_M^{\natural\top}]$ *satisfies the SSC as well. Then, at the limit of* $t \to \infty$, *if* $\boldsymbol{f}^\natural \in \mathcal{F}$, *the optimal solutions* $(\{\widehat{\boldsymbol{A}}_m\}, \widehat{\boldsymbol{f}})$ *of (6) satisfies* $\widehat{\boldsymbol{A}}_m = \boldsymbol{A}_m^\natural \boldsymbol{\Pi}$ *for all* $m$ *and* $\widehat{\boldsymbol{f}}(\boldsymbol{x}) = \boldsymbol{\Pi}^\top \boldsymbol{f}^\natural(\boldsymbol{x})$ *for all* $\boldsymbol{x} \sim \mathcal{D}$, *where* $\boldsymbol{\Pi}$ *is a permutation matrix.*

The proof is via connecting the CCEM problem to the classic nonnegative matrix factorization (NMF) problem at the limit of $t \to \infty$; see Sec. D. It is also worth to note that Theorem 2 does not require complete observation of the annotations to guarantee identifiability. Another remark is that $\boldsymbol{W}^\natural$ and $\boldsymbol{F}^\natural$ satisfying the SSC is more relaxed compared to that they satisfying the NCSA and NAPA,

respectively. For example, one may need class specialists with much higher expertise to satisfy NCSA than to satisfy SSC on $\boldsymbol{W}^\natural$—also see geometric illustration in Sec. L. Compared to the result in Theorem 1, Theorem 2 implies that when $N$ is very large, the CCEM should work under less stringent conditions relative to the NCSA and NAPA. Nonetheless, the assumption that both the GTP and the annotators' confusion matrices should meet certain requirements may not always hold. In the following, we further show that the SSC on either $\boldsymbol{W}^\natural$ or $\boldsymbol{F}^\natural$ can be relaxed:

**Theorem 3** *Suppose that the incomplete labeling paradigm in Assumption 1 holds with $S = \Omega(t)$ and $N = O(t^3)$, for a certain $t > 0$. Then, at the limit of $t \to \infty$, if $\boldsymbol{f}^\natural \in \mathcal{F}$, we have $\boldsymbol{A}_m^* = \boldsymbol{A}_m^\natural \boldsymbol{\Pi}$ for all $m$ and $\boldsymbol{f}^*(\boldsymbol{x}) = \boldsymbol{\Pi}^\top \boldsymbol{f}^\natural(\boldsymbol{x})$ for all $\boldsymbol{x} \sim \mathcal{D}$, when any of the following conditions hold:*

*(a) If there exists $\mathcal{Z} = \{n_1, \ldots, n_T\}$ such that $(\boldsymbol{F}_\mathcal{Z}^\natural)^\top = [\boldsymbol{f}^\natural(\boldsymbol{x}_{n_1}), \ldots, \boldsymbol{f}^\natural(\boldsymbol{x}_{n_T})]^\top$ satisfies the SSC, $\mathrm{rank}(\boldsymbol{W}^\natural) = K$, and $(\{\boldsymbol{A}_m^*\}, \boldsymbol{f}^*)$ is the optimal solution of (6) with the maximal $\log\det(\boldsymbol{F}^* \boldsymbol{F}^{*\top})$.*

*(b) If $\boldsymbol{W}^\natural = [\boldsymbol{A}_1^{\natural\top}, \ldots, \boldsymbol{A}_M^{\natural\top}]^\top$ satisfies the SSC, $\mathrm{rank}(\boldsymbol{F}^\natural) = K$, and $(\boldsymbol{A}_m^*, \boldsymbol{f}^*)$ is the optimal solution of (6) with the maximal $\log\det((\boldsymbol{W}^*)^\top \boldsymbol{W}^*)$.*

Theorem 3(a) shows that if the number of annotated data items is large and the GTP-outputs associated with $\boldsymbol{x}_n$ are diverse enough to satisfy the SSC, then $\boldsymbol{f}^\natural$ and $\boldsymbol{A}_m^\natural$ are identifiable even if there are no class specialists available. Compared to Theorem 2, the identifiability is clearly enhanced, as the conditions on the annotators have been relaxed. Theorem 3(b) implies that if $\boldsymbol{W}^\natural$ satisfies SSC, identifiability can be established irrespective of the geometry of $\boldsymbol{F}^\natural$. Intuitively, when there are more annotators available, $\boldsymbol{W}^\natural$ is more likely to satisfy SSC (the relation between the size of $\boldsymbol{W}^\natural$ and SSC was shown in (Ibrahim et al., 2019, Theorem 4) under reasonable conditions). Hence, in practice, Theorem 3(b) may offer ensured performance in cases where more annotators are available.

Similar to the proof in Theorem 2, the proof here connects the CCEM-based E2E crowdsourcing problem to the simplex-structured matrix factorization (SSMF) problem (see (Fu et al., 2015; 2018)) at the limit of $t \to \infty$. The detailed proof is provided in Sec. E. We should mention that various connections between matrix factorization models and data labeling problems were also observed in related prior works, e.g, the classic approaches based on Dawid & Skene model (Dawid & Skene, 1979; Zhang et al., 2016b; Ibrahim et al., 2019; Ibrahim & Fu, 2021) and E2E learning under noisy labels (Li et al., 2021). Nonetheless, the former does not consider cross-entropy losses or involve any feature extractor, whereas, the latter does not consider multiple annotators or incomplete labeling.

**Implementation via Regularization** Theorem 3 naturally leads to regularized versions of CCEM. Correspondingly, we have the following two cases:

Following Theorem 3(a), when the GTP is believed to be diverse over different $\boldsymbol{x}_n$'s, we propose to employ the following regularized CCEM:

$$\underset{\boldsymbol{A}_m \in \mathcal{A}, \forall m, \ \boldsymbol{f} \in \mathcal{F}}{\text{minimize}} \quad -\frac{1}{|\mathcal{S}|} \sum_{(m,n) \in \mathcal{S}} \sum_{k=1}^{K} \mathbb{I}[\widehat{y}_n^{(m)} = k] \log[\boldsymbol{A}_m \boldsymbol{f}(\boldsymbol{x}_n)]_k - \lambda \log\det \boldsymbol{F} \boldsymbol{F}^\top, \quad (8)$$

where $\boldsymbol{F} = [\boldsymbol{f}(\boldsymbol{x}_1), \ldots, \boldsymbol{f}(\boldsymbol{x}_N)]^\top$. In similar spirit, following Theorem 3(b), we propose the following regularized CCEM when $M$ is large (so that $\boldsymbol{W}^\natural$ likely satisfies SSC):

$$\underset{\boldsymbol{A}_m \in \mathcal{A}, \forall m, \ \boldsymbol{f} \in \mathcal{F}}{\text{minimize}} \quad -\frac{1}{|\mathcal{S}|} \sum_{(m,n) \in \mathcal{S}} \sum_{k=1}^{K} \mathbb{I}[\widehat{y}_n^{(m)} = k] \log[\boldsymbol{A}_m \boldsymbol{f}(\boldsymbol{x}_n)]_k - \lambda \log\det \boldsymbol{W}^\top \boldsymbol{W}, \quad (9)$$

where $\boldsymbol{W} = [\boldsymbol{A}_1^\top, \ldots, \boldsymbol{A}_M^\top]^\top$. The implementations for (8) and (9) are called as geometry-regularized crowdsourcing network, abbreviated as `GeoCrowdNet(F)` and `GeoCrowdNet(W)`, respectively.

**Remark 1** Based on our analyses, our suggested *rule of thumb* for choosing regularization is as follows: When $M$ is large but $N$ is relatively small, using `GeoCrowdNet(W)` is recommended, as $\boldsymbol{W}$ is more likely to satisfy the SSC compared to $\boldsymbol{F}^\top$. However, when $N$ is large, using `GeoCrowdNet(F)` is expected to have a better performance as $\boldsymbol{F}^\top$ is likely to satisfy the SSC in this case. Under big data settings, $N$ is often large and the GTP outputs of $\boldsymbol{x}_n$'s are often reasonably diverse. Hence, (8) oftentimes works well, as will be seen in our experiments. Nonetheless, for cases where the geometry of $\boldsymbol{F}^\top$ violates the SSC, the employment of more annotators can help via using (9)—which is also an intuitive advantage of crowdsourcing.

Table 1: Average test accuracy ($\pm$ std) of the proposed methods and the baselines on MNIST & Fashion-MNIST dataset under various $(N, M)$'s; labels are produced by machine annotators; $p = 0.1$.

| Methods | MNIST | | | | Fashion-MNIST | | | |
|---|---|---|---|---|---|---|---|---|
| | Case 1 | | Case 2 | | Case 1 | | Case 2 | |
| | $(1000, 15)$ | $(1000, 20)$ | $(5000, 5)$ | $(10000, 5)$ | $(1000, 15)$ | $(1000, 30)$ | $(5000, 5)$ | $(10000, 5)$ |
| GeoCrowdNet(F) | $79.89 \pm 3.08$ | $82.18 \pm 3.48$ | $\mathbf{85.92 \pm 2.73}$ | $\mathbf{87.21 \pm 2.47}$ | $78.98 \pm 2.83$ | $84.47 \pm 1.64$ | $\mathbf{80.60 \pm 0.46}$ | $\mathbf{83.68 \pm 2.17}$ |
| GeoCrowdNet(W) | $80.97 \pm 1.31$ | $83.69 \pm 2.37$ | $77.79 \pm 8.97$ | $82.37 \pm 9.18$ | $79.80 \pm 4.23$ | $85.56 \pm 1.91$ | $72.36 \pm 3.84$ | $74.03 \pm 7.41$ |
| GeoCrowdNet($\lambda = 0$) | $71.15 \pm 6.73$ | $69.17 \pm 2.61$ | $71.66 \pm 4.48$ | $60.29 \pm 7.91$ | $70.92 \pm 4.14$ | $81.88 \pm 4.41$ | $69.31 \pm 4.77$ | $73.04 \pm 7.56$ |
| TraceReg | $70.14 \pm 6.93$ | $78.09 \pm 3.52$ | $63.71 \pm 10.76$ | $64.45 \pm 10.14$ | $75.06 \pm 3.43$ | $83.79 \pm 2.15$ | $69.82 \pm 6.46$ | $72.21 \pm 8.44$ |
| CrowdLayer | $65.72 \pm 4.81$ | $72.90 \pm 2.31$ | $51.25 \pm 6.24$ | $53.12 \pm 14.02$ | $63.91 \pm 2.11$ | $76.50 \pm 3.68$ | $64.73 \pm 6.89$ | $63.18 \pm 3.55$ |
| MBEM | $37.24 \pm 8.54$ | $33.39 \pm 5.72$ | $28.14 \pm 6.24$ | $26.90 \pm 2.01$ | $23.14 \pm 2.36$ | $25.43 \pm 5.68$ | $37.34 \pm 3.73$ | $38.62 \pm 5.45$ |
| CoNAL | $51.33 \pm 4.27$ | $55.59 \pm 5.32$ | $44.79 \pm 7.01$ | $39.41 \pm 9.21$ | $59.76 \pm 3.34$ | $71.44 \pm 6.22$ | $52.15 \pm 6.08$ | $54.61 \pm 7.54$ |
| Max-MIG | $\mathbf{81.32 \pm 1.31}$ | $83.34 \pm 3.36$ | $83.71 \pm 1.23$ | $81.05 \pm 0.65$ | $70.95 \pm 5.61$ | $81.27 \pm 5.37$ | $73.45 \pm 5.21$ | $73.62 \pm 3.12$ |
| NN-MV | $78.98 \pm 3.21$ | $82.78 \pm 1.45$ | $83.44 \pm 1.98$ | $83.10 \pm 2.86$ | $62.61 \pm 6.38$ | $73.61 \pm 3.88$ | $61.95 \pm 3.73$ | $71.64 \pm 2.25$ |
| NN-DSEM | $73.95 \pm 4.90$ | $77.57 \pm 4.06$ | $65.81 \pm 2.74$ | $69.09 \pm 3.21$ | $55.89 \pm 5.97$ | $73.64 \pm 4.44$ | $31.40 \pm 3.86$ | $35.10 \pm 4.89$ |

## 5 EXPERIMENTS

**Baselines.** The proposed methods are compared with a number of existing E2E crowdsourcing methods, namely, `TraceReg`(Tanno et al., 2019), `MBEM` (Khetan et al., 2018), `CrowdLayer` (Rodrigues & Pereira, 2018), `CoNAL` (Chu et al., 2021), and `Max-MIG` (Cao et al., 2019). In addition to these baselines, we also learn neural network classifiers using the labels aggregated from *majority voting* and the classic EM algorithm proposed by Dawid &S kene (Dawid & Skene, 1979), denoted as `NN-MV` and `NN-DSEM`, respectively.

**Real-Data Experiments with Machine Annotations.** The settings are as follows:

*Dataset.* We use the MNIST dataset (Deng, 2012) and Fashion-MNIST dataset (Xiao et al., 2017); see more details in Sec. N.

*Noisy Label Generation.* We train a number of machine classifiers to produce noisy labels. Five types of classifiers, including support vector machines (SVM), $k$-Nearest Neighbour (kNN), logistic regression, convolutional neural network (CNN), and fully connected neural network are considered to act as annotators. To create more annotators, we change some of their parameters (e.g., the number of nearest neighbour parameter $k$ of kNN and the number of epochs for training the CNN). We consider two different cases: *(i) Case 1: NCSA (Assumption 4) holding:* Each annotator is chosen to be an all-class expert with probability $0.1$. If an annotator is chosen to be an all-class expert, it is trained carefully so that its classification accuracy exceeds a certain threshold; see details in Sec. N. Note that the existence of an expert implies that the NCSA and the SSC are likely satisfied by $\boldsymbol{W}^\natural$, but the reverse is not necessarily true. We use this strategy to enforce NCSA only for validation purpose. It does not mean that the CCEM criterion needs the existence of experts to work. *(ii) Case 2: NCSA not holding:* Each machine annotator is trained by randomly choosing a small subset of the training data (with 100 to 500 samples) so that the accuracy of these machine annotators are low. This way, any all-class expert or class-specialist is unlikely to exist.

We use the two cases to validate our theorems. Under our analyses, `GeoCrowdNet(W)` is expected to work better under Case 1, as NCSA approximately implies SSC. In addition, `GeoCrowdNet(F)` does not rely on the geometry of $\boldsymbol{W}^\natural$, and thus can work under both cases, if $N$ is reasonably large (which implies that $(\boldsymbol{F}^\natural)^\top$ is more likely to satisfy the SSC). Once the machine annotators are trained, we let them label unseen data items of size $N$. To evaluate the methods under incomplete labeling, an annotator labels any data item with probability $p = 0.1$. Under this labeling strategy, every data item is only labeled by a small subset of the annotators.

*Settings.* The neural network classifier architecture for MNIST dataset is chosen to be Lenet-5 (Lecun et al., 1998), which consists of two sets of convolutional and max pooling layers, followed by a flattening convolutional layer, two fully-connected layers and finally a softmax layer. For Fashion-MNIST dataset, we use the ResNet-18 architecture (He et al., 2016). Adam (Kingma & Ba, 2015) is used an optimizer with weight decay of $10^{-4}$ and batch size of 128. The regularization parameter $\lambda$ and the initial learning rate of the Adam optimizer are chosen via grid search method using the validation set from $\{0.01, 0.001, 0.0001\}$ and $\{0.01, 0.001\}$, respectively. We choose the same neural network structures for all the baselines. The confusion matrices are initialized with identity matrices of size $K$ for proposed methods and the baselines `TraceReg` and `CrowdLayer`.

*Results.* Table 1 presents the average label prediction accuracy on the testing data of the MNIST and the Fashion-MNIST over 5 random trials, for various cases. One can observe that `GeoCrowdNet(F)`

Table 2: Average test accuracy of the proposed methods and the baselines on LabelMe ($M = 59, K = 8$) and Music ($M = 44, K = 10$) datasets.

| Methods | LabelMe | Music |
|---|---|---|
| GeoCrowdNet(F) | $\mathbf{85.85 \pm 0.33}$ | $\mathbf{67.13 \pm 1.27}$ |
| GeoCrowdNet(W) | $81.59 \pm 0.73$ | $66.46 \pm 1.77$ |
| GeoCrowdNet ($\lambda$=0) | $80.35 \pm 0.40$ | $65.93 \pm 1.81$ |
| TraceReg | $80.89 \pm 0.73$ | $66.40 \pm 0.90$ |
| Crowdlayer | $81.41 \pm 0.90$ | $65.53 \pm 1.74$ |
| MBEM | $80.05 \pm 3.31$ | $\mathbf{69.00 \pm 2.15}$ |
| CoNAL | $81.68 \pm 2.72$ | $60.93 \pm 0.61$ |
| Max-MIG | $\mathbf{85.03 \pm 0.44}$ | $64.33 \pm 2.31$ |
| NN-MV | $79.57 \pm 1.24$ | $63.40 \pm 2.68$ |
| NN-DSEM | $78.26 \pm 1.18$ | $64.66 \pm 2.34$ |

performs the best when there are more number of annotated items, even when there are no class specialist annotators present (i.e., case 2). On the other hand, GeoCrowdNet(W) starts showing its advantage over GeoCrowdNet(F) when there are more number of annotators and class specialists among them. Both are consistent with our theorems. In addition, the baseline MaxMIG performs competitively when there are all-class experts—which is consistent with their identifiability analysis. However, when there are no such experts (case 2), its performance drops compared to the proposed methods, especially GeoCrowdNet(F) whose identifiability does not rely on the existence of all-class experts or class specialists. Overall, GeoCrowdNet(F) exhibits consistently good performance.

**Real-Data Experiments with Human Annotators.** The settings are as follows:

*Datasets.* We consider two different datasets, namely, LabelMe ($M = 59$) (Rodrigues et al., 2017; Russell et al., 2007) and Music ($M = 44$) (Rodrigues et al., 2014), both having noisy and incomplete labels provided by human workers from AMT.

*Settings.* For LabelMe, we employ the pretrained VGG-16 embeddings followed by a fully connected layer as used in (Rodrigues & Pereira, 2018). For Music, we choose a similar architecture, but with batch normalization layers. We use a batch size of 128 for LabelMe and a batch size of 100 for Music dataset. All other settings are the same as the machine annotator case. More details of the datasets and architecture can be seen in Sec. N.

*Results.* Table 2 shows the average label prediction accuracies on the test data for 5 random trials. One can see that GeoCrowdNet ($\lambda$=0) works reasonably well relative to the machine annotation case in the previous experiment. This may be because the number of annotators for both LabelMe and Music are reasonably large ($M = 59$ and $M = 44$, respectively), which makes $\boldsymbol{W}^\natural$ satisfying Assumption 4 become easier than before. This validates our claims in Theorem 1, but also shows the limitations of the plain-vanilla CCEM—that is, it may require a fairly large number of annotators to start showing promising results. Similar to the previous experiment, both GeoCrowdNet(F) and GeoCrowdNet(W) outperform the unregularized version GeoCrowdNet ($\lambda$=0), showing the advantages of the identifiability-enhancing regularization terms. More experiments and details are presented in Sec. N, due to page limitations.

## 6  CONCLUSION

In this work, we revisited the CCEM criterion—one of the most popular E2E learning criteria for crowdsourced label integration. We provided the first finite-sample identifiability characterization of the confusion matrices and the neural classifier which are learned using the CCEM criterion. Compared to many exiting identifiability results, our guarantees are under more relaxed and more realistic settings. In particular, a take-home point revealed in our analysis is that CCEM can provably identify the desired model parameters even if the annotators are dependent, which is a surprising but favorable result. We also proposed two regularized variants of the CCEM, based on the insights learned from our identifiability analysis for the plain-vanilla CCEM. The regularized CCEM criteria provably enhance the identifiability of the confusion matrices and the neural classifier under more challenging scenarios. We evaluated the proposed approaches on various synthetic and real datasets. The results corroborate our theoretical claims.

**Acknowledgement.** This work is supported in part by the National Science Foundation under Project NSF IIS-2007836.

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

**Supplementary Material of " Deep Learning From Crowdsourced Labels: Coupled Cross-entropy Minimization, Identifiability, and Regularization"**

## A   NOTATION

$x$, $\boldsymbol{x}$, and $\boldsymbol{X}$ represent a scalar, a vector, and a matrix, respectively. $[\boldsymbol{x}]_i$ and $\boldsymbol{x}(i)$ both denote the $i$th entry of the vector $\boldsymbol{x}$. $\boldsymbol{X}(:, j)$ denotes the $j$th column vector of $\boldsymbol{X}$ and $\boldsymbol{X}(i, :)$ denotes the $i$th row vector of $\boldsymbol{X}$. $[\boldsymbol{X}]_{i,j}$ and $\boldsymbol{X}(i, j)$ both mean the $(i, j)$th entry of $\boldsymbol{X}$. $\|\boldsymbol{x}\|_2$ and $\|\boldsymbol{X}\|_{\mathrm{F}}$ mean the Euclidean (Frobenius) norm of the augment. $[I]$ means an integer set $\{1, 2, \ldots, I\}$. $^{\top}$ denote transpose. $\boldsymbol{\Delta}_K = \{\boldsymbol{x} \in \mathbb{R}^K : \sum_i \boldsymbol{x}(i) = 1, \boldsymbol{x} \geq \boldsymbol{0}\}$ denotes the probability simplex. $\boldsymbol{X} \geq \boldsymbol{0}$ implies that all the entries of the matrix $\boldsymbol{X}$ are nonnegative. $\mathbb{I}[A]$ denotes an indicator function for the event $A$ such that $\mathbb{I}[A] = 1$ if the event $A$ happens, otherwise $\mathbb{I}[A] = 0$. $\mathsf{CE}(\boldsymbol{x}, y) = -\sum_{k=1}^{K} \mathbb{I}[y = k] \log(\boldsymbol{x}(k))$ denotes the cross entropy function. $\boldsymbol{I}_K$ denotes an identity matrix of size $K \times K$. $\sigma_{\max}(\boldsymbol{X})$ and $\sigma_{\min}(\boldsymbol{X})$ denote the largest and the smallest singular values of the matrix $\boldsymbol{X}$, respectively. $\mathsf{cone}(\boldsymbol{X})$ denotes the conic hull formed by the columns of the matrix $\boldsymbol{X}$. $\det(\boldsymbol{X})$ represents the determinant of $\boldsymbol{X}$. $\mathsf{vec}(\boldsymbol{X})$ denotes the vectorization operation that concatenates the columns of $\boldsymbol{X}$. $\mathsf{trace}(\boldsymbol{X})$ outputs the trace (the sum of the diagonal elements) of $\boldsymbol{X}$.

## B   PROOF OF THEOREM 1

If the vectors $\boldsymbol{p}_n^{(m)}$ for all $m, n$ are available, then one can represent the model in (2) as a low-rank *nonnegative matrix factorization* (NMF) model as follows:

$$\underbrace{\begin{bmatrix} \boldsymbol{p}_1^{(1)} & \cdots & \boldsymbol{p}_N^{(1)} \\ \vdots & \ddots & \vdots \\ \boldsymbol{p}_1^{(M)} & \cdots & \boldsymbol{p}_N^{(M)} \end{bmatrix}}_{\boldsymbol{P} \in \mathbb{R}^{MK \times N}} = \underbrace{\begin{bmatrix} \boldsymbol{A}_1 \\ \vdots \\ \boldsymbol{A}_M \end{bmatrix}}_{\boldsymbol{W} \in \mathbb{R}^{MK \times K}} \underbrace{[\boldsymbol{f}(\boldsymbol{x}_1) \quad \ldots \quad \boldsymbol{f}(\boldsymbol{x}_N)]}_{\boldsymbol{F} \in \mathbb{R}^{K \times N}}, \tag{10}$$

where the factors $\boldsymbol{W}$ and $\boldsymbol{F}$ are both nonnegative per their physical meaning.

Let $\boldsymbol{P}^{\natural}$ denote the ground-truth and $\boldsymbol{p}_n^{\natural(m)} \in \mathbb{R}^K$ the $(m, n)$th block in $\boldsymbol{P}^{\natural}$, following the representation in (10). Note that we do not observe the entire $\boldsymbol{P}^{\natural}$ but only the $\widehat{y}_n^{(m)}$ sampled from $\boldsymbol{p}_n^{\natural(m)}$ for $(m, n) \in \mathcal{S}$. We first show that the CCEM criterion in (6) *implicitly* estimates $\boldsymbol{P}^{\natural}$ from incomplete labels indexed by $\mathcal{S}$. To this end, we consider the objective function in (6a):

$$D_{\mathcal{S}}(\boldsymbol{P}; \widehat{\mathcal{Y}}) \triangleq -\frac{1}{S} \sum_{(m,n) \in \mathcal{S}} \sum_{k=1}^{K} \mathbb{I}[\widehat{y}_n^{(m)} = k] \log \boldsymbol{P}((m-1)K + k, n), \tag{11}$$

where $\widehat{\mathcal{Y}}$ denotes the set of observed noisy labels, i.e., $\{\widehat{y}_n^{(m)}\}_{(m,n) \in \mathcal{S}}$. Let $\{\widehat{\boldsymbol{A}}_m\}_{m=1}^M$ and $\widehat{\boldsymbol{f}}$ denote the estimates given by the learning criterion in (6). Then, the criterion (6) helps define the following term:

$$\widehat{\boldsymbol{P}} \triangleq \underset{\boldsymbol{P} \in \mathcal{P}}{\arg \min} \, D_{\mathcal{S}}(\boldsymbol{P}; \widehat{\mathcal{Y}}), \tag{12}$$

where $\widehat{\boldsymbol{P}} = \widehat{\boldsymbol{W}} \begin{bmatrix} \widehat{\boldsymbol{f}}(\boldsymbol{x}_1) & \ldots & \widehat{\boldsymbol{f}}(\boldsymbol{x}_N) \end{bmatrix}$, $\widehat{\boldsymbol{W}} = [\widehat{\boldsymbol{A}}_1^{\top}, \ldots, \widehat{\boldsymbol{A}}_M^{\top}]^{\top}$,

$$\mathcal{P} \triangleq \{\boldsymbol{P} \in \mathbb{R}^{MK \times N} \mid \boldsymbol{P} = \boldsymbol{W} \begin{bmatrix} \boldsymbol{f}(\boldsymbol{x}_1) & \ldots & \boldsymbol{f}(\boldsymbol{x}_N) \end{bmatrix}, \boldsymbol{W} \in \mathcal{W}, \, \boldsymbol{f} \in \mathcal{F}\},$$

$$\mathcal{W} \triangleq \{\boldsymbol{W} = [\boldsymbol{A}_1^{\top}, \ldots, \boldsymbol{A}_M^{\top}]^{\top} \in \mathbb{R}^{MK \times K} \mid \mathbf{1}^{\top} \boldsymbol{A}_m = \mathbf{1}^{\top}, \boldsymbol{A}_m \geq \boldsymbol{0}, \, \forall m\}, \text{and}$$

$$\mathcal{F} \subset \{\boldsymbol{f}(\boldsymbol{x}) \in \mathbb{R}^K \mid \boldsymbol{f}(\boldsymbol{x}) \in \boldsymbol{\Delta}_K, \, \forall \boldsymbol{x} \in \mathbb{R}^D\} \text{ is the neural network function class.}$$

Our main goal is to characterize the estimation errors of $\{\widehat{\boldsymbol{A}}_m\}_{m=1}^M$ and $\widehat{\boldsymbol{f}}$. This can be done via first bounding the estimation errors $\|\boldsymbol{p}_n^{\natural(m)} - \widehat{\boldsymbol{p}}_n^{(m)}\|_2^2$, where $\boldsymbol{p}_n^{\natural(m)}$ is given by:

$$\boldsymbol{p}_n^{\natural(m)} = \boldsymbol{A}_m^{\natural} \boldsymbol{f}^{\natural}(\boldsymbol{x}_n), \, \forall m, n,$$

in which $\boldsymbol{A}_m^{\natural}$ and $\boldsymbol{f}^{\natural}$ denote the $m$th ground-truth confusion matrix and the GTP, respectively, and $\widehat{\boldsymbol{p}}_n^{(m)}$ denote the $(m, n)$th block in $\widehat{\boldsymbol{P}}$.

## B.1 ESTIMATING $\|\boldsymbol{p}_n^{\natural(m)} - \widehat{\boldsymbol{p}}_n^{(m)}\|_2^2$

We first show the following proposition:

**Proposition 1** *Under the assumptions in Theorem 1, the following result hold with probability at least $1 - \delta$:*

$$\frac{1}{NM} \sum_{n=1}^{N} \sum_{m=1}^{M} \|\boldsymbol{p}_n^{\natural(m)} - \widehat{\boldsymbol{p}}_n^{(m)}\|_2^2 \leq 4\sqrt{2}K\beta\mathfrak{R}_{\mathcal{S}}(\mathcal{P}) + 10\log(\beta)\frac{\sqrt{2\log(4/\delta)}}{\sqrt{S}} + 2\beta\sqrt{K}\nu, \quad (13)$$

*where $\mathfrak{R}_{\mathcal{S}}(\mathcal{P}) = \frac{16}{\sqrt{S}}\sqrt{MK\log\left(4S\sqrt{K}\right) + (2\|\boldsymbol{X}\|_{\mathrm{F}}\mathscr{R}_{\mathcal{F}})^{\frac{1}{2}}}$ and is related to the empirical Rademacher complexity of the set $\mathcal{P}$ under the observed samples $\mathcal{S}$.*

The proof is provided in Sec. C. Proposition 1 provides an upper-bound for $\|\boldsymbol{P}^{\natural} - \widehat{\boldsymbol{P}}\|_{\mathrm{F}}^2$. To achieve the main goal, i.e., characterizing the estimation errors of $\{\widehat{\boldsymbol{A}}_m\}_{m=1}^{M}$ and $\widehat{\boldsymbol{f}}$, we require an upper-bound for $\sum_{m=1}^{M} \|\boldsymbol{p}_n^{\natural(m)} - \widehat{\boldsymbol{p}}_n^{(m)}\|_2^2$ as well. The following result gives a tighter upper bound for $\sum_{m=1}^{M} \|\boldsymbol{p}_n^{\natural(m)} - \widehat{\boldsymbol{p}}_n^{(m)}\|_2^2$:

**Proposition 2** *Under the assumptions in Theorem 1, for any $\alpha > 0$ and $\delta \in (0, 1)$, the following result holds with probability at least $1 - \frac{1}{N^\alpha}$:*

$$\frac{1}{M} \sum_{m=1}^{M} \|\boldsymbol{p}_n^{\natural(m)} - \widehat{\boldsymbol{p}}_n^{(m)}\|_2^2$$

$$\leq N^\alpha \left( 4\sqrt{2}K\beta\mathfrak{R}_{\mathcal{S}}(\mathcal{P}) + 10\log(\beta)\frac{\sqrt{2\log(4/\delta)}}{\sqrt{S}} + 2\beta\sqrt{K}\nu + 4\delta \right), \quad \forall n, \quad (14)$$

*where $\mathfrak{R}_{\mathcal{S}}(\mathcal{P}) = \frac{16}{\sqrt{S}}\sqrt{MK\log\left(4S\sqrt{K}\right) + (2\|\boldsymbol{X}\|_{\mathrm{F}}\mathscr{R}_{\mathcal{F}})^{\frac{1}{2}}}$.*

The proof is provided in Sec. I. Note that the bound in (14) looks divergent at the first glance, as the second term in the R.H.S. could go to infinity. However, given a large enough $S$, one can choose appropriate $\alpha$ and $\delta$ such that the bound in (14) does not explode.

## B.2 ESTIMATION ERRORS FOR $\widehat{\boldsymbol{A}}_m$ AND $\widehat{\boldsymbol{f}}$

In this section, we characterize the estimation error of the confusion matrices and the GTP. We start with using simplified notations to represent the results in Propositions 1-2, i.e.,

$$\|\boldsymbol{P}^{\natural} - \widehat{\boldsymbol{P}}\|_{\mathrm{F}}^2 \leq \zeta^2, \quad (15)$$

$$\sum_{m=1}^{M} \|\boldsymbol{p}_n^{\natural(m)} - \widehat{\boldsymbol{p}}_n^{(m)}\|_2^2 \leq \varphi^2, \forall n. \quad (16)$$

with probability greater than $1 - \delta$ and $1 - \frac{1}{N^\alpha}$, respectively.

Assumptions 4-5 imply that there exists index sets $\Lambda = \{\tilde{m}_1, \dots, \tilde{m}_K\}$ and $\Psi = \{\tilde{n}_1, \dots, \tilde{n}_K\}$ such that:

$$\boldsymbol{W}^{\natural}(\Lambda, :) = \boldsymbol{I}_K + \boldsymbol{N}_W$$
$$\boldsymbol{F}^{\natural}(:, \Psi) = \boldsymbol{I}_K + \boldsymbol{N}_F,$$

where $\|\boldsymbol{N}_W\|_{\mathrm{F}} \leq \sqrt{K}\xi_1$, and $\|\boldsymbol{N}_F\|_{\mathrm{F}} \leq \sqrt{K}\xi_2$. Note that the conditions do not require any confusion matrices to be near identity; i.e., we do not need any annotators to be all-class specialists.

Nonetheless, for notation simplicity, we let $\Lambda = \{1, \ldots, K\}$ and $\Psi = \{1, \ldots, K\}$, which is without loss of generality for our proof in the sequel. Under this simplification, the following holds:

$$
\boldsymbol{W}^{\natural} = \begin{bmatrix} \boldsymbol{A}_1^{\natural} \\ \boldsymbol{A}_2^{\natural} \\ \vdots \\ \boldsymbol{A}_M^{\natural} \end{bmatrix}, \ \boldsymbol{F}^{\natural} = \begin{bmatrix} \boldsymbol{f}^{\natural}(\boldsymbol{x}_1) & \ldots & \boldsymbol{f}^{\natural}(\boldsymbol{x}_N) \end{bmatrix} = \begin{bmatrix} \boldsymbol{F}_1^{\natural} & \boldsymbol{F}_2^{\natural} \end{bmatrix},
$$

where $\boldsymbol{A}_1^{\natural} = \boldsymbol{I}_K + \boldsymbol{N}_W$, $\boldsymbol{F}_1^{\natural} = \boldsymbol{I}_K + \boldsymbol{N}_F$, and $\boldsymbol{F}_2^{\natural} \in \mathbb{R}^{K \times (N-K)}$, $\|\boldsymbol{N}_W\|_{\mathrm{F}} \le \sqrt{K}\xi_1$, and $\|\boldsymbol{N}_F\|_{\mathrm{F}} \le \sqrt{K}\xi_2$.

Let $\widehat{\boldsymbol{A}}_m$ and $\widehat{\boldsymbol{f}}$ denote the estimates of $\boldsymbol{A}_m^{\natural}$ and $\boldsymbol{f}^{\natural}$, respectively, using the learning criterion (6) and construct

$$
\widehat{\boldsymbol{W}} = \begin{bmatrix} \widehat{\boldsymbol{A}}_1 \\ \widehat{\boldsymbol{A}}_2 \\ \vdots \\ \widehat{\boldsymbol{A}}_M \end{bmatrix}, \ \widehat{\boldsymbol{F}} = \begin{bmatrix} \widehat{\boldsymbol{f}}(\boldsymbol{x}_1) & \ldots & \widehat{\boldsymbol{f}}(\boldsymbol{x}_N) \end{bmatrix} = \begin{bmatrix} \widehat{\boldsymbol{F}}_1 & \widehat{\boldsymbol{F}}_2 \end{bmatrix},
$$

where $\widehat{\boldsymbol{A}}_m \in \mathbb{R}^{K \times K}$, $\widehat{\boldsymbol{F}}_1 \in \mathbb{R}^{K \times K}$, and $\widehat{\boldsymbol{F}}_2 \in \mathbb{R}^{K \times (N-K)}$. Then, we have:

$$
\begin{aligned}
\|\boldsymbol{P}^{\natural} - \widehat{\boldsymbol{P}}\|_{\mathrm{F}}^2 &= \|\boldsymbol{W}^{\natural}\boldsymbol{F}^{\natural} - \widehat{\boldsymbol{W}}\widehat{\boldsymbol{F}}\|_{\mathrm{F}}^2 \\
&= \|\boldsymbol{I}_K + \boldsymbol{N}_W + \boldsymbol{N}_F + \boldsymbol{N}_W\boldsymbol{N}_F - \widehat{\boldsymbol{A}}_1\widehat{\boldsymbol{F}}_1\|_{\mathrm{F}}^2 + \|\boldsymbol{F}_2^{\natural} + \boldsymbol{N}_W\boldsymbol{F}_2^{\natural} - \widehat{\boldsymbol{A}}_1\widehat{\boldsymbol{F}}_2\|_{\mathrm{F}}^2 \\
&\quad + \sum_{m=2}^{M} \|\boldsymbol{A}_m^{\natural} + \boldsymbol{A}_m^{\natural}\boldsymbol{N}_F - \widehat{\boldsymbol{A}}_m\widehat{\boldsymbol{F}}_1\|_{\mathrm{F}}^2 + \sum_{m=2}^{M} \|\boldsymbol{A}_m^{\natural}\boldsymbol{F}_2^{\natural} - \widehat{\boldsymbol{A}}_m\widehat{\boldsymbol{F}}_2\|_{\mathrm{F}}^2. \quad (17)
\end{aligned}
$$

Let us define the error matrices as below:

$$
\boldsymbol{E}_A^{(1)} \triangleq \widehat{\boldsymbol{A}}_1 - \underbrace{(\boldsymbol{I}_K + \boldsymbol{N}_W)\boldsymbol{\Pi}}_{\boldsymbol{A}_1^{\natural}\boldsymbol{\Pi}}, \ \boldsymbol{E}_A^{(m)} \triangleq \widehat{\boldsymbol{A}}_m - \boldsymbol{A}_m^{\natural}\boldsymbol{\Pi}, \ \forall m > 1, \quad (18)
$$

$$
\boldsymbol{E}_F^{(1)} \triangleq \widehat{\boldsymbol{F}}_1 - \underbrace{\boldsymbol{\Pi}^{\top}(\boldsymbol{I}_K + \boldsymbol{N}_F)}_{\boldsymbol{\Pi}^{\top}\boldsymbol{F}_1^{\natural}}, \ \boldsymbol{E}_F^{(2)} \triangleq \widehat{\boldsymbol{F}}_2 - \boldsymbol{\Pi}^{\top}\boldsymbol{F}_2^{\natural}, \quad (19)
$$

where $\boldsymbol{\Pi} \in [0,1]^{K \times K}$ is a column permutation matrix and is the same across all the error blocks. We hope to characterize the norm of these error matrices.

**Upper bound for $\|\boldsymbol{E}_A^{(1)}\|_{\mathrm{F}}$ and $\|\boldsymbol{E}_F^{(1)}\|_{\mathrm{F}}$.** We start by considering the first term on the R.H.S. of (17), i.e., $\|\boldsymbol{I}_K + \boldsymbol{N}_W + \boldsymbol{N}_F + \boldsymbol{N}_W\boldsymbol{N}_F - \widehat{\boldsymbol{A}}_1\widehat{\boldsymbol{F}}_1\|_{\mathrm{F}}^2$. From (16), we have the following with probability greater than $1 - \frac{K}{N^{\alpha}}$:

$$
\begin{aligned}
\varphi^2 &\ge \|\boldsymbol{I}_K + \boldsymbol{N}_W + \boldsymbol{N}_F + \boldsymbol{N}_W\boldsymbol{N}_F - \widehat{\boldsymbol{A}}_1\widehat{\boldsymbol{F}}_1\|_{\mathrm{F}}^2 \\
&= \sum_{k=1}^{K} |1 + \widetilde{\boldsymbol{N}}(k,k) - \widehat{\boldsymbol{A}}_1(k,:)\widehat{\boldsymbol{F}}_1(:,k)|^2 + \sum_{j=1}^{K}\sum_{k \ne j} |\widetilde{\boldsymbol{N}}(j,k) - \widehat{\boldsymbol{A}}_1(j,:)\widehat{\boldsymbol{F}}_1(:,k)|^2, \quad (20)
\end{aligned}
$$

where $\widetilde{\boldsymbol{N}} = \boldsymbol{N}_W + \boldsymbol{N}_F + \boldsymbol{N}_W\boldsymbol{N}_F$. The absolute value of the largest entry in $\widetilde{\boldsymbol{N}}$ can be bounded by $\xi_1 + \xi_2 + \sqrt{K}\xi_1\xi_2$. Let us denote $\kappa = \varphi + \xi_1 + \xi_2 + \sqrt{K}\xi_1\xi_2$ for conciseness. Then, from (20), we get the following conditions:

$$
1 - \kappa \le |\widehat{\boldsymbol{A}}_1(k,:)\widehat{\boldsymbol{F}}_1(:,k)| \le 1 + \kappa, \ \forall k \in [K] \quad (21a)
$$

$$
|\widehat{\boldsymbol{A}}_1(k,:)\widehat{\boldsymbol{F}}_1(:,j)| \le \kappa, \forall k \ne j. \quad (21b)
$$

From the conditions (21a) and (21b), we have the following result:

**Lemma 1** *Assume that the conditions in* (21a) *and* (21b) *hold and that* $\kappa \leq \frac{1}{K+1}$. *Then, we have the following relations satisfied:*

$$arg\ max_\ell \widehat{\boldsymbol{A}}_1(k,\ell) \neq arg\ max_\ell \widehat{\boldsymbol{A}}_1(j,\ell),\ \forall k \neq j \tag{22a}$$

$$arg\ max_q \widehat{\boldsymbol{F}}_1(q,k) \neq arg\ max_q \widehat{\boldsymbol{F}}_1(q,j),\ \forall k \neq j \tag{22b}$$

$$arg\ max_\ell \widehat{\boldsymbol{A}}_1(k,\ell) = arg\ max_q \widehat{\boldsymbol{F}}_1(q,k),\ \forall k. \tag{22c}$$

The proof of the lemma is given in Sec. F.

From the lower bound condition in (21a), we have the following result:

$$\sum_{k=1}^{K}(1-\kappa) \leq \sum_{k=1}^{K} \widehat{\boldsymbol{A}}_1(k,1)\widehat{\boldsymbol{F}}_1(1,k) + \cdots + \widehat{\boldsymbol{A}}_1(k,K)\widehat{\boldsymbol{F}}_1(K,k)$$

$$\leq \sum_{k=1}^{K} \max_{\ell \in [K]} \widehat{\boldsymbol{A}}_1(k,\ell)(\widehat{\boldsymbol{F}}_1(1,k) + \cdots + \widehat{\boldsymbol{F}}_1(K,k))$$

$$= \sum_{k=1}^{K} \max_{\ell \in [K]} \widehat{\boldsymbol{A}}_1(k,\ell) \tag{23}$$

where the last equality is obtained due to the probability simplex constraints on the columns of $\widehat{\boldsymbol{F}}_1$.

From the lower bound condition in (21a), we have the below result as well:

$$\sum_{k=1}^{K}(1-\kappa) \leq \sum_{k=1}^{K} \left( \widehat{\boldsymbol{A}}_1(k,1)\widehat{\boldsymbol{F}}_1(1,k) + \cdots + \widehat{\boldsymbol{A}}_1(k,K)\widehat{\boldsymbol{F}}_1(K,k) \right)$$

$$\leq \sum_{k=1}^{K} \max_\ell \widehat{\boldsymbol{F}}_1(k,\ell)(\widehat{\boldsymbol{A}}_1(1,k) + \cdots + \widehat{\boldsymbol{A}}_1(K,k))$$

$$= \sum_{k=1}^{K} \max_\ell \widehat{\boldsymbol{F}}_1(k,\ell). \tag{24}$$

where the last equality is obtained due to the probability simplex constraints on all the columns of $\widehat{\boldsymbol{A}}_1$.

Next, we characterize the term $\|\boldsymbol{E}_A^{(1)}\|_{\mathrm{F}}^2$. To achieve this, by employing the result (22a) in Lemma 1, we fix the column permutation $\boldsymbol{\Pi}$ as follows:

$$\boldsymbol{\Pi}(j,k) = \begin{cases} 1, & j = arg\ max_\ell \widehat{\boldsymbol{A}}_1(k,\ell) \\ 0, & \text{otherwise.} \end{cases} \tag{25}$$

We get the following set of relations:

$$
\begin{aligned}
\|\widehat{\boldsymbol{A}}_1 - \boldsymbol{I}_K \boldsymbol{\Pi}\|_{\mathrm{F}}^2 &= \sum_{k=1}^{K} \|\widehat{\boldsymbol{A}}_1(k,:) - \boldsymbol{e}_k^\top \boldsymbol{\Pi}\|_2^2 \\
&= \sum_{k=1}^{K} \|\widehat{\boldsymbol{A}}_1(k,:)\|_2^2 + K - 2 \sum_{k=1}^{K} \widehat{\boldsymbol{A}}_1(k,:) \boldsymbol{\Pi}^\top \boldsymbol{e}_k \\
&= \|\widehat{\boldsymbol{A}}_1\|_{\mathrm{F}}^2 + K - 2 \sum_{k=1}^{K} \widehat{\boldsymbol{A}}_1(k,:) \boldsymbol{\Pi}^\top \boldsymbol{e}_k \\
&\leq 2K - 2 \sum_{k=1}^{K} \widehat{\boldsymbol{A}}_1(k,:) \boldsymbol{\Pi}^\top \boldsymbol{e}_k \\
&\overset{(a)}{=} 2K - 2 \sum_{k=1}^{K} \max_\ell \widehat{\boldsymbol{A}}_1(k,\ell) \\
&\overset{(b)}{\leq} 2K - 2 \sum_{k=1}^{K} (1 - \kappa) = 2K\kappa,
\end{aligned}
$$

where the last inequality $(b)$ is by (23) and the relation $(a)$ is obtained by choosing $\boldsymbol{\Pi}$ as defined in (25).

Hence, we have the following with probability greater than $1 - \frac{K}{N^\alpha}$:

$$
\|\boldsymbol{E}_A^{(1)}\|_{\mathrm{F}} \leq \|\widehat{\boldsymbol{A}}_1 - \boldsymbol{I}_K \boldsymbol{\Pi}\|_{\mathrm{F}} + \|\boldsymbol{N}_W\|_{\mathrm{F}} \leq \sqrt{2K\kappa} + \sqrt{K}\xi_1. \tag{26}
$$

We proceed to consider the error term $\|\boldsymbol{E}_F^{(1)}\|_{\mathrm{F}}^2$. To achieve this, consider the following:

$$
\begin{aligned}
\|\widehat{\boldsymbol{F}}_1 - \boldsymbol{\Pi}^\top \boldsymbol{I}_K\|_{\mathrm{F}}^2 &= \sum_{k=1}^{K} \|\widehat{\boldsymbol{F}}_1(k,:) - \boldsymbol{e}_k^\top \boldsymbol{\Pi}^\top\|_2^2 \\
&= \sum_{k=1}^{K} \|\widehat{\boldsymbol{F}}_1(k,:)\|_2^2 + K - 2 \sum_{k=1}^{K} \widehat{\boldsymbol{F}}_1(k,:) \boldsymbol{\Pi}^\top \boldsymbol{e}_k \\
&= \|\widehat{\boldsymbol{F}}_1\|_{\mathrm{F}}^2 + K - 2 \sum_{k=1}^{K} \widehat{\boldsymbol{F}}_1(k,:) \boldsymbol{\Pi}^\top \boldsymbol{e}_k \\
&\leq 2K - 2 \sum_{k=1}^{K} \widehat{\boldsymbol{F}}_1(k,:) \boldsymbol{\Pi}^\top \boldsymbol{e}_k \\
&\overset{(a)}{=} 2K - 2 \sum_{k=1}^{K} \max_\ell \widehat{\boldsymbol{F}}_1(k,\ell) \\
&\overset{(b)}{\leq} 2K - 2 \sum_{k=1}^{K} (1 - \kappa) = 2K\kappa,
\end{aligned}
$$

where the last inequality $(b)$ is by (24) and the relation $(a)$ is by combining the definition of $\boldsymbol{\Pi}$ in (25) and (22c) in Lemma 1. Hence, we get the following with probability greater than $1 - \frac{K}{N^\alpha}$:

$$
\|\boldsymbol{E}_F^{(1)}\|_{\mathrm{F}} \leq \|\widehat{\boldsymbol{F}}_1 - \boldsymbol{I}_K \boldsymbol{\Pi}\|_{\mathrm{F}} + \|\boldsymbol{N}_F\|_{\mathrm{F}} \leq \sqrt{2K\kappa} + \sqrt{K}\xi_2. \tag{27}
$$

**Upper bound for $\|\boldsymbol{E}_F^{(2)}\|_{\mathrm{F}}$.** Next, we consider the second term in (17), i.e., $\|\boldsymbol{F}_2^\natural + \boldsymbol{N}_W \boldsymbol{F}_2^\natural - \widehat{\boldsymbol{A}}_1 \widehat{\boldsymbol{F}}_2\|_{\mathrm{F}}$. From (15), we have the following with probability greater than $1 - \delta$:

$$
\begin{aligned}
\zeta \geq \|\boldsymbol{F}_2^\natural + \boldsymbol{N}_W \boldsymbol{F}_2^\natural - \widehat{\boldsymbol{A}}_1 \widehat{\boldsymbol{F}}_2\|_{\mathrm{F}} &= \|\boldsymbol{F}_2^\natural + \boldsymbol{N}_W \boldsymbol{F}_2^\natural - (\boldsymbol{I}_K \boldsymbol{\Pi} + \boldsymbol{N}_W \boldsymbol{\Pi} + \boldsymbol{E}_A^{(1)})(\boldsymbol{E}_F^{(2)} + \boldsymbol{\Pi}^\top \boldsymbol{F}_2^\natural)\|_{\mathrm{F}} \\
&= \|(\boldsymbol{I}_K \boldsymbol{\Pi} + \boldsymbol{N}_W \boldsymbol{\Pi} + \boldsymbol{E}_A^{(1)}) \boldsymbol{E}_F^{(2)} + \boldsymbol{E}_A^{(1)} \boldsymbol{\Pi}^\top \boldsymbol{F}_2^\natural\|_{\mathrm{F}} \\
&= \|\widehat{\boldsymbol{A}}_1 \boldsymbol{E}_F^{(2)} + \boldsymbol{E}_A^{(1)} \boldsymbol{\Pi}^\top \boldsymbol{F}_2^\natural\|_{\mathrm{F}} \\
&\overset{(a)}{\geq} \left| \|\widehat{\boldsymbol{A}}_1 \boldsymbol{E}_F^{(2)}\|_{\mathrm{F}} - \|\boldsymbol{E}_A^{(1)} \boldsymbol{\Pi}^\top \boldsymbol{F}_2^\natural\|_{\mathrm{F}} \right| \\
&\overset{(b)}{\geq} \left| \sigma_{\min}(\widehat{\boldsymbol{A}}_1) \|\boldsymbol{E}_F^{(2)}\|_{\mathrm{F}} - \|\boldsymbol{E}_A^{(1)}\|_{\mathrm{F}} \|\boldsymbol{F}_2^\natural\|_{\mathrm{F}} \right| \\
&\overset{(c)}{\geq} \left| \sigma_{\min}(\widehat{\boldsymbol{A}}_1) \|\boldsymbol{E}_F^{(2)}\|_{\mathrm{F}} - (\sqrt{2K\kappa} + \sqrt{K}\xi_1) \|\boldsymbol{F}_2^\natural\|_{\mathrm{F}} \right| \quad (28)
\end{aligned}
$$

where the inequality $(a)$ is by using the triangle inequality, $(b)$ is obtained by applying the following two relations for any two matrices $\boldsymbol{A} \in \mathbb{R}^{K \times K}, \boldsymbol{B} \in \mathbb{R}^{K \times L},\ L \geq K$:

$$
\|\boldsymbol{A}\boldsymbol{B}\|_{\mathrm{F}} \geq \sigma_{\min}(\boldsymbol{A}) \|\boldsymbol{B}\|_{\mathrm{F}} \quad (29)
$$
$$
\|\boldsymbol{A}\boldsymbol{B}\|_{\mathrm{F}} \leq \|\boldsymbol{A}\|_{\mathrm{F}} \|\boldsymbol{B}\|_{\mathrm{F}}. \quad (30)
$$

The inequality $(c)$ is obtained by applying (26).

Hence, (28) combined with the fact that $\|\boldsymbol{F}_2\|_{\mathrm{F}} \leq \|\boldsymbol{F}^\natural\|_{\mathrm{F}}$ gives the following relation with probability greater than $1 - \delta - \frac{K}{N^\alpha}$:

$$
\|\boldsymbol{E}_F^{(2)}\|_{\mathrm{F}} \leq \frac{\zeta + (\sqrt{2K\kappa} + \sqrt{K}\xi_1) \|\boldsymbol{F}^\natural\|_{\mathrm{F}}}{\sigma_{\min}(\widehat{\boldsymbol{A}}_1)}. \quad (31)
$$

Next, we use the following lemma to characterize $\sigma_{\min}(\widehat{\boldsymbol{A}}_1)$:

**Lemma 2** *Suppose that the matrix $\boldsymbol{X} \in \mathbb{R}^{K \times K}$ takes the form $\boldsymbol{X} = \boldsymbol{I}_K + \boldsymbol{E}_1 + \boldsymbol{E}_2$. Assume that $\|\boldsymbol{E}_1\|_F \leq \upsilon_1$ and $\|\boldsymbol{E}_2\|_F \leq \upsilon_2$, for a certain $\upsilon_1, \upsilon_2 > 0$. Then, we have*

$$
\sigma_{\min}(\boldsymbol{X}) \geq |1 - \upsilon_1 - \upsilon_2|.
$$

The proof is relegated to Sec. G. Applying Lemma 2, we get the following with probability greater than $1 - \delta - \frac{K}{N^\alpha}$:

$$
\|\boldsymbol{E}_F^{(2)}\|_{\mathrm{F}} \leq \frac{\zeta + (\sqrt{2K\kappa} + \sqrt{K}\xi_1) \|\boldsymbol{F}^\natural\|_{\mathrm{F}}}{|1 - \sqrt{2K\kappa} - 2\sqrt{K}\xi_1|} \leq \frac{c_1(\zeta + \sqrt{K\kappa} \|\boldsymbol{F}^\natural\|_{\mathrm{F}})}{|1 - \sqrt{K\kappa}|} \quad (32)
$$

for certain constant $c_1 > 0$. where we used the fact that $\xi_1 \leq \sqrt{\kappa}$ since $\kappa = \varphi + \xi_1 + \xi_2 + \sqrt{K}\xi_1\xi_2$ and and $\kappa \leq 1$.

**Upper bound for $\|\boldsymbol{E}_A^{(m)}\|_{\mathrm{F}},\ m > 1$.** We consider the third term on the R.H.S. of (17). From (16), for each $m$, we have the following with probability greater than $1 - \frac{K}{N^\alpha}$:

$$
\begin{aligned}
\sqrt{K}\varphi \geq \|\boldsymbol{A}_m^\natural + \boldsymbol{A}_m^\natural \boldsymbol{N}_F - \widehat{\boldsymbol{A}}_m \widehat{\boldsymbol{F}}_1\|_{\mathrm{F}} &= \|\boldsymbol{A}_m^\natural + \boldsymbol{A}_m^\natural \boldsymbol{N}_F - (\boldsymbol{A}_m^\natural \boldsymbol{\Pi} + \boldsymbol{E}_A^{(m)})(\boldsymbol{\Pi}^\top \boldsymbol{I}_K + \boldsymbol{\Pi}^\top \boldsymbol{N}_F + \boldsymbol{E}_F^{(1)})\|_{\mathrm{F}} \\
&= \|\boldsymbol{E}_A^{(m)}(\boldsymbol{\Pi}^\top \boldsymbol{I}_K + \boldsymbol{\Pi}^\top \boldsymbol{N}_F + \boldsymbol{E}_F^{(1)}) + \boldsymbol{A}_m^\natural \boldsymbol{\Pi} \boldsymbol{E}_F^{(1)}\|_{\mathrm{F}} \\
&= \|\boldsymbol{E}_A^{(m)} \widehat{\boldsymbol{F}}_1 + \boldsymbol{A}_m^\natural \boldsymbol{\Pi} \boldsymbol{E}_F^{(1)}\|_{\mathrm{F}} \\
&\overset{(a)}{\geq} \left| \|\boldsymbol{E}_A^{(m)} \widehat{\boldsymbol{F}}_1\|_{\mathrm{F}} - \|\boldsymbol{A}_m^\natural \boldsymbol{\Pi} \boldsymbol{E}_F^{(1)}\|_{\mathrm{F}} \right| \\
&\overset{(b)}{\geq} \left| \sigma_{\min}(\widehat{\boldsymbol{F}}_1) \|\boldsymbol{E}_A^{(m)}\|_{\mathrm{F}} - \sigma_{\max}(\boldsymbol{A}_m^\natural) \|\boldsymbol{E}_F^{(1)}\|_{\mathrm{F}} \right| \\
&\overset{(c)}{\geq} \left| \sigma_{\min}(\widehat{\boldsymbol{F}}_1) \|\boldsymbol{E}_A^{(m)}\|_{\mathrm{F}} - (\sqrt{2K\kappa} + \sqrt{K}\xi_2) \sigma_{\max}(\boldsymbol{A}_m^\natural) \right|,
\end{aligned}
$$

where the relation $(a)$ by the triangle inequality, $(b)$ is by applying (29) and (30), and $(c)$ is via (27).

Hence, $\forall m > 1$, we get the following with probability greater than $1 - \frac{2K}{N^\alpha}$:

$$\|\boldsymbol{E}_A^{(m)}\|_{\mathrm{F}} \leq \frac{\sqrt{K}\varphi + (\sqrt{2K\kappa} + \sqrt{K}\xi_2)\sigma_{\max}(\boldsymbol{A}_m^\natural)}{|1 - \sqrt{2K\kappa} - 2\sqrt{K}\xi_2|} \leq \frac{c_2\sqrt{K\kappa}\sigma_{\max}(\boldsymbol{A}_m^\natural)}{|1 - \sqrt{K\kappa}|} \tag{33}$$

for certain constant $c_2 > 0$, where we used the fact that $\varphi \leq \sqrt{\kappa}$ and $\xi_2 \leq \sqrt{\kappa}$ since $\kappa = \varphi + \xi_1 + \xi_2 + \sqrt{K}\xi_1\xi_2$ and $\kappa \leq 1$. In the above, we have also applied $\sigma_{\min}(\widehat{\boldsymbol{F}}_1) \geq |1 - \sqrt{2K\kappa} - 2\sqrt{K}\xi_2|$ following Lemma 2 and (27).

**Putting Together.** From (33), we have the following with probability greater than $1 - \frac{2K}{N^\alpha}$:

$$\|\boldsymbol{E}_A^{(m)}\|_{\mathrm{F}}^2 = \|\widehat{\boldsymbol{A}}_m - \boldsymbol{A}_m^\natural \boldsymbol{\Pi}\|_{\mathrm{F}}^2 \leq \frac{c_2^2 K^2 \kappa}{|1 - \sqrt{K\kappa}|^2}, \quad \forall m. \tag{34}$$

where we have used the fact that $\sigma_{\max}(\boldsymbol{A}_m^\natural) \leq \|\boldsymbol{A}_m^\natural\|_{\mathrm{F}} \leq \sqrt{K}$.

Similarly, by combining (27) and (32), we have the following with probability greater than $1 - \delta - \frac{2K}{N^\alpha}$

$$\begin{aligned}
\|\widehat{\boldsymbol{F}} - \boldsymbol{\Pi}^\top \boldsymbol{F}^\natural\|_{\mathrm{F}}^2 &= \|\boldsymbol{E}_F^{(1)}\|_{\mathrm{F}}^2 + \|\boldsymbol{E}_F^{(2)}\|_{\mathrm{F}}^2 \\
&\leq (\sqrt{2K\kappa} + \sqrt{K}\xi_2)^2 + \frac{c_1^2(\zeta + \sqrt{K\kappa}\|\boldsymbol{F}^\natural\|_{\mathrm{F}})^2}{|1 - \sqrt{K\kappa}|^2} \\
&\leq \frac{c_3(\zeta + \sqrt{NK\kappa})^2}{|1 - \sqrt{K\kappa}|^2}
\end{aligned} \tag{35}$$

for certain constant $c_3 > 0$, where we have used the fact that $\|\boldsymbol{F}^\natural\|_{\mathrm{F}} \leq \sqrt{N}$.

We hope to characterize the generalization performance of the predicted function $\widehat{\boldsymbol{f}}$ from (35). Towards this, we have the following result:

**Lemma 3** *Under Assumptions 1 and 2, the following holds with probability greater than $1 - \delta$:*

$$\mathbb{E}_{\boldsymbol{x}\sim\mathcal{D}}\left[\|\widehat{\boldsymbol{f}}(\boldsymbol{x}) - \boldsymbol{\Pi}^\top \boldsymbol{f}^\natural(\boldsymbol{x})\|_2^2\right] \leq \frac{1}{N}\|\widehat{\boldsymbol{F}} - \boldsymbol{\Pi}^\top \boldsymbol{F}^\natural\|_{\mathrm{F}}^2 + 64N^{-5/8}\left(2\|\boldsymbol{X}\|_{\mathrm{F}}\mathscr{R}_\mathcal{F}\right)^{\frac{1}{4}}$$
$$+ 16\sqrt{\frac{2\log(4/\delta)}{N}}. \tag{36}$$

The proof is given in Sec. H. Hence, combining Lemma 3 with (35), we have the following with probability greater than $1 - 2\delta - \frac{2K}{N^\alpha}$

$$\mathbb{E}_{\boldsymbol{x}\sim\mathcal{D}}\left[\|\widehat{\boldsymbol{f}}(\boldsymbol{x}) - \boldsymbol{\Pi}^\top \boldsymbol{f}^\natural(\boldsymbol{x})\|_2^2\right] \leq \frac{c_3(\zeta + \sqrt{NK\kappa})^2}{N|1 - \sqrt{K\kappa}|^2} + 64N^{-5/8}\left(2\|\boldsymbol{X}\|_{\mathrm{F}}\mathscr{R}_\mathcal{F}\right)^{\frac{1}{4}} + 16\sqrt{\frac{2\log(4/\delta)}{N}}, \tag{37}$$

where

$$\kappa \leq \varphi + 2\sqrt{K}(\xi_1 + \xi_2),$$

$$\zeta^2 \leq 4\sqrt{2}NMK\beta\mathfrak{R}_\mathcal{S}(\mathcal{P}) + 10NM\log(\beta)\frac{\sqrt{2\log(4/\delta)}}{\sqrt{S}} + 2NM\beta\sqrt{K}\nu$$

$$\varphi^2 \leq 4\sqrt{2}N^\alpha MK\beta\mathfrak{R}_\mathcal{S}(\mathcal{P}) + 10N^\alpha M\log(\beta)\frac{\sqrt{2\log(4/\delta)}}{\sqrt{S}} + 2N^\alpha M\beta\sqrt{K}\nu + 4N^\alpha M\delta,$$

$$\mathfrak{R}_\mathcal{S}(\mathcal{P}) = \frac{16}{\sqrt{S}}\sqrt{MK\log\left(4S\sqrt{K}\right) + (2\|\boldsymbol{X}\|_{\mathrm{F}}\mathscr{R}_\mathcal{F})^{\frac{1}{2}}}.$$

Also note that the final bounds in (34) and (37) are satisfied only if $\kappa \leq \frac{1}{K+1}$ as given by Lemma 1. This gives the final conditions on $\xi_1$, $\xi_2$, $\nu$, and $S$. By letting $\delta = \frac{1}{S}$, we obtain the final results in Theorem 1.

## C    PROOF OF PROPOSITION 1

Let us consider the following notation:

$$\boldsymbol{P}(\omega) = \boldsymbol{p}_n^{(m)}, \tag{38}$$

where $\omega = (m, n) \in [M] \times [N]$; i.e., $\boldsymbol{P}(\omega)$ "reads out" the $(m, n)$th block in (10). We also define $\widehat{\mathcal{Y}}(\omega) = \widehat{y}_n^{(m)}$. With this notation, we have

$$D_{\mathcal{S}}(\boldsymbol{P}; \widehat{\mathcal{Y}}) = \frac{1}{S} \sum_{s=1}^{S} \mathsf{CE}(\boldsymbol{P}(\omega_s), \widehat{\mathcal{Y}}(\omega_s)),$$

where $\mathsf{CE}(\boldsymbol{x}, y) = -\sum_{k=1}^{K} \mathbb{I}[y = k] \log(\boldsymbol{x}(k))$, $S = |\mathcal{S}|$, and $\omega_s = (m_s, n_s) \in \mathcal{S}$. Under Assumption 1, we can define:

$$D_{\Pi}(\boldsymbol{P}, \widehat{\mathcal{Y}}) \triangleq \mathbb{E}_{\mathcal{S} \sim \Pi}[\mathsf{CE}(\boldsymbol{P}(\omega), \widehat{\mathcal{Y}}(\omega))] = \sum_{m=1}^{M} \sum_{n=1}^{N} \pi_n^{(m)} \mathsf{CE}(\boldsymbol{p}_n^{(m)}, \widehat{y}_n^{(m)}),$$

where $\Pi$ denotes the uniform distribution and $\pi_n^{(m)} = \frac{1}{NM}$ denotes the probability of observing the annotation for the index pair $(m, n)$.

Eq. (12) implies

$$D_{\mathcal{S}}(\widehat{\boldsymbol{P}}; \widehat{\mathcal{Y}}) \leq D_{\mathcal{S}}(\widetilde{\boldsymbol{P}}; \widehat{\mathcal{Y}}), \tag{39}$$

where $\widetilde{\boldsymbol{P}}$ is defined using the following construction:

$$\widetilde{\boldsymbol{P}} = \boldsymbol{W}^{\natural} \begin{bmatrix} \widetilde{\boldsymbol{f}}(\boldsymbol{x}_1) & \dots & \widetilde{\boldsymbol{f}}(\boldsymbol{x}_N) \end{bmatrix},$$

and $\widetilde{\boldsymbol{f}}$ is a learning function constructed under Assumption 3. To be specific, $\widetilde{\boldsymbol{f}}$ satisfies

$$\|\widetilde{\boldsymbol{f}}(\boldsymbol{x}) - \boldsymbol{f}^{\natural}(\boldsymbol{x})\|_2 \leq \nu, \ \forall \boldsymbol{x} \sim \mathcal{D}.$$

Hence, by taking expectation w.r.t. $\widehat{\mathcal{Y}}$ we have

$$
\begin{aligned}
\mathbb{E}[D_{\Pi}(\widehat{\boldsymbol{P}}; \widehat{\mathcal{Y}}) - D_{\Pi}(\boldsymbol{P}^{\natural}; \widehat{\mathcal{Y}})] &= \mathbb{E}[D_{\Pi}(\widehat{\boldsymbol{P}}; \widehat{\mathcal{Y}})] - D_{\mathcal{S}}(\widehat{\boldsymbol{P}}; \widehat{\mathcal{Y}}) + D_{\mathcal{S}}(\boldsymbol{P}^{\natural}; \widehat{\mathcal{Y}}) - \mathbb{E}[D_{\Pi}(\boldsymbol{P}^{\natural}; \widehat{\mathcal{Y}})] \\
&\quad + D_{\mathcal{S}}(\widehat{\boldsymbol{P}}; \widehat{\mathcal{Y}}) - D_{\mathcal{S}}(\widetilde{\boldsymbol{P}}; \widehat{\mathcal{Y}}) + D_{\mathcal{S}}(\widetilde{\boldsymbol{P}}; \widehat{\mathcal{Y}}) - D_{\mathcal{S}}(\boldsymbol{P}^{\natural}; \widehat{\mathcal{Y}}) \\
&\leq \mathbb{E}[D_{\Pi}(\widehat{\boldsymbol{P}}; \widehat{\mathcal{Y}})] - D_{\mathcal{S}}(\widehat{\boldsymbol{P}}; \widehat{\mathcal{Y}}) + D_{\mathcal{S}}(\boldsymbol{P}^{\natural}; \widehat{\mathcal{Y}}) - \mathbb{E}[D_{\Pi}(\boldsymbol{P}^{\natural}; \widehat{\mathcal{Y}})] \\
&\quad + D_{\mathcal{S}}(\widetilde{\boldsymbol{P}}; \widehat{\mathcal{Y}}) - D_{\mathcal{S}}(\boldsymbol{P}^{\natural}; \widehat{\mathcal{Y}}) \\
&\leq \sup_{\boldsymbol{P} \in \mathcal{P}} \left| D_{\mathcal{S}}(\boldsymbol{P}; \widehat{\mathcal{Y}}) - \mathbb{E}[D_{\Pi}(\boldsymbol{P}; \widehat{\mathcal{Y}})] \right| + \left| D_{\mathcal{S}}(\boldsymbol{P}^{\natural}; \widehat{\mathcal{Y}}) - \mathbb{E}[D_{\Pi}(\boldsymbol{P}^{\natural}; \widehat{\mathcal{Y}})] \right| \\
&\quad + \left| D_{\mathcal{S}}(\widetilde{\boldsymbol{P}}; \widehat{\mathcal{Y}}) - D_{\mathcal{S}}(\boldsymbol{P}^{\natural}; \widehat{\mathcal{Y}}) \right|, \tag{40}
\end{aligned}
$$

where the first inequality is obtained from (39).

Let us consider the L.H.S. of (40):

$$
\begin{aligned}
\mathbb{E}&\left[ D_{\Pi}(\widehat{\boldsymbol{P}}; \widehat{\mathcal{Y}}) - D_{\Pi}(\boldsymbol{P}^{\natural}; \widehat{\mathcal{Y}}) \right] \\
&= \sum_{n=1}^{N} \sum_{m=1}^{M} \left( \pi_n^{(m)} \sum_{k=1}^{K} -\boldsymbol{P}^{\natural}((m-1)K + k, n) \log \widehat{\boldsymbol{P}}((m-1)K + k, n) \right. \\
&\qquad \left. + \boldsymbol{P}^{\natural}((m-1)K + k, n) \log \boldsymbol{P}^{\natural}((m-1)K + k, n) \right) \\
&= \sum_{n=1}^{N} \sum_{m=1}^{M} \pi_n^{(m)} \sum_{k=1}^{K} \boldsymbol{P}^{\natural}((m-1)K + k, n) \log \frac{\boldsymbol{P}^{\natural}((m-1)K + k, n)}{\widehat{\boldsymbol{P}}((m-1)K + k, n)} \\
&= \mathsf{D_{KL}}\left( \boldsymbol{P}^{\natural}, \widehat{\boldsymbol{P}} \right), \tag{41}
\end{aligned}
$$

where expectation is taken w.r.t. $\widehat{\mathcal{Y}}$ (while taking expectation, we used the uniform probability $\pi_n^{(m)} = \frac{1}{NM}, \forall, n, m$ for observing each annotation $\widehat{y}_n^{(m)}$ and used the ground-truth probability $\boldsymbol{P}^\natural((m-1)K+k, n)$ for each event $\mathbb{I}[\widehat{y}_n^{(m)} = k]$) and $\mathsf{D}_{\mathsf{KL}}(\boldsymbol{P}^\natural, \widehat{\boldsymbol{P}})$ is the average Kullback–Leibler (KL) divergence between the entries of the matrices $\boldsymbol{P}^\natural$ and $\widehat{\boldsymbol{P}} \in \mathbb{R}^{MK \times N}$, which is given by

$$\mathsf{D}_{\mathsf{KL}}\left(\boldsymbol{P}^\natural, \widehat{\boldsymbol{P}}\right) = \frac{1}{NM} \sum_{n=1}^{N} \sum_{m=1}^{M} \mathsf{D}_{\mathsf{KL}}\left(\boldsymbol{p}_n^{\natural(m)}, \widehat{\boldsymbol{p}}_n^{(m)}\right).$$

**Upper-bounding the first term on the R.H.S of** (40)**.** Next, we characterize the first term on the R.H.S. of (40). To achieve this, we invoke the following theorem (Theorem 26.5 in (Shalev-Shwartz & Ben-David, 2014)) :

**Theorem 4** *(Shalev-Shwartz & Ben-David, 2014, Theorem 26.5) Assume that for all $y$ and for all $\boldsymbol{x}$, we have $|\mathsf{CE}(\boldsymbol{x}; y)| \leq z_{\max}$. Then for any $\boldsymbol{P} \in \mathcal{P}$, the following holds with probability greater than $1 - \delta$:*

$$\left| D_{\mathcal{S}}(\boldsymbol{P}; \widehat{\mathcal{Y}}) - \mathbb{E}[D_\Pi(\boldsymbol{P}; \widehat{\mathcal{Y}})] \right| \leq 2\mathfrak{R}_{\mathcal{S}}(\ell \circ \mathcal{P} \circ \mathcal{S}) + 4z_{\max}\sqrt{\frac{2\log(4/\delta)}{S}}, \qquad (42)$$

*where $\ell \circ \mathcal{P} \circ \mathcal{S}$ denotes the set*

$$\ell \circ \mathcal{P} \circ \mathcal{S} \triangleq \left\{ \left( \mathsf{CE}(\boldsymbol{P}(\omega_1); \widehat{\mathcal{Y}}(\omega_1)), \ldots, \mathsf{CE}(\boldsymbol{P}(\omega_S); \widehat{\mathcal{Y}}(\omega_S)) \right) \mid \boldsymbol{P} \in \mathcal{P} \right\}$$

*and $\mathfrak{R}_{\mathcal{S}}(\mathcal{X})$ denotes the empirical Rademacher complexity of the set $\mathcal{X}$.*

To apply Theorem 4, we will characterize $\mathfrak{R}_{\mathcal{S}}(\ell \circ \mathcal{P} \circ \mathcal{S})$ which is defined as follows (Shalev-Shwartz & Ben-David, 2014):

$$\mathfrak{R}_{\mathcal{S}}(\ell \circ \mathcal{P} \circ \mathcal{S}) \triangleq \frac{1}{S}\mathbb{E}\left[ \sup_{\boldsymbol{P} \in \mathcal{P}} \sum_{s=1}^{S} \sigma_s \mathsf{CE}(\boldsymbol{P}(\omega_s); \widehat{\mathcal{Y}}(\omega_s)) \right], \qquad (43)$$

where expectation is w.r.t. the independent Rademacher random variables $\sigma_s \in \{-1, 1\}$. Note that $\boldsymbol{P}(\omega)$ is a vector-valued [see Eq. (38)]. Hence, we invoke the following contraction result to upper bound $\mathfrak{R}_{\mathcal{S}}(\ell \circ \mathcal{P} \circ \mathcal{S})$:

**Lemma 4** *(Maurer, 2016) Let $\mathcal{P}$ be a class of mappings $\{\boldsymbol{P} : \mathcal{X} \to \mathbb{R}^K\}$, where $\mathcal{X}$ be any set and $(\omega_1, \ldots, \omega_S) \in \mathcal{X}^S$. Also assume that $\ell : \mathbb{R}^K \to \mathbb{R}$ has the Lipschitz constant $L$. Then*

$$\mathbb{E}\left[ \sup_{\boldsymbol{P} \in \mathcal{P}} \sum_{s=1}^{S} \sigma_s \ell(\boldsymbol{P}(\omega_s)) \right] \leq \sqrt{2}L\mathbb{E}\left[ \sup_{\boldsymbol{P} \in \mathcal{P}} \sum_{s,k} \sigma_{sk} \boldsymbol{P}_k(w_s) \right]$$

*where $\boldsymbol{P}_k(w)$ denotes the kth component of $\boldsymbol{P}(\omega)$, $\sigma_{sk}$ is an independent (doubly indexed) Rademacher random variable and the expectations are taken w.r.t. the Rademacher random variables.*

Let us define a vector $\boldsymbol{z} \triangleq (\boldsymbol{P}_1(\omega_1), \boldsymbol{P}_2(\omega_1), \ldots, \boldsymbol{P}_{K-1}(\omega_s), \boldsymbol{P}_K(\omega_s)) \in \mathbb{R}^{SK}$ and the set $\mathcal{Z} \triangleq \{\boldsymbol{z} = (\boldsymbol{P}_1(\omega_1), \ldots, \boldsymbol{P}_K(\omega_s)) \mid \boldsymbol{P} \in \mathcal{P}\}$. With these definitions, we apply Lemma 4 in (43) and obtain

$$\mathfrak{R}_{\mathcal{S}}(\ell \circ \mathcal{P} \circ \mathcal{S}) \leq \frac{\sqrt{2}\beta}{S}\mathbb{E}\left[ \sup_{\boldsymbol{P} \in \mathcal{P}} \sum_{s,k} \sigma_{s,k} \boldsymbol{P}_k(w_s) \right]$$

$$= \frac{\sqrt{2}\beta}{S}\mathbb{E}\left[ \sup_{\boldsymbol{z} \in \mathcal{Z}} \sum_{i} \sigma_i \boldsymbol{z}(i) \right]$$

$$= \sqrt{2}\beta K \frac{1}{SK}\mathbb{E}\left[ \sup_{\boldsymbol{z} \in \mathcal{Z}} \sum_{i} \sigma_i \boldsymbol{z}(i) \right] = \sqrt{2}\beta K \mathfrak{R}_{\mathcal{S}}(\mathcal{Z}), \qquad (44)$$

where $\beta$ is an upper bound of the Lipschitz constant of the cross entropy loss function $\mathsf{CE}(\boldsymbol{x}; y) = -\sum_{k=1}^{K} \mathbb{I}[y = k] \log \boldsymbol{x}(k)$ when $\boldsymbol{x} \in \boldsymbol{\Delta}_K$ with $\boldsymbol{x}(k) > (1/\beta). \forall k$.

Next, we will characterize $\mathfrak{R}_{\mathcal{S}}(\mathcal{Z})$ using the covering number of the set $\mathcal{Z}$.

**Definition 2** *(Vershynin, 2012) The $\epsilon$-net covering of the set $\mathcal{Z}$ (denoted as $\overline{\mathcal{Z}}$) is a finite subset of $\mathcal{Z}$ (i.e., $\overline{\mathcal{Z}} \subseteq \mathcal{Z}$) such that for any $\boldsymbol{z} \in \mathcal{Z}$, there exists an $\overline{\boldsymbol{z}} \in \overline{\mathcal{Z}}$ satisfying $\|\boldsymbol{z} - \overline{\boldsymbol{z}}\|_2^2 \leq \epsilon$. The smallest cardinality of the $\epsilon$-nets of $\overline{\mathcal{Z}}$ is known as the covering number of $\mathcal{Z}$, which is denoted as $\overline{\mathsf{N}}(\epsilon, \mathcal{Z})$.*

Let us consider a pair of vectors $\boldsymbol{z}, \overline{\boldsymbol{z}} \in \mathcal{Z}$ as below:

$$\boldsymbol{z} = (\boldsymbol{P}_1(\omega_1), \ldots, \boldsymbol{P}_K(\omega_s)) \in \mathbb{R}^{SK}, \ \boldsymbol{P}(\omega_s) = \boldsymbol{A}_{m_s}\boldsymbol{F}(:,n_s),$$
$$\overline{\boldsymbol{z}} = (\overline{\boldsymbol{P}}_1(\omega_1), \ldots, \overline{\boldsymbol{P}}_K(\omega_s)) \in \mathbb{R}^{SK}, \ \overline{\boldsymbol{P}}(\omega_s) = \overline{\boldsymbol{A}}_{m_s}\overline{\boldsymbol{F}}(:,n_s),$$

where $\omega_s = (m_s, n_s)$ and all $\boldsymbol{A}_m, \overline{\boldsymbol{A}}_m, \boldsymbol{F}$, and $\overline{\boldsymbol{F}}$ satisfy the nonnegativity constraints and have unit $\ell_1$-norm on the columns. Then, we have

$$\|\boldsymbol{z} - \overline{\boldsymbol{z}}\|^2 = \sum_{s=1}^{S} \|\boldsymbol{A}_{m_s}\boldsymbol{F}(:,n_s) - \overline{\boldsymbol{A}}_{m_s}\overline{\boldsymbol{F}}(:,n_s)\|_2^2$$

$$\leq \sum_{s=1}^{S} \|\boldsymbol{A}_{m_s}\boldsymbol{F}(:,n_s) - \overline{\boldsymbol{A}}_{m_s}\overline{\boldsymbol{F}}(:,n_s)\|_2$$

$$= \sum_{s=1}^{S} \|\boldsymbol{A}_{m_s}\boldsymbol{F}(:,n_s) - \overline{\boldsymbol{A}}_{m_s}\boldsymbol{F}(:,n_s) + \overline{\boldsymbol{A}}_{m_s}\boldsymbol{F}(:,n_s) - \overline{\boldsymbol{A}}_{m_s}\overline{\boldsymbol{F}}(:,n_s)\|_2$$

$$\leq \sum_{s=1}^{S} \|\boldsymbol{A}_{m_s} - \overline{\boldsymbol{A}}_{m_s}\|_{\mathrm{F}}\|\boldsymbol{F}(:,n_s)\|_2 + \|\overline{\boldsymbol{A}}_{m_s}\|_{\mathrm{F}}\|\boldsymbol{F}(:,n_s) - \overline{\boldsymbol{F}}(:,n_s)\|_2$$

$$\leq \sum_{s=1}^{S} \|\boldsymbol{A}_{m_s} - \overline{\boldsymbol{A}}_{m_s}\|_{\mathrm{F}} + \sqrt{K}\sum_{s=1}^{S} \|\boldsymbol{F}(:,n_s) - \overline{\boldsymbol{F}}(:,n_s)\|_2$$

where the first inequality is by $\|\boldsymbol{x}\|_2^2 \leq \|\boldsymbol{x}\|_2$ if the entries of $\boldsymbol{x}$ are smaller than 1. The second inequality is by triangle inequality. The last inequality uses the fact the Frobenius norm of $\overline{\boldsymbol{A}}_m$'s are bounded by $\sqrt{K}$ and the $\ell_2$ norm of any column of $\boldsymbol{F}$ is bounded by 1. Hence, to obtain an $\varepsilon$-net covering for the set $\mathcal{Z}$ (i.e., $\|\boldsymbol{z} - \overline{\boldsymbol{z}}\| \leq \varepsilon$), we only need to show that there exists a $\frac{\varepsilon^2}{2\sqrt{K}}$-net covering for $\mathcal{F} \circ \mathcal{S}$ and a $\frac{\varepsilon^2}{2S}$-net covering for each $\boldsymbol{A}_m$'s since

$$\sum_{s=1}^{S} \frac{\varepsilon^2}{2S} + \sqrt{K}\frac{\varepsilon^2}{2\sqrt{K}} = \varepsilon^2.$$

Here $\mathcal{F} \circ \mathcal{S}$ denotes

$$\mathcal{F} \circ \mathcal{S} = \{[\boldsymbol{f}(\boldsymbol{x}_{n_1}), \ldots, \boldsymbol{f}(\boldsymbol{x}_{n_S})] \in \mathbb{R}^{K \times S} \mid \boldsymbol{f} \in \mathcal{F}\},$$

where $(m_s, n_s) = \omega_s \in \mathcal{S}$. Note that the full rank matrix $\boldsymbol{A}_m \in \mathbb{R}^{K \times K}$ can be represented as a $K^2$-dimensional vector whose Euclidean norm is bounded by $\sqrt{K}$. Hence, the cardinality of the $\frac{\varepsilon^2}{2S}$-net covering for $\boldsymbol{A}_m \in \mathbb{R}^{K \times K}$ is at most $\left(\frac{4SK\sqrt{K}}{\varepsilon^2}\right)^{K^2}$ (Shalev-Shwartz & Ben-David, 2014). Next, we consider the covering number corresponding to the function class $\mathcal{F} \circ \mathcal{S}$. Using Lemma 14 of (Lin & Zhang, 2019), we get the cardinality of the $\frac{\varepsilon^2}{2\sqrt{K}}$-net covering for $\mathcal{F} \circ \mathcal{S}$ as below:

$$\overline{\mathsf{N}}\left(\frac{\varepsilon^2}{2\sqrt{K}}, \mathcal{F} \circ \mathcal{S}\right) \leq \exp\left(\sqrt{\frac{2\sqrt{K}\|\boldsymbol{X}\|_{\mathrm{F}}\mathscr{R}_{\mathcal{F}}}{\varepsilon^2}}\right),$$

where $\boldsymbol{X} = [\boldsymbol{x}_{n_1}, \ldots, \boldsymbol{x}_{n_S}] \in \mathbb{R}^{d \times S}$ and the parameter $\mathscr{R}_{\mathcal{F}}$ is from Assumption 2.

Using the covering number results, the cardinality of the $\varepsilon$-net covering of set $\mathcal{Z}$ is bounded by the following:

$$\overline{\mathsf{N}}(\varepsilon, \mathcal{Z}) \leq \left(\frac{4SK\sqrt{K}}{\varepsilon^2}\right)^{MK^2} \times \exp\left(\sqrt{\frac{2\sqrt{K}\|\boldsymbol{X}\|_{\mathrm{F}}\mathscr{R}_{\mathcal{F}}}{\varepsilon^2}}\right). \tag{45}$$

Now that we have characterized $\overline{\mathsf{N}}(\epsilon, \mathcal{Z})$, we invoke the below lemma to obtain the Rademacher complexity $\mathfrak{R}_{\mathcal{S}}(\mathcal{Z})$:

**Lemma 5** *(Bartlett et al., 2017, Lemma A.5) The empirical Rademacher complexity of the set $\mathcal{Z}$ with respect to the observed set $\mathcal{S}$ having size $SK$ is upper bounded as follows:*

$$\mathfrak{R}_{\mathcal{S}}(\mathcal{Z}) \leq \inf_{a>0} \left( \frac{4a}{\sqrt{SK}} + \frac{12}{SK} \int_a^{\sqrt{SK}} \sqrt{\log \overline{\mathsf{N}}(\mu, \mathcal{Z})} d\mu \right). \tag{46}$$

We apply (45) in Lemma 5 and obtain

$$
\begin{aligned}
\mathfrak{R}_{\mathcal{S}}(\mathcal{Z}) &\overset{(a)}{\leq} \inf_{a>0} \left( \frac{4a}{\sqrt{SK}} + \frac{12}{SK} \sqrt{SK} \sqrt{\log \overline{\mathsf{N}}(a, \mathcal{Z})} \right) \\
&\overset{(b)}{\leq} \inf_{a>0} \left( \frac{4a}{\sqrt{SK}} + \frac{12}{\sqrt{SK}} \sqrt{MK^2 \log\left( \frac{4SK\sqrt{K}}{a^2} \right) + \left( \frac{2\sqrt{K}\|\boldsymbol{X}\|_{\mathrm{F}} \mathscr{R}_{\mathcal{F}}}{a^2} \right)^{\frac{1}{2}}} \right) \\
&\overset{(c)}{\leq} \frac{4}{\sqrt{S}} + \frac{12}{\sqrt{SK}} \sqrt{MK^2 \log\left( 4S\sqrt{K} \right) + (2\|\boldsymbol{X}\|_{\mathrm{F}} \mathscr{R}_{\mathcal{F}})^{\frac{1}{2}}} \\
&\leq \frac{16}{\sqrt{S}} \sqrt{MK \log\left( 4S\sqrt{K} \right) + (2\|\boldsymbol{X}\|_{\mathrm{F}} \mathscr{R}_{\mathcal{F}})^{\frac{1}{2}}}. \tag{47}
\end{aligned}
$$

In the above, the first inequality $(a)$ is obtained by using the relation

$$\int_a^{\sqrt{SK}} \sqrt{\log \overline{\mathsf{N}}(\mu, \mathcal{Z})} d\mu \leq \sqrt{SK} \sqrt{\log \overline{\mathsf{N}}(a, \mathcal{Z})},$$

which holds because $\sqrt{\log \overline{\mathsf{N}}(\mu, \mathcal{Z})}$ decreases monotonically as $\mu$ increases. The inequality $(b)$ is obtained by applying (45), and $(c)$ is obtained by fixing $a = \sqrt{K}$ which is smaller than $\sqrt{SK}$.

Combining the upperbound of $\mathfrak{R}_{\mathcal{S}}(\mathcal{Z})$ given by (47) with the upper bound of $\mathfrak{R}_{\mathcal{S}}(\ell \circ \mathcal{P} \circ \mathcal{S})$ as given by (44) and with the result in (42), we get that with probability greater than $1 - \delta$,

$$\left| D_{\mathcal{S}}(\boldsymbol{P}; \widehat{\mathcal{Y}}) - \mathbb{E}[D_{\Pi}(\boldsymbol{P}; \widehat{\mathcal{Y}})] \right| \leq 2\sqrt{2}\beta K \mathfrak{R}_{\mathcal{S}}(\mathcal{Z}) + 4z_{\max} \sqrt{\frac{2\log(4/\delta)}{S}}, \tag{48}$$

where $\mathfrak{R}_{\mathcal{S}}(\mathcal{Z})$ is upper bounded by (47) and $z_{\max}$ is the upperbound of the value of the function $\mathsf{CE}(\boldsymbol{x}, y)$ which can be characterized as below:

$$z_{\max} = \max_{\substack{\boldsymbol{x}(k) > \frac{1}{\beta} \\ y \in [K]}} \mathsf{CE}(\boldsymbol{x}, y) \leq \max_{\substack{\boldsymbol{x}(k) > \frac{1}{\beta} \\ y \in [K]}} - \sum_{k=1}^{K} \mathbb{I}[y = k] \log \boldsymbol{x}(k) \leq \max_{u > \frac{1}{\beta}} - \log u = \log(\beta). \tag{49}$$

**Upper-bounding the second term on the R.H.S of** (40). Next, we proceed to upper bound the second term on the R.H.S of (40). Let us consider the Hoeffding's inequality

**Lemma 6** *Let $Z_1, \ldots, Z_S$ be independent bounded random variables with $Z_s \in [z_{\min}, z_{\max}]$ for all $s$ where $-\infty < z_{\min} \leq z_{\max} < \infty$. Then for all $t \geq 0$,*

$$\Pr\left( \frac{1}{S} \sum_{s=1}^{S} (Z_s - \mathbb{E}[Z_s]) \geq t \right) \leq \exp\left( -\frac{2St^2}{(z_{\max} - z_{\min})^2} \right).$$

To use Lemma 6, let us define the random variable $Z_n^{(m)}$ as follows:

$$Z_n^{(m)} \triangleq \mathsf{CE}(\boldsymbol{p}_n^{\natural(m)}, \widehat{y}_n^{(m)}),$$

where $\boldsymbol{p}_n^{\natural(m)} = \boldsymbol{A}_m^{\natural} \boldsymbol{f}^{\natural}(\boldsymbol{x}_n)$. The maximum and minimum values of $Z_n^{(m)}$ are $z_{\max} = \log(\beta)$ (see (49)) and $z_{\min} = 0$, respectively. Then, invoking Lemma 6, one can obtain

$$\Pr\left( D_{\mathcal{S}}(\boldsymbol{P}^{\natural}; \widehat{\mathcal{Y}}) - \mathbb{E}[D_{\Pi}(\boldsymbol{P}^{\natural}; \widehat{\mathcal{Y}})] \geq t \right) \leq \exp\left( -\frac{2St^2}{(\log(\beta))^2} \right). \tag{50}$$

Hence, by substituting $t = \log(\beta)\sqrt{\frac{\log\left(\frac{1}{\delta}\right)}{2S}}$, where $\delta \in (0,1)$ in (50), we get that with probability greater than $1 - \delta$

$$D_{\mathcal{S}}(\boldsymbol{P}^{\natural}; \widehat{\mathcal{Y}})] - \mathbb{E}[D_{\Pi}(\boldsymbol{P}^{\natural}; \widehat{\mathcal{Y}})] \le \log(\beta)\sqrt{\frac{\log\left(\frac{1}{\delta}\right)}{2S}}. \tag{51}$$

**Upper-bounding the third term on the R.H.S of** (40)**.**

$$
\begin{aligned}
\left| D_{\mathcal{S}}(\widetilde{\boldsymbol{P}}; \widehat{\mathcal{Y}}) - D_{\mathcal{S}}(\boldsymbol{P}^{\natural}; \widehat{\mathcal{Y}}) \right| &= \left| -\frac{1}{S} \sum_{(m,n) \in \mathcal{S}} \sum_{k=1}^{K} \mathbb{I}[\widehat{y}_n^{(m)} = k] \log \widetilde{\boldsymbol{P}}((m-1)K + k, n) \right. \\
&\quad \left. +\frac{1}{S} \sum_{(m,n) \in \mathcal{S}} \sum_{k=1}^{K} \mathbb{I}[\widehat{y}_n^{(m)} = k] \log \boldsymbol{P}^{\natural}((m-1)K + k, n) \right| \\
&\le \frac{1}{S} \sum_{(m,n) \in \mathcal{S}} \sum_{k=1}^{K} \mathbb{I}[\widehat{y}_n^{(m)} = k] \left| \log[\boldsymbol{A}_m^{\natural} \boldsymbol{f}^{\natural}(\boldsymbol{x}_n)]_k - \log[\boldsymbol{A}_m^{\natural} \widetilde{\boldsymbol{f}}(\boldsymbol{x}_n)]_k \right| \\
&\le \frac{1}{S} \sum_{(m,n) \in \mathcal{S}} \sum_{k=1}^{K} \mathbb{I}[\widehat{y}_n^{(m)} = k]\beta \left| [\boldsymbol{A}_m^{\natural} \boldsymbol{f}^{\natural}(\boldsymbol{x}_n)]_k - [\boldsymbol{A}_m^{\natural} \widetilde{\boldsymbol{f}}(\boldsymbol{x}_n)]_k \right| \\
&\le \frac{1}{S} \sum_{(m,n) \in \mathcal{S}} \sum_{k=1}^{K} \mathbb{I}[\widehat{y}_n^{(m)} = k]\beta \| \boldsymbol{A}_m^{\natural}(k,:) \|_2 \| \boldsymbol{f}^{\natural}(\boldsymbol{x}_n) - \widetilde{\boldsymbol{f}}(\boldsymbol{x}_n) \|_2 \\
&\le \beta \sqrt{K} \nu, \tag{52}
\end{aligned}
$$

where the first inequality uses the triangle inequality, the second inequality uses the Lipschitz continuity of $\log$ function, the third inequality is via Cauchy Schwartz inequality, and the last inequality employs Assumption 3.

**Putting Together.** Hence, by combining the result in (48) with (41), (40), (51), and (52), we get that with probability greater than $1 - 2\delta$,

$$\mathsf{D}_{\mathsf{KL}}\left(\boldsymbol{P}^{\natural}, \widehat{\boldsymbol{P}}\right) \le 2\sqrt{2}\beta K \mathfrak{R}_{\mathcal{S}}(\mathcal{Z}) + 4\log(\beta)\sqrt{\frac{2\log(4/\delta)}{S}} + \log(\beta)\sqrt{\frac{\log\left(\frac{1}{\delta}\right)}{2S}} + \beta\sqrt{K}\nu. \tag{53}$$

Using Pinsker's inequality (S, 1960; Fedotov et al., 2003), we get

$$
\begin{aligned}
\mathsf{D}_{\mathsf{KL}}\left(\boldsymbol{P}^{\natural}, \widehat{\boldsymbol{P}}\right) = \frac{1}{NM} \sum_{n=1}^{N} \sum_{m=1}^{M} \mathsf{D}_{\mathsf{KL}}\left(\boldsymbol{p}_n^{\natural(m)}, \widehat{\boldsymbol{p}}_n^{(m)}\right) &\ge \frac{1}{2NM} \sum_{n=1}^{N} \sum_{m=1}^{M} \| \boldsymbol{p}_n^{\natural(m)} - \widehat{\boldsymbol{p}}_n^{(m)} \|_1^2 \\
&\ge \frac{1}{2NM} \sum_{n=1}^{N} \sum_{m=1}^{M} \| \boldsymbol{p}_n^{\natural(m)} - \widehat{\boldsymbol{p}}_n^{(m)} \|_2^2
\end{aligned}
$$

where the last inequality uses the fact that $\|\boldsymbol{x}\|_1 \ge \|\boldsymbol{x}\|_2$. The above relation combined with (53), implies that with probability greater than $1 - 2\delta$:

$$\frac{1}{NM} \sum_{n=1}^{N} \sum_{m=1}^{M} \| \boldsymbol{p}_n^{\natural(m)} - \widehat{\boldsymbol{p}}_n^{(m)} \|_2^2 \le 4\sqrt{2}K\beta\mathfrak{R}_{\mathcal{S}}(\mathcal{Z}) + 10\log(\beta)\frac{\sqrt{2\log(4/\delta)}}{\sqrt{S}} + 2\beta\sqrt{K}\nu, \tag{54}$$

where $\mathfrak{R}_{\mathcal{S}}(\mathcal{Z})$ is upper-bounded in (47)

# D PROOF OF THEOREM 2

The proof of Theorem 2 utilizes some key results from the proof of Theorem 1. We start by employing the result from Proposition 1. Let us fix $\delta = \frac{1}{S}$ and $\alpha = \frac{1}{8}$ in the result of Proposition 1. Then, it

implies that when $\boldsymbol{f}^\natural \in \mathcal{F}$, i.e., $\nu = 0$, $S \geq C_1 t$, $N \leq C_2 t^3$, for certain constants $C_1, C_2 > 0$ and at the limit of $t \to \infty$, we get the following:

$$\|\boldsymbol{P}^\natural - \widehat{\boldsymbol{P}}\|_{\mathrm{F}}^2 = 0,$$

with probability 1, i.e.,

$$\widehat{\boldsymbol{P}} = \boldsymbol{W}^\natural \boldsymbol{F}^\natural. \tag{55}$$

On the other hand, the matrix $\widehat{\boldsymbol{P}}$ can be constructed using the estimates of the CCEM criterion (6) via

$$\widehat{\boldsymbol{P}} = \widehat{\boldsymbol{W}} \widehat{\boldsymbol{F}}. \tag{56}$$

Hence, in order to identify $\boldsymbol{W}^\natural$ and $\boldsymbol{F}^\natural$ from the NMF model in (55), we invoke the following result:

**Lemma 7** *(Huang et al., 2014) Consider the matrix factorization model $\boldsymbol{Z} = \boldsymbol{X}\boldsymbol{Y}$, where $\boldsymbol{X} \in \mathbb{R}^{I \times K}$, $\boldsymbol{Y} \in \mathbb{R}^{K \times J}$ and $\mathsf{rank}(\boldsymbol{X}) = \mathsf{rank}(\boldsymbol{Y}) = K$. If $\boldsymbol{X}, \boldsymbol{Y} \geq 0$ and both $\boldsymbol{X}$ and $\boldsymbol{Y}$ satisfy SSC, then any $\widehat{\boldsymbol{X}}$ and $\widehat{\boldsymbol{Y}}$ that satisfy $\boldsymbol{Z} = \widehat{\boldsymbol{X}}\widehat{\boldsymbol{Y}}$ must have the following form:*

$$\widehat{\boldsymbol{X}} = \boldsymbol{X}\boldsymbol{\Pi}\boldsymbol{\Sigma}, \; \widehat{\boldsymbol{Y}} = \boldsymbol{\Sigma}^{-1}\boldsymbol{\Pi}^\top \boldsymbol{Y},$$

*where $\boldsymbol{\Pi}$ is a column permutation matrix and $\boldsymbol{\Sigma}$ is a diagonal nonnegative and scaling matrix.*

Next, we show that Lemma 7 can be applied given the conditions in Theorem 2. The matrix $\boldsymbol{F}_{\mathcal{Z}}^\natural = [\boldsymbol{f}^\natural(\boldsymbol{x}_{n_1}), \ldots, \boldsymbol{f}^\natural(\boldsymbol{x}_{n_T})]$ includes a subset of columns of $\boldsymbol{F}^\natural$, i.e., $\mathrm{cone}(\boldsymbol{F}_{\mathcal{Z}}^\natural) \subseteq \mathrm{cone}(\boldsymbol{F}^\natural)$. Hence, we have

$$\mathcal{C} \subseteq \mathrm{cone}(\boldsymbol{F}_{\mathcal{Z}}^\natural) \implies \mathcal{C} \subseteq \mathrm{cone}(\boldsymbol{F}^\natural), \tag{57}$$

where $\mathcal{C} = \{\boldsymbol{x} \in \mathbb{R}^K | \sqrt{K-1}\|\boldsymbol{x}\|_2 \leq \boldsymbol{1}^\top \boldsymbol{x}\}$ is the second-order cone.

Also for any orthogonal matrix $\boldsymbol{Q} \in \mathbb{R}^{K \times K}$ except for the permutation matrices, the below holds:

$$\mathrm{cone}(\boldsymbol{F}_{\mathcal{Z}}^\natural) \not\subset \mathrm{cone}\{\boldsymbol{Q}\} \implies \mathrm{cone}(\boldsymbol{F}^\natural) \not\subset \mathrm{cone}\{\boldsymbol{Q}\}. \tag{58}$$

Eqs (57) and (58) imply that $\boldsymbol{F}^\natural$ satisfies SSC, given the assumption that $\boldsymbol{F}_{\mathcal{Z}}^\natural$ satisfies SSC. In addition, the column scaling does not affect the conic hull, i.e,

$$\mathrm{cone}(\boldsymbol{F}^\natural) = \mathrm{cone}(\boldsymbol{\Sigma}^{-1}\boldsymbol{F}^\natural),$$

$$\mathrm{cone}(\boldsymbol{W}^{\natural\top}) = \mathrm{cone}(\boldsymbol{\Sigma}\boldsymbol{W}^{\natural\top}).$$

Since both $\boldsymbol{W}^\natural$ and $\boldsymbol{F}^\natural$ satisfy SSC, $\mathsf{rank}(\boldsymbol{F}^\natural) = \mathsf{rank}(\boldsymbol{W}^\natural) = K$ hold (see (Huang et al., 2016)). Hence, the conditions in Lemma 7 holds for (55). Comparing (55) and (56) and invoking Lemma 7, we get

$$\widehat{\boldsymbol{W}} = \boldsymbol{W}^\natural \boldsymbol{\Pi}, \tag{59a}$$

$$\widehat{\boldsymbol{F}} = \boldsymbol{\Pi}^\top \boldsymbol{F}^\natural, \tag{59b}$$

where the scaling $\boldsymbol{\Sigma}$ is automatically removed due to the sum-to-one constraints on the columns of $\widehat{\boldsymbol{A}}_m$'s and $\widehat{\boldsymbol{F}}$.

The first result (59a) implies that

$$\widehat{\boldsymbol{A}}_m = \boldsymbol{A}_m^\natural \boldsymbol{\Pi}, \forall m.$$

Applying the second result (59b) in Lemma 3 along with the assumption that $N \to \infty$, we get

$$\mathbb{E}_{\boldsymbol{x} \sim \mathcal{D}} \left[ \|\widehat{\boldsymbol{f}}(\boldsymbol{x}) - \boldsymbol{\Pi}^\top \boldsymbol{f}^\natural(\boldsymbol{x})\|_2^2 \right] = 0. \tag{60}$$

**Fact 1** Let $X$ be a nonnegative random variable with $\mathbb{E}[X] = 0$. Then, $X$ is zero almost surely, i.e.,

$$\Pr(X = 0) = 1.$$

Employing Fact 1 in (60), we get that

$$\|\widehat{\boldsymbol{f}}(\boldsymbol{x}) - \boldsymbol{\Pi}^\top \boldsymbol{f}^\natural(\boldsymbol{x})\|_2^2 = 0, \forall \boldsymbol{x} \sim \mathcal{D}, \implies \widehat{\boldsymbol{f}}(\boldsymbol{x}) = \boldsymbol{\Pi}^\top \boldsymbol{f}^\natural(\boldsymbol{x}), \forall \boldsymbol{x} \sim \mathcal{D},$$

due to the nonnegativity of $\boldsymbol{f}^\natural$ and $\widehat{\boldsymbol{f}}$.

## E    PROOF OF THEOREM 3

To prove Theorem 3, let us first consider the CCEM criterion in (6). Let $\widehat{W}$ and $\widehat{F}$ be any optimal solution of (6) and

$$\widehat{P} = \widehat{W}\widehat{F}. \tag{61}$$

From Proposition 1, by fixing $\delta = \frac{1}{S}$ and $\alpha = \frac{1}{8}$ and using the conditions in the statement of Theorem 3, i.e., $f^\natural \in \mathcal{F}$ implying $\nu = 0$, $S \geq C_1 t$, $N \leq C_2 t^3$, for certain constants $C_1, C_2 > 0$, we get the following at the limit of $t \to \infty$:

$$\|P^\natural - \widehat{P}\|_{\mathrm{F}}^2 = 0,$$

with probability 1, i.e.,

$$\widehat{P} = W^\natural F^\natural. \tag{62}$$

**Proof of Theorem 3(a):**    We consider the following result which is distilled and summarized from the proof of Theorem 1 in (Fu et al., 2015):

**Lemma 8** *Suppose a matrix $Y \in \mathbb{R}^{K \times J}$ satisfies $Y \geq 0$, $\mathbf{1}^\top Y = \mathbf{1}^\top$ , rank$(Y) = K$, and SSC. Then, for any $\widehat{Y} = QY$ satisfying $\widehat{Y} \geq 0$, $\mathbf{1}^\top \widehat{Y} = \mathbf{1}^\top$, the following holds:*

$$|\det(Q)| \leq 1,$$

*The equality holds only if $Q$ is a permutation matrix.*

Let us start by considering the following criterion:

$$\underset{W,F}{\text{maximize}} \ \det(FF^\top) \tag{63a}$$

$$\text{s.t. } \widehat{P} = WF \tag{63b}$$

$$F \geq 0, \mathbf{1}^\top F = \mathbf{1}^\top \tag{63c}$$

where $\widehat{P}$ is the optimal solution from the CCEM criterion (6) and hence, as shown before, $\widehat{P}$ satisfies (62).

Let $W^*$ and $F^*$ be optimal solutions of (63), then the following holds:

$$\det(FF^{*\top}) \geq \det(F^\natural F^{\natural\top}). \tag{64}$$

From (62), we observe that $W^*$ and $F^*$ satisfies $W^* = W^\natural Q^{-1}$, $F^* = QF^\natural$, for a certain invertible matrix $Q$. Let us assume that $Q$ is not a permutation matrix, Then, we have

$$\det(F^* F^{*\top}) = \det(Q^\top F^\natural F^{\natural\top} Q)$$
$$= |\det(Q)|^2 \det(F^\natural F^{\natural\top})$$
$$< \det(F^\natural F^{\natural\top})$$

where the last inequality is from Lemma 8 using the SSC condition on $F^\natural$. Note that the result is a contradiction from (64). Hence, $Q$ must be a permutation matrix. This implies that the optimal solution $W^*$ and $F^*$ of Problem 63 satisfies the following:

$$W^* = W^\natural \Pi, \tag{65a}$$
$$F^* = \Pi^\top F^\natural, \tag{65b}$$

Following the last part of Theorem 2, by using Lemma 3 and Fact 1, we have

$$A_m^* = A_m^\natural \Pi, \forall m.$$
$$f^*(x) = \Pi^\top f^\natural(x), \forall x \sim \mathcal{D}.$$

Note that since $\log$ is a monotonically increasing function, by using $\log \det(\boldsymbol{F}\boldsymbol{F}^\top)$ in (63) in place of $\det(\boldsymbol{F}\boldsymbol{F})^\top$ does not change the optimal solution, yet it keeps the objective function in differentiable domain (this is because, $\det(\boldsymbol{X})$ becomes zero if $\boldsymbol{X}$ is singular). Hence, we can conclude that the following criterion

$$\underset{\boldsymbol{W},\boldsymbol{F}}{\text{maximize}} \ \log \det(\boldsymbol{F}\boldsymbol{F}^\top) \tag{66a}$$

$$\text{s.t. } \widehat{\boldsymbol{P}} = \boldsymbol{W}\boldsymbol{F} \tag{66b}$$

$$\boldsymbol{F} \geq \mathbf{0}, \tag{66c}$$

also results in the same optimal solutions $\boldsymbol{W}^*$ and $\boldsymbol{F}^*$ satisfying (65).

**Proof of Theorem 3(b):** We consider the following result which is a modified version of Lemma A.1 from (Huang et al., 2015):

**Lemma 9** *Suppose a matrix $\boldsymbol{X} \in \mathbb{R}^{I \times K}$ satisfies $\boldsymbol{X} \geq \mathbf{0}$, $\mathbf{1}^\top \boldsymbol{X} = \rho \mathbf{1}^\top$, $\rho > 0$, $\mathrm{rank}(\boldsymbol{X}) = K$, and SSC. Then, for any $\widehat{\boldsymbol{X}} = \boldsymbol{X}\boldsymbol{Q}$ satisfying $\widehat{\boldsymbol{X}} \geq \mathbf{0}$, $\mathbf{1}^\top \widehat{\boldsymbol{X}} = \rho \mathbf{1}^\top$, the following holds:*
$$|\det(\boldsymbol{Q})| \leq 1,$$
*The equality holds only if $\boldsymbol{Q}$ is a permutation matrix.*

Note that, in Lemma 9, we are given that $\mathbf{1}^\top \boldsymbol{X} = \rho \mathbf{1}^\top$. Then, by multiplying $\boldsymbol{Q}$ on both sides, we obtain

$$\mathbf{1}^\top \boldsymbol{X}\boldsymbol{Q} = \rho \mathbf{1}^\top \boldsymbol{Q}. \tag{67}$$

Since $\mathbf{1}^\top \boldsymbol{X}\boldsymbol{Q} = \rho \mathbf{1}^\top$ also holds by the assumption in Lemma 9, combining with (67), we get

$$\rho \mathbf{1}^\top \boldsymbol{Q} = \rho \mathbf{1}^\top \implies \mathbf{1}^\top \boldsymbol{Q} = \mathbf{1}^\top. \tag{68}$$

The result in (68) is used in order to conclude Lemma 9 from Lemma A.1 of (Huang et al., 2015).

We consider the following criterion:

$$\underset{\boldsymbol{W},\boldsymbol{F}}{\text{maximize}} \ \det(\boldsymbol{W}^\top \boldsymbol{W}) \tag{69a}$$

$$\text{s.t. } \widehat{\boldsymbol{P}} = \boldsymbol{W}\boldsymbol{F} \tag{69b}$$

$$\boldsymbol{W} \geq \mathbf{0}, \mathbf{1}^\top \boldsymbol{W} = M\mathbf{1}^\top, \tag{69c}$$

where we used the constraint $\mathbf{1}^\top \boldsymbol{W} = M\mathbf{1}^\top$ since $\mathbf{1}^\top \boldsymbol{A}_m = \mathbf{1}^\top$ for all $m$.

Let $\boldsymbol{W}^*$ and $\boldsymbol{F}^*$ be optimal solutions of (69), then we have:

$$\det(\boldsymbol{W}^{*\top}\boldsymbol{W}^*) \geq \det(\boldsymbol{W}^{\natural\top}\boldsymbol{W}^\natural). \tag{70}$$

Applying (62), $\boldsymbol{W}^*$ and $\boldsymbol{F}^*$ satisfies $\boldsymbol{W}^* = \boldsymbol{W}^\natural \boldsymbol{Q}$, $\boldsymbol{F}^* = \boldsymbol{Q}^{-1}\boldsymbol{F}^\natural$, for a certain invertible matrix $\boldsymbol{Q}$. Let us assume that $\boldsymbol{Q}$ is not a permutation matrix, Then, we have

$$\det(\boldsymbol{W}^{*\top}\boldsymbol{W}^*) = \det(\boldsymbol{Q}^\top \boldsymbol{W}^{\natural\top}\boldsymbol{W}^\natural \boldsymbol{Q})$$
$$= |\det(\boldsymbol{Q})|^2 \det(\boldsymbol{W}^{\natural\top}\boldsymbol{W}^\natural)$$
$$< \det(\boldsymbol{W}^{\natural\top}\boldsymbol{W}^\natural)$$

where the last inequality is from Lemma 9 using the SSC condition on $\boldsymbol{W}^\natural$ which is a contradiction from (70). Hence, $\boldsymbol{Q}$ must be a permutation matrix. This implies that the optimal solution $\boldsymbol{W}^*$ and $\boldsymbol{F}^*$ of Problem 69 satisfies the following:

$$\boldsymbol{W}^* = \boldsymbol{W}^\natural \boldsymbol{\Pi}, \tag{71a}$$

$$\boldsymbol{F}^* = \boldsymbol{\Pi}^\top \boldsymbol{F}^\natural, \tag{71b}$$

Similar to the last part of Theorem 2, by using Lemma 3 and Fact 1, we have

$$\boldsymbol{A}_m^* = \boldsymbol{A}_m^\natural \boldsymbol{\Pi}, \forall m.$$

$$\boldsymbol{f}^*(\boldsymbol{x}) = \boldsymbol{\Pi}^\top \boldsymbol{f}^\natural(\boldsymbol{x}), \forall \boldsymbol{x} \sim \mathcal{D}.$$

In this case as well, the optimal solutions do not change if we employ $\log \det(\boldsymbol{W}^\top \boldsymbol{W})$ in (69) in place of $\det(\boldsymbol{W}^\top \boldsymbol{W})$.

**Remark 2** Geometrically, Theorem 3 seeks maximum volume solutions w.r.t. the conic hulls of $\boldsymbol{F}$ and $\boldsymbol{W}^\top$, respectively. One can also note that in both cases of Theorem 3, the corresponding optimal solutions do not change if we minimize $\log\det(\boldsymbol{W}^\top\boldsymbol{W})$ in (63) and minimize $\log\det(\boldsymbol{F}\boldsymbol{F}^\top)$ in (69). However, relying on the SSC of $\boldsymbol{F}^\natural$ ($\boldsymbol{W}^\natural$) and minimizing the volume of conic hull of $\boldsymbol{W}^\top$ ($\boldsymbol{F}$) may be inefficient since $\boldsymbol{P}^\natural = \boldsymbol{W}^\natural\boldsymbol{F}^\natural$ does not hold exactly in practice.

## F    PROOF OF LEMMA 1

Let us define some notations:
$$\ell_k^* = \arg\max_{\ell\in[K]}\widehat{\boldsymbol{A}}_1(k,\ell)$$
$$q_k^* = \arg\max_{q\in[K]}\widehat{\boldsymbol{F}}_1(q,k).$$

Consider the following conditions given in Lemma 1
$$1 - \kappa \le |\widehat{\boldsymbol{A}}_1(k,:)\widehat{\boldsymbol{F}}_1(:,k)| \le 1 + \kappa, \ \forall k \in [K] \tag{72a}$$
$$|\widehat{\boldsymbol{A}}_1(k,:)\widehat{\boldsymbol{F}}_1(:,j)| \le \kappa, \forall k \ne j. \tag{72b}$$

### F.1    PROVING $\ell_k^* \ne q_j^*$, $\forall k \ne j$

We begin by noticing the below relation from (72a):
$$(1-\kappa) \le \widehat{\boldsymbol{A}}_1(k,1)\widehat{\boldsymbol{F}}_1(1,k) + \cdots + \widehat{\boldsymbol{A}}_1(k,K)\widehat{\boldsymbol{F}}_1(K,k)$$
$$\le \max_{\ell\in[K]}\widehat{\boldsymbol{A}}_1(k,\ell)(\widehat{\boldsymbol{F}}_1(1,k) + \cdots + \widehat{\boldsymbol{F}}_1(K,k))$$
$$= \max_{\ell\in[K]}\widehat{\boldsymbol{A}}_1(k,\ell)$$
$$= \widehat{\boldsymbol{A}}_1(k,\ell_k^*), \ \forall k, \tag{73}$$

where the first equality employs the probability simplex constraints on the columns of $\widehat{\boldsymbol{F}}$.

We further proceed to prove by using contradiction. First, assume the below for certain $k \ne j$,
$$\ell_k^* = \arg\max_\ell\widehat{\boldsymbol{A}}_1(k,\ell) = \arg\max_q\widehat{\boldsymbol{F}}_1(q,j) = q_j^*. \tag{74}$$
Under the assumption (74), consider (72b). For (72b) to hold, we need to satisfy the below for a certain $k \ne j$:
$$\widehat{\boldsymbol{A}}_1(k,\ell_k^*)\widehat{\boldsymbol{F}}_1(q_j^*,j) < \kappa$$
$$\implies \widehat{\boldsymbol{F}}_1(q_j^*,j) < \frac{\kappa}{1-\kappa}$$

where the last relation is obtained from (73). This also implies that
$$1 = \sum_{q=1}^K \widehat{\boldsymbol{F}}_1(q,k) < \frac{K\kappa}{1-\kappa}$$
$$\implies \kappa > \frac{1}{K+1}.$$

However, Lemma 1 assumes that $\kappa \le \frac{1}{K+1}$, Hence, the assumption (74) does not hold and we get
$$\arg\max_\ell\widehat{\boldsymbol{A}}_1(k,\ell) \ne \arg\max_q\widehat{\boldsymbol{F}}_1(q,j), \ k \ne j. \tag{75}$$

### F.2    PROVING $\ell_k^* = q_k^*$, $\forall k$

From the result in (75), we consider (72b) for a certain $k \ne j$.
$$\widehat{\boldsymbol{A}}_1(k,\ell_k^*)\widehat{\boldsymbol{F}}_1(q,j) \le \kappa$$
$$\implies \widehat{\boldsymbol{F}}_1(q,j) \le \frac{\kappa}{1-\kappa}, \ q \ne q_j^*. \tag{76}$$

where the last relation is obtained from (73).

Since $\sum_q \widehat{\boldsymbol{F}}_1(q, j) = 1$, we get

$$\widehat{\boldsymbol{F}}_1(q_j^*, j) \geq \frac{1 - K\kappa}{1 - \kappa}, \forall j. \tag{77}$$

From the result in (75) and the condition in (72b), we also have the following for a certain $k \neq j$

$$\widehat{\boldsymbol{A}}_1(k, \ell)\widehat{\boldsymbol{F}}_1(q_j^*, j) \leq \kappa$$

$$\implies \widehat{\boldsymbol{A}}_1(k, \ell) \leq \frac{\kappa(1 - \kappa)}{1 - K\kappa}, \ \ell \neq \ell_k^*, \tag{78}$$

where the last relation by (77).

Next, we proceed to prove by contradiction. First, assume the below for certain $k$

$$\ell_k^* \neq q_k^*. \tag{79}$$

Hence, for each $k$, we have

$$|\widehat{\boldsymbol{A}}_1(k, :)\widehat{\boldsymbol{F}}_1(:, k)| = \widehat{\boldsymbol{A}}_1(k, 1)\widehat{\boldsymbol{F}}_1(1, k) + \cdots + \widehat{\boldsymbol{A}}_1(k, K)\widehat{\boldsymbol{F}}_1(K, k)$$

$$\leq \sum_{q \neq q_k^*} \widehat{\boldsymbol{F}}_1(q, k) + \widehat{\boldsymbol{A}}_1(k, \ell), \ \ell \neq \ell_k^*$$

$$\leq \frac{(K - 1)\kappa}{1 - \kappa} + \frac{\kappa(1 - \kappa)}{1 - K\kappa} \tag{80}$$

where we have used the fact that all the entries of $\widehat{\boldsymbol{A}}_1$ and $\widehat{\boldsymbol{F}}_1$ are smaller than 1, in the first inequality and have applied (76) and (78) in the last inequality.

The lower-bound condition in (72a) gives that for each $k$,

$$|\widehat{\boldsymbol{A}}_1(k, :)\widehat{\boldsymbol{F}}_1(:, k)| \geq 1 - \kappa. \tag{81}$$

Then, comparing both (80) and (81), we hope to have

$$\frac{(K - 1)\kappa}{1 - \kappa} \geq 1 - \kappa$$

$$\Rightarrow \kappa \geq \frac{K + 1 - \sqrt{(K + 1)^2 - 4}}{2}$$

However, $\frac{K+1-\sqrt{(K+1)^2-4}}{2} \geq \frac{1}{K+1}$, which leads to a contradiction to our assumption that $\kappa \leq \frac{1}{K+1}$. Hence, the assumption (79) does not hold and we get

$$\arg\max_\ell \widehat{\boldsymbol{A}}_1(k, \ell) = \arg\max_q \widehat{\boldsymbol{F}}_1(q, k), \ \forall k. \tag{82}$$

Combining both (82) and (82), we get

$$\arg\max_\ell \widehat{\boldsymbol{A}}_1(k, \ell) \neq \arg\max_\ell \widehat{\boldsymbol{A}}_1(j, \ell), \ \forall k \neq j$$

$$\arg\max_q \widehat{\boldsymbol{F}}_1(q, k) \neq \arg\max_q \widehat{\boldsymbol{F}}_1(q, j), \ \forall k \neq j.$$

## G  PROOF OF LEMMA 2

We have

$$\sigma_{\min}(\boldsymbol{X}) = \sigma_{\min}(\boldsymbol{I}_K + \boldsymbol{E}_1 + \boldsymbol{E}_2) = \min_{\|\boldsymbol{x}\|_2 = 1} \|(\boldsymbol{I}_K + \boldsymbol{E}_1 + \boldsymbol{E}_2)\boldsymbol{x}\|_2$$

$$= \min_{\|\boldsymbol{x}\|_2 = 1} \|\boldsymbol{I}_K \boldsymbol{x} + (\boldsymbol{E}_1 + \boldsymbol{E}_2)\boldsymbol{x}\|_2 \overset{(a)}{\geq} \min_{\|\boldsymbol{x}\|_2 = 1} |\|\boldsymbol{I}_K \boldsymbol{x}\|_2 - \|(\boldsymbol{E}_1 + \boldsymbol{E}_2)\boldsymbol{x}\|_2|$$

$$\geq \left| \min_{\|\boldsymbol{x}\|_2 = 1} \|\boldsymbol{I}_K \boldsymbol{x}\|_2 - \max_{\|\boldsymbol{x}\|_2 = 1} \|(\boldsymbol{E}_1 + \boldsymbol{E}_2)\boldsymbol{x}\|_2 \right| = \left| 1 - \max_{\|\boldsymbol{x}\|_2 = 1} \|(\boldsymbol{E}_1 + \boldsymbol{E}_2)\boldsymbol{x}\|_2 \right|$$

$$= |1 - \|\boldsymbol{E}_1 + \boldsymbol{E}_2\|_2| \overset{(b)}{\geq} |1 - \|\boldsymbol{E}_1\|_{\mathrm{F}} - \|\boldsymbol{E}_2\|_{\mathrm{F}}|$$

$$\overset{(c)}{\geq} |1 - \upsilon_1 - \upsilon_2| \tag{83}$$

where the inequality $(a)$ employs the triangle inequality, $(b)$ is obtained via the matrix norm equivalence relation $\|\boldsymbol{E}_1 + \boldsymbol{E}_2\|_2 \leq \|\boldsymbol{E}_1 + \boldsymbol{E}_2\|_F \leq \|\boldsymbol{E}_1\|_F + \|\boldsymbol{E}_2\|_F$, $(c)$ is obtained by the assumption that $\|\boldsymbol{E}_1\|_F \leq \upsilon_1$ and $\|\boldsymbol{E}_2\|_F \leq \upsilon_2$.

## H  PROOF OF LEMMA 3

First, let us define the following w.r.t the loss function $g(\boldsymbol{f}, \boldsymbol{y}) = \|\boldsymbol{f} - \boldsymbol{y}\|_2^2$, $\forall \boldsymbol{f} \in \mathcal{F}, \boldsymbol{y} \in \mathbb{R}^K$ and the input data $\boldsymbol{X} = (\boldsymbol{x}_1, \ldots, \boldsymbol{x}_N)$ where each $\boldsymbol{x}_n$ is sampled i.i.d. from the distribution $\mathcal{D}$ under Assumption 1:

$$L_{\boldsymbol{X}}(\widehat{\boldsymbol{f}}) \triangleq \frac{1}{N} \sum_{n=1}^N g(\widehat{\boldsymbol{f}}(\boldsymbol{x}_n), \boldsymbol{f}^{\natural}(\boldsymbol{x}_n)) = \frac{1}{N} \sum_{n=1}^N \|\widehat{\boldsymbol{f}}(\boldsymbol{x}_n) - \boldsymbol{\Pi}^\top \boldsymbol{f}^{\natural}(\boldsymbol{x}_n)\|_2^2 = \frac{1}{N} \|\widehat{\boldsymbol{F}} - \boldsymbol{\Pi}^\top \boldsymbol{F}^{\natural}\|_F^2,$$
(84a)

$$L_{\mathcal{D}}(\widehat{\boldsymbol{f}}) \triangleq \mathbb{E}_{\boldsymbol{x} \sim \mathcal{D}} \left[ g(\widehat{\boldsymbol{f}}(\boldsymbol{x}), \boldsymbol{f}^{\natural}(\boldsymbol{x})) \right] = \mathbb{E}_{\boldsymbol{x} \sim \mathcal{D}} \left[ \|\widehat{\boldsymbol{f}}(\boldsymbol{x}) - \boldsymbol{\Pi}^\top \boldsymbol{f}^{\natural}(\boldsymbol{x})\|_2^2 \right]. \tag{84b}$$

We invoke Theorem 26.5 in (Shalev-Shwartz & Ben-David, 2014) and get that with probability greater than $1 - \delta$:

$$L_{\mathcal{D}}(\widehat{\boldsymbol{f}}) \leq L_{\boldsymbol{X}}(\widehat{\boldsymbol{f}}) + 2\mathfrak{R}_N(g \circ \mathcal{F}) + 4\bar{c}\sqrt{\frac{2\log(4/\delta)}{N}}$$

$$\implies \mathbb{E}_{\boldsymbol{x} \sim \mathcal{D}} \left[ \|\widehat{\boldsymbol{f}}(\boldsymbol{x}) - \boldsymbol{\Pi}^\top \boldsymbol{f}^{\natural}(\boldsymbol{x})\|_2^2 \right] \leq \frac{1}{N} \|\widehat{\boldsymbol{F}} - \boldsymbol{\Pi}^\top \boldsymbol{F}^{\natural}\|_F^2 + 4\mathfrak{R}_N(\mathcal{F}) + 16\sqrt{\frac{2\log(4/\delta)}{N}} \tag{85}$$

where the last inequality utilizes the definitions in (84), the contraction lemma (Lemma 26.9 from (Shalev-Shwartz & Ben-David, 2014)) and also applied $\bar{c} = 4$ since $|g(\boldsymbol{f}, \boldsymbol{y})| \leq 4$ in our case. The term $\mathfrak{R}_N(\mathcal{F})$ denotes the empirical Rademacher complexity of the neural network function class $\mathcal{F}$ which is upperbounded via the sensitive complexity parameter $\mathscr{R}_{\mathcal{F}}$ as follows (Lin & Zhang, 2019):

$$\mathfrak{R}_N(\mathcal{F}) \leq 16 N^{-5/8} \left( 2\|\boldsymbol{X}\|_F \mathscr{R}_{\mathcal{F}} \right)^{\frac{1}{4}}.$$

## I  PROOF OF PROPOSITION 2

Our goal is to bound $\frac{1}{M} \sum_{m=1}^M \|\boldsymbol{p}_n^{\natural(m)} - \widehat{\boldsymbol{p}}_n^{(m)}\|_2^2$ for each $n$. To achieve this, let us define the random variable $U := \frac{1}{NM} \sum_{n=1}^N \sum_{m=1}^M \|\boldsymbol{p}_n^{\natural(m)} - \widehat{\boldsymbol{p}}_n^{(m)}\|_2^2$. From Proposition 1, the following result hold:
$$\Pr(U \leq \vartheta(\delta)) \geq 1 - \delta,$$

where randomness is due to all $\boldsymbol{x}_n$'s, $\widehat{y}_n^{(m)}$'s, and $\mathcal{S}$ and $\vartheta(\delta)$ is given by the R.H.S. of (54).

Then we have

$$\begin{aligned}
\mathbb{E}[U] &= \int_0^{u_{\max}} h(u) u \, du = \int_0^{\vartheta(\delta)} h(u) u \, du + \int_{\vartheta(\delta)}^{u_{\max}} h(u) u \, du \\
&\leq \vartheta(\delta) \int_0^{\vartheta(\delta)} h(u) du + u_{\max} \int_{\vartheta(\delta)}^{u_{\max}} h(u) du \\
&\leq \vartheta(\delta) + u_{\max}\delta \\
&\leq \vartheta(\delta) + 4\delta
\end{aligned} \tag{86}$$

where $u_{\max}$ denotes the maximum value of the random variable $U$ and is given by $u_{\max} \leq 4$ and $h(u)$ denotes the probability density function of the random variable $U$.

Also, we have

$$\mathbb{E}[U] = \frac{1}{N} \sum_{n=1}^N \mathbb{E}[U_n] \tag{87}$$

where $U_n \triangleq \frac{1}{M} \sum_{m=1}^M \|\boldsymbol{p}_n^{\natural(m)} - \widehat{\boldsymbol{p}}_n^{(m)}\|_2^2$.

To proceed, we have the following lemma:

**Lemma 10** *Let $U_n \triangleq \frac{1}{M} \sum_{m=1}^{M} \|\boldsymbol{p}_n^{\natural(m)} - \widehat{\boldsymbol{p}}_n^{(m)}\|_2^2$. Then*

$$\mathbb{E}[U_n] = \mathbb{E}[U_{n'}], \ \forall, n, n',$$

*where expectation is taken w.r.t. all $\boldsymbol{x}_n$'s, $\widehat{y}_n^{(m)}$'s, and $\mathcal{S}$.*

The proof is provided in Sec. J.

Combining Lemma 10 with (86) and (87), we get

$$\mathbb{E}[U_n] \leq \vartheta(\delta) + 4\delta.$$

Applying Markov inequality, we get the following for any $\tau > 0$

$$\mathsf{Pr}(U_n \leq \tau \mathbb{E}[U_n]) \geq 1 - \frac{1}{\tau}$$

$$\implies \mathsf{Pr}(U_n \leq \tau \vartheta(\delta) + 4\tau\delta) \geq 1 - \frac{1}{\tau}$$

Letting $\tau = N^\alpha$, where $0 < \alpha < 1$, we get that with probability greater than $1 - \frac{1}{N^\alpha}$

$$\frac{1}{M} \sum_{m=1}^{M} \|\boldsymbol{p}_n^{\natural(m)} - \widehat{\boldsymbol{p}}_n^{(m)}\|_2^2 \leq N^\alpha \vartheta(\delta) + 4N^\alpha \delta.$$

## J  PROOF OF LEMMA 10

Let us define the following:

$$u(\boldsymbol{x}_n, \widehat{\boldsymbol{\theta}}_{\boldsymbol{v}_1,\ldots,\boldsymbol{v}_n,\ldots,\boldsymbol{v}_{n'},\ldots,\boldsymbol{v}_N}) \triangleq \frac{1}{M} \sum_{m=1}^{M} \|\boldsymbol{p}_n^{\natural(m)} - \widehat{\boldsymbol{p}}_n^{(m)}\|_2^2$$

$$= \frac{1}{M} \sum_{m=1}^{M} \|\boldsymbol{A}_m^\natural \boldsymbol{f}^\natural(\boldsymbol{x}_n) - \widehat{\boldsymbol{A}}_m \widehat{\boldsymbol{f}}(\boldsymbol{x}_n)\|_2^2$$

where $\boldsymbol{v}_n = \{\boldsymbol{x}_n, \{\widehat{\boldsymbol{y}}_n^{(m)}\}_{m \in \mathcal{S}_n}, \mathcal{S}_n\}$ and $\widehat{\boldsymbol{\theta}}$ denotes estimates $\{\widehat{\boldsymbol{A}}_m\}$ and $\widehat{\boldsymbol{f}}$ which are obtained using the data $\{\boldsymbol{v}_1, \ldots, \boldsymbol{v}_n, \ldots, \boldsymbol{v}_{n'}, \ldots, \boldsymbol{v}_N\} = \{\boldsymbol{x}_1, \ldots, \boldsymbol{x}_N, \widehat{\mathcal{Y}}, \mathcal{S}\}$.

One can observe that for any $n' \neq n$,

$$\widehat{\boldsymbol{\theta}}_{\boldsymbol{v}_1,\ldots,\boldsymbol{v}_n,\ldots,\boldsymbol{v}_{n'},\ldots,\boldsymbol{v}_N} = \widehat{\boldsymbol{\theta}}_{\boldsymbol{v}_1,\ldots,\boldsymbol{v}_{n'},\ldots,\boldsymbol{v}_n,\ldots,\boldsymbol{v}_N} \tag{88}$$

since the order of $\boldsymbol{v}_n$ does not affect the value of the estimates. Then we have

$$\mathop{\mathbb{E}}_{\substack{\boldsymbol{x}_n, \boldsymbol{x}_{n'}, \\ \boldsymbol{x}_j, \, j \neq n, n'}} [u(\boldsymbol{x}_n, \widehat{\boldsymbol{\theta}}_{\boldsymbol{v}_1,\ldots,\boldsymbol{v}_n,\ldots,\boldsymbol{v}_{n'},\ldots,\boldsymbol{v}_N})] = \mathop{\mathbb{E}}_{\substack{\boldsymbol{x}_n, \boldsymbol{x}_{n'}, \\ \boldsymbol{x}_j, \, j \neq n, n'}} [u(\boldsymbol{x}_n, \widehat{\boldsymbol{\theta}}_{\boldsymbol{v}_1,\ldots,\boldsymbol{v}_{n'},\ldots,\boldsymbol{v}_n,\ldots,\boldsymbol{v}_N})]$$

$$= \mathop{\mathbb{E}}_{\substack{\boldsymbol{x}_n, \boldsymbol{x}_{n'}, \\ \boldsymbol{x}_j, \, j \neq n, n'}} [u(\boldsymbol{x}_{n'}, \widehat{\boldsymbol{\theta}}_{\boldsymbol{v}_1,\ldots,\boldsymbol{v}_{n'},\ldots,\boldsymbol{v}_n,\ldots,\boldsymbol{v}_N})] \tag{89}$$

where the first equality is by (88) and the last equality is obtained since $\boldsymbol{x}_n$ and $\boldsymbol{x}_{n'}$ are identically distributed.

From (89), we can obtain that

$$\mathop{\mathbb{E}}_{\boldsymbol{x}_1,\ldots,\boldsymbol{x}_N} [u(\boldsymbol{x}_n, \widehat{\boldsymbol{\theta}}_{\boldsymbol{v}_1,\ldots,\boldsymbol{v}_n,\ldots,\boldsymbol{v}_{n'},\ldots,\boldsymbol{v}_N})] = \mathop{\mathbb{E}}_{\boldsymbol{x}_1,\ldots,\boldsymbol{x}_N} [u(\boldsymbol{x}_{n'}, \widehat{\boldsymbol{\theta}}_{\boldsymbol{v}_1,\ldots,\boldsymbol{v}_{n'},\ldots,\boldsymbol{v}_n,\ldots,\boldsymbol{v}_N})]$$

$$\implies \mathbb{E}_{\widehat{\mathcal{Y}}} \left[ \mathop{\mathbb{E}}_{\boldsymbol{x}_1,\ldots,\boldsymbol{x}_N} [u(\boldsymbol{x}_n, \widehat{\boldsymbol{\theta}}_{\boldsymbol{v}_1,\ldots,\boldsymbol{v}_n,\ldots,\boldsymbol{v}_{n'},\ldots,\boldsymbol{v}_N})] \right] = \mathbb{E}_{\widehat{\mathcal{Y}}} \left[ \mathop{\mathbb{E}}_{\boldsymbol{x}_1,\ldots,\boldsymbol{x}_N} [u(\boldsymbol{x}_{n'}, \widehat{\boldsymbol{\theta}}_{\boldsymbol{v}_1,\ldots,\boldsymbol{v}_{n'},\ldots,\boldsymbol{v}_n,\ldots,\boldsymbol{v}_N})] \right]$$

$$\implies \mathbb{E}_{\mathcal{S}} \left[ \mathbb{E}_{\widehat{\mathcal{Y}}} \left[ \mathop{\mathbb{E}}_{\boldsymbol{x}_1,\ldots,\boldsymbol{x}_N} [u(\boldsymbol{x}_n, \widehat{\boldsymbol{\theta}}_{\boldsymbol{v}_1,\ldots,\boldsymbol{v}_n,\ldots,\boldsymbol{v}_{n'},\ldots,\boldsymbol{v}_N})] \right] \right] = \mathbb{E}_{\mathcal{S}} \left[ \mathbb{E}_{\widehat{\mathcal{Y}}} \left[ \mathop{\mathbb{E}}_{\boldsymbol{x}_1,\ldots,\boldsymbol{x}_N} [u(\boldsymbol{x}_{n'}, \widehat{\boldsymbol{\theta}}_{\boldsymbol{v}_1,\ldots,\boldsymbol{v}_{n'},\ldots,\boldsymbol{v}_n,\ldots,\boldsymbol{v}_N})] \right] \right]$$

$$\implies \mathop{\mathbb{E}}_{\boldsymbol{v}_1,\ldots,\boldsymbol{v}_N} [u(\boldsymbol{x}_n, \widehat{\boldsymbol{\theta}}_{\boldsymbol{v}_1,\ldots,\boldsymbol{v}_n,\ldots,\boldsymbol{v}_{n'},\ldots,\boldsymbol{v}_N})] = \mathop{\mathbb{E}}_{\boldsymbol{v}_1,\ldots,\boldsymbol{v}_N} [u(\boldsymbol{x}_{n'}, \widehat{\boldsymbol{\theta}}_{\boldsymbol{v}_1,\ldots,\boldsymbol{v}_{n'},\ldots,\boldsymbol{v}_n,\ldots,\boldsymbol{v}_N})].$$

$$
\boldsymbol{W}^{\natural} =
\begin{bmatrix}
\vdots \\
\boldsymbol{A}^{\natural}_{m_1}(1,:) \\
\vdots \\
\boldsymbol{A}^{\natural}_{m_2}(2,:) \\
\vdots \\
\boldsymbol{A}^{\natural}_{m_3}(3,:) \\
\vdots
\end{bmatrix}
\begin{matrix}
\\
\approx \ \boldsymbol{e}_1^{\mathsf{T}} \\
\\
\approx \ \boldsymbol{e}_2^{\mathsf{T}} \\
\\
\approx \ \boldsymbol{e}_3^{\mathsf{T}} \\
\\
\end{matrix}
\qquad
\boldsymbol{W}^{\natural} =
\begin{bmatrix}
\vdots \\
\vdots \\
\boldsymbol{A}^{\natural}_{m_*} \\
\vdots \\
\vdots
\end{bmatrix}
\approx \boldsymbol{I}_3
$$

Figure 1: (left) $\boldsymbol{W}^{\natural} = [\boldsymbol{A}^{\natural\top}_1, \ldots, \boldsymbol{A}^{\natural\top}_M]^{\top}$ satisfying NCSA with $K = 3$, meaning that there exists specialists for each class $k \in [K]$ and (right) all-class expert annotator $m^*$ among $M$ annotators, meaning $\boldsymbol{A}^{\natural}_{m_*} \approx \boldsymbol{I}_K$.

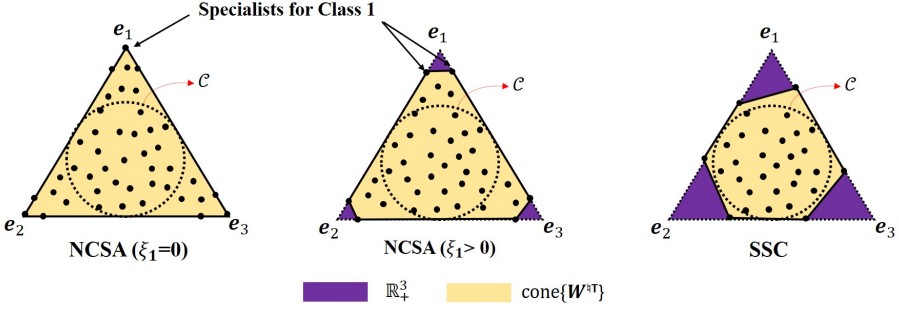

Figure 2: The dots denote the rows of $\boldsymbol{W}^{\natural}$ and the circle denote the second-order cone $\mathcal{C}$.

## K MORE DETAILS ON THE COMPLEXITY MEASURE $\mathscr{R}_{\mathcal{F}}$

The work in (Lin & Zhang, 2019) introduces a complexity measure for measuring the expressiveness of different family of neural network classes. For example, consider a general convolution neural network (CNN) architecture with $L$ layers such that

$$
\boldsymbol{f}(\boldsymbol{x}) = \boldsymbol{\sigma}_L(\gamma_L(\boldsymbol{H}_L)\boldsymbol{\sigma}_{L-1}(\gamma_{L-1}(\boldsymbol{H}_{L-1})\ldots\boldsymbol{\sigma}_1(\gamma_1(\boldsymbol{H}_1)\boldsymbol{x})\ldots)), \ \forall \boldsymbol{f} \in \mathcal{F},
$$

where $\boldsymbol{\sigma}_i(\cdot) = [\sigma_i(\cdot), \ldots, \sigma_i(\cdot)] \in \mathbb{R}^{d_i}$ represents an element-wise activation function at $i$th layer satisfying $\sigma_i(0) = 0$ and $\sigma_i$ being $\rho$-Lipschitz continuous for all $i$. Let $\mathcal{G}_{\mathrm{F}}$ denote the set of layer indices with fully connected layers and $\mathcal{G}_{\mathrm{C}}$ denote the layer indices with convolutional layers. If $i \in \mathcal{G}_{\mathrm{F}}$, we have $\gamma_i(\boldsymbol{H}_i) = \boldsymbol{H}_i \in \mathbb{R}^{d_i \times d_{i-1}}$. Suppose $i \in \mathcal{G}_{\mathrm{C}}$, then we have $\gamma_i(\boldsymbol{H}_i)\boldsymbol{z}_{i-1} = \boldsymbol{H}_i \circledast \boldsymbol{z}_{i-1}$, where the matrix $\boldsymbol{H}_i \in \mathbb{R}^{c_i \times r_i}$ contains $c_i$ convolutional filters each of which having dimension $r_i$, $\boldsymbol{z}_{i-1}$ denotes the output of the $(i-1)$th layer, and $\circledast$ denotes the convolution operation. Here $d_0 = d$ is the dimension of the of the input data items $\boldsymbol{x}_n$'s. Then, its sensitive complexity $\mathscr{R}_{\mathcal{F}}$ is defined as follows:

$$
\mathscr{R}_{\mathcal{F}} = 2\rho^L L^2 \left( \sum_{i \in \mathcal{G}_{\mathrm{F}}} d_i^2 d_{i-1}^2 + \sum_{i \in \mathcal{G}_{\mathrm{C}}} c_i^2 r_i^2 \sqrt{d_i/c_i} \right).
$$

Note that such as a general CNN architecture covers many popular neural networks, e.g., fully connected neural networks, CNNs such as Lenet-5 (Lecun et al., 1998), VGG-16 (Liu & Deng, 2015), and so on.

## L NEAR-CLASS SPECIALIST ASSUMPTION AND SSC

NCSA is a relaxed condition compared to having all-class expert annotators or having diagonally dominant annotators–see Fig. 1. It can be understood that NCSA assumption does not require any single annotator to be specialists with respect to all classes. Nonetheless, having an all-class expert annotator $m_*$ satisfies NCSA, but not vice versa.

The SSC (Definition 1) is a relaxed condition compared to the NCSA (Assumption 4). To illustrate this, we consider the geometry of the matrix $\boldsymbol{W}^\natural$ satisfying the NCSA and the SSC as shown in Fig. 2.

# M   ALGORITHM DESCRIPTION

In this section, we detail the implementation of the regularized criteria in (8) and (9). The neural network predictor function $\boldsymbol{f} \in \mathcal{F}$ is parameterized using $\boldsymbol{\theta}$ and can be denoted as $\boldsymbol{f}_\theta : \mathbb{R}^d \to \boldsymbol{\Delta}$. Let $\boldsymbol{\psi} = [\mathsf{vec}(\boldsymbol{A}_1)^\top, \dots, \mathsf{vec}(\boldsymbol{A}_M)^\top, \mathsf{vec}(\boldsymbol{\theta})^\top]^\top$.

Let us also define the following for certain batch $\mathcal{B} \subset [N]$:

$$\mathcal{L}_F(\boldsymbol{\psi}, \mathcal{B}) = -\frac{1}{|\mathcal{B}|} \sum_{n \in \mathcal{B}} \sum_{m \in \mathcal{S}_n} \sum_{k=1}^K \mathbb{I}[\widehat{y}_n^{(m)} = k] \log[\boldsymbol{A}_m \boldsymbol{f}_\theta(\boldsymbol{x}_n)]_k - \lambda \log \det \widetilde{\boldsymbol{F}} \widetilde{\boldsymbol{F}}^\top,$$

$$\mathcal{L}_W(\boldsymbol{\psi}, \mathcal{B}) = -\frac{1}{|\mathcal{B}|} \sum_{n \in \mathcal{B}} \sum_{m \in \mathcal{S}_n} \sum_{k=1}^K \mathbb{I}[\widehat{y}_n^{(m)} = k] \log[\boldsymbol{A}_m \boldsymbol{f}_\theta(\boldsymbol{x}_n)]_k - \lambda \log \det \boldsymbol{W}^\top \boldsymbol{W},$$

where $\widetilde{\boldsymbol{F}} = [\boldsymbol{f}(\boldsymbol{x}_{n_1}), \dots, \boldsymbol{f}(\boldsymbol{x}_{n_{|\mathcal{B}|}})]$, $n_1, \dots, n_{|\mathcal{B}|} \in \mathcal{B}$ and $\boldsymbol{W} = [\boldsymbol{A}_1^\top, \dots, \boldsymbol{A}_M^\top]^\top$.

Let $\nabla_\psi \mathcal{L}_F(\boldsymbol{\psi}, \mathcal{B})$ denote the stochastic gradient of $\mathcal{L}_F(\boldsymbol{\psi}, \mathcal{B})$ w.r.t. $\boldsymbol{\psi}$ and $\nabla_\psi \mathcal{L}_W(\boldsymbol{\psi}, \mathcal{B})$ denote the stochastic gradient of $\mathcal{L}_W(\boldsymbol{\psi}, \mathcal{B})$ w.r.t. $\boldsymbol{\psi}$. Under these notations, the proposed methods `GeoCrowdNet(F)` and `GeoCrowdNet(W)` are described in Algorithm 1 and 2, respectively. The Python implementation of the algorithms is available in GitHub[1].

---

**Algorithm 1:** `GeoCrowdNet(F)`

---

**input**  :Data $\{\boldsymbol{x}_n, \widehat{y}_n^{(m)}, m \in \mathcal{S}_n\}_{n=1}^N$
1 Initialize $\boldsymbol{A}_m$'s to identity matrices $\boldsymbol{I}_K$;
2 Initialize the parameters $\boldsymbol{\theta}$ of the neural network function class $\mathcal{F}$;
3 $\boldsymbol{\psi} = [\mathsf{vec}(\boldsymbol{A}_1)^\top, \dots, \mathsf{vec}(\boldsymbol{A}_M)^\top, \mathsf{vec}(\boldsymbol{\theta})^\top]^\top$;
4 **while** *stopping criterion is not reached* **do**
5    **while** *stopping criterion is not reached* **do**
6       Draw a random batch $\mathcal{B}$;
7       Compute $\nabla_\psi \mathcal{L}_F(\boldsymbol{\psi}, \mathcal{B})$;
8       $\boldsymbol{\psi} \leftarrow \mathsf{AdamOptimizer}(\boldsymbol{\psi}, \nabla_\psi \mathcal{L}_F(\boldsymbol{\psi}, \mathcal{B}))$;
9       **for** $m = 1$ **to** $M$ **do**
10          $\boldsymbol{A}_m \leftarrow \mathsf{softmax}(\boldsymbol{A}_m)$                ▷ softmax operation on the columns of $\boldsymbol{A}_m$
11       **end**
12    **end**
13 **end**
**output**:Estimates $\widehat{\boldsymbol{A}}_m, \forall m, \widehat{\boldsymbol{\theta}}$

---

[1]`https://github.com/shahanaibrahimosu/end-to-end-crowdsourcing`

---

**Algorithm 2:** `GeoCrowdNet(W)`

---

**input** : Data $\{\boldsymbol{x}_n, \widehat{y}_n^{(m)}, m \in \mathcal{S}_n\}_{n=1}^N$
1 Initialize $\boldsymbol{A}_m$'s to identity matrices $\boldsymbol{I}_K$;
2 Initialize the parameters $\boldsymbol{\theta}$ of the neural network function class $\mathcal{F}$;
3 $\boldsymbol{\psi} = [\text{vec}(\boldsymbol{A}_1)^\top, \dots, \text{vec}(\boldsymbol{A}_M)^\top, \text{vec}(\boldsymbol{\theta})^\top]^\top$;
4 **while** *stopping criterion is not reached* **do**
5    **while** *stopping criterion is not reached* **do**
6       Draw a random batch $\mathcal{B}$;
7       Compute $\nabla_{\boldsymbol{\psi}} \mathcal{L}_W(\boldsymbol{\psi}, \mathcal{B})$;
8       $\boldsymbol{\psi} \leftarrow \text{AdamOptimizer}(\boldsymbol{\psi}, \nabla_{\boldsymbol{\psi}} \mathcal{L}_W(\boldsymbol{\psi}, \mathcal{B}))$;
9       **for** $m = 1$ **to** $M$ **do**
10          $\boldsymbol{A}_m \leftarrow \text{softmax}(\boldsymbol{A}_m)$                   ▷ softmax operation on the columns of $\boldsymbol{A}_m$
11       **end**
12    **end**
   **output :** Estimates $\widehat{\boldsymbol{A}}_m, \forall m, \widehat{\boldsymbol{\theta}}$;
13 **end**

---

# N  ADDITIONAL EXPERIMENTS AND IMPLEMENTATION DETAILS

## N.1  SYNTHETIC ANNOTATORS

**Dataset.** We consider the CIFAR-10 (Krizhevsky, 2009) dataset for synthetic noisy label experiments. CIFAR-10 consists of $60,000$ labeled color images of animals, vehicles and so on, each having a size of $32 \times 32$ and belonging $K = 10$ different classes. We use $45,000$ images for training, $5,000$ images for validation, and $10,000$ images for testing.

**Noisy Label Generation.** In order to produce noisy annotations for the images of the training data, we simulate $M = 5$ synthetic annotators. We randomly choose an annotator $m^*$ among $M$ annotators and its confusion matrix is generated by $\boldsymbol{A}_{m*}^\natural = \boldsymbol{I}_K + \gamma \text{rand}(K, K)$, followed by normalization w.r.t. the $\ell_1$ norm of the corresponding columns. Here, $\boldsymbol{I}_K$ denotes the identity matrix of size $K$ and $\text{rand}(K, K)$ denotes the $K \times K$ matrix with its entries randomly chosen from a uniform distribution between 0 and 1. The parameter $\gamma$ controls how well the annotator $m^*$ correctly identifies the ground-truth labels. The confusion matrices of the remaining $M - 1$ annotators are generated such that they provide labels uniformly at random—i.e., the $M - 1$ annotators are all unreliable. This type of modeling for the confusion matrices resembles the *hammer-spammer* model as employed in the works (Rodrigues & Pereira, 2018; Tanno et al., 2019). Using the ground-truth label $y_n$'s provided by the dataset and the generated confusion matrices, the noisy annotations $\widehat{y}_n^{(m)}$'s are produced. Each annotation $\widehat{y}_n^{(m)}$ is observed independently at random such that only 20% of the total annotations are available for training. Note that, the true labels are not accessible by any of the methods.

**Neural Network Architecture and Settings.** For the CIFAR-10 dataset, we choose the ResNet-9 architecture (He et al., 2016). Adam (Kingma & Ba, 2015) is used as the optimizer with weight decay of $10^{-4}$ and a batch size of 128. The regularization parameter $\lambda$ and the initial learning rate of the Adam optimizer are chosen via grid search method over the validation set from $\{0.01, 0.001, 0.0001\}$ and $\{0.01, 0.001\}$, respectively. We choose the same neural network structures for all the baselines. The confusion matrices are initialized with identity matrices for the proposed method and the baselines `TraceReg` and `CrowdLayer`.

**Results.** Table 3 presents the results under various values of $\gamma$—when $\gamma$ is smaller, the chance of annotator $m^*$ correctly labeling the data items is better. One can see that the proposed approach, namely, `GeoCrowdNet(F)`, outperforms the baselines in all the scenarios under test. When annotator $m^*$'s labeling accuracy drops drastically (i.e., when $\gamma$ becomes larger), `GeoCrowdNet(F)` performs the best. This implies that even if there are no class specialists, `GeoCrowdNet(F)` works well under the large $N$ cases.

Figs. 3, 4 and 5 show the ground-truth confusion matrices and the corresponding estimates when $\gamma = 0.01$ for the `GeoCrowdNet` methods. One can see all methods estimate the confusion matrices reasonably well, corroborating our identifiability claims.

Table 3: Average test accuracy of the proposed methods and the baselines on the CIFAR-10 dataset ($K = 10$), labeled by $M = 5$ synthetic annotators.

| Methods | $\gamma = 0.01$ | $\gamma = 0.2$ | $\gamma = 0.3$ |
|---|---|---|---|
| GeoCrowdNet(F) | $\mathbf{84.82 \pm 0.43}$ | $\mathbf{71.66 \pm 1.08}$ | $\mathbf{67.97 \pm 1.44}$ |
| GeoCrowdNet(W) | $82.76 \pm 0.46$ | $69.20 \pm 0.59$ | $63.86 \pm 1.64$ |
| GeoCrowdNet ($\lambda$=0) | $82.50 \pm 0.38$ | $69.14 \pm 1.29$ | $63.73 \pm 2.64$ |
| TraceReg | $82.54 \pm 0.34$ | $70.01 \pm 1.16$ | $65.01 \pm 2.30$ |
| Crowdlayer | $\mathbf{83.01 \pm 0.23}$ | $69.91 \pm 1.48$ | $\mathbf{65.70 \pm 2.32}$ |
| MBEM | $81.01 \pm 0.32$ | $69.45 \pm 1.56$ | $64.15 \pm 2.21$ |
| CoNAL | $81.41 \pm 0.43$ | $68.56 \pm 1.78$ | $63.21 \pm 1.98$ |
| Max-MIG | $82.01 \pm 0.78$ | $\mathbf{69.96 \pm 1.23}$ | $62.01 \pm 2.46$ |
| NN-MV | $50.96 \pm 0.83$ | $37.90 \pm 1.19$ | $33.24 \pm 1.17$ |
| NN-DSEM | $52.02 \pm 0.80$ | $37.95 \pm 1.41$ | $33.64 \pm 0.69$ |

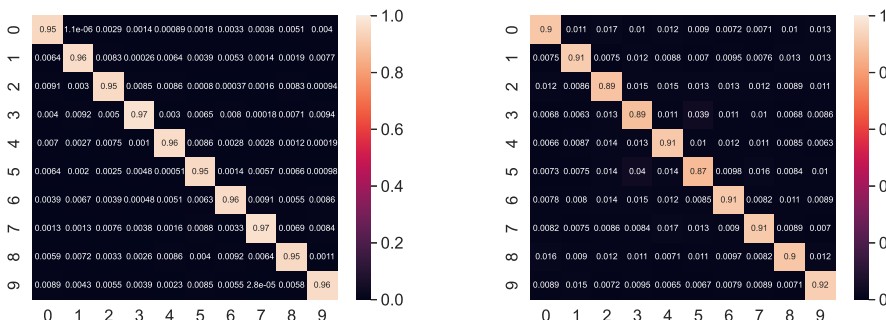

The ground-truth $\boldsymbol{A}_{m*}^{\natural}$ (left) and the estimate $\widehat{\boldsymbol{A}}_{m*}$ (right)

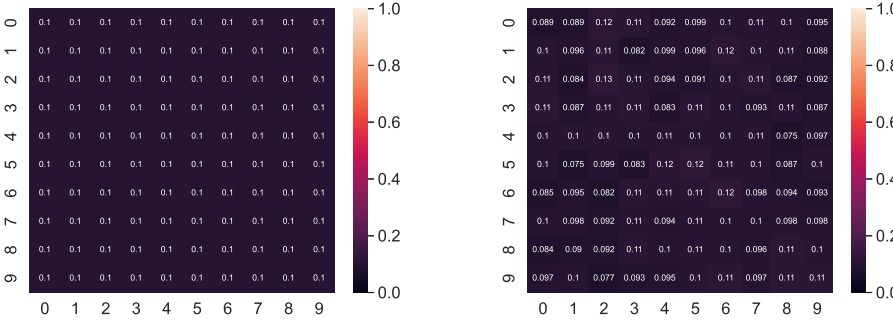

The ground-truth $\boldsymbol{A}_{m}^{\natural}$ (left) and the estimate $\widehat{\boldsymbol{A}}_{m}$ (right), $m \neq m^*$

Figure 3: The illustration of the ground-truth confusion matrices and the estimated confusion matrices by the proposed approach GeoCrowdNet(F) for CIFAR-10 dataset, with $M = 5$ machine annotators and $\gamma = 0.01$. The average mean squared error (MSE) of the confusion matrices is $0.088$.

## N.2 Real Data Experiments - Machine Annotations

Here, we provide additional details of the real data experiments. For experiments with machine annotations, we consider the MNIST and the Fashion-MNIST datasets. Each image in these datasets is of size $28 \times 28$ in the grey-scale format. We consider $57,000$ images as training data, $3000$ images for validation, and $10,000$ images for testing.

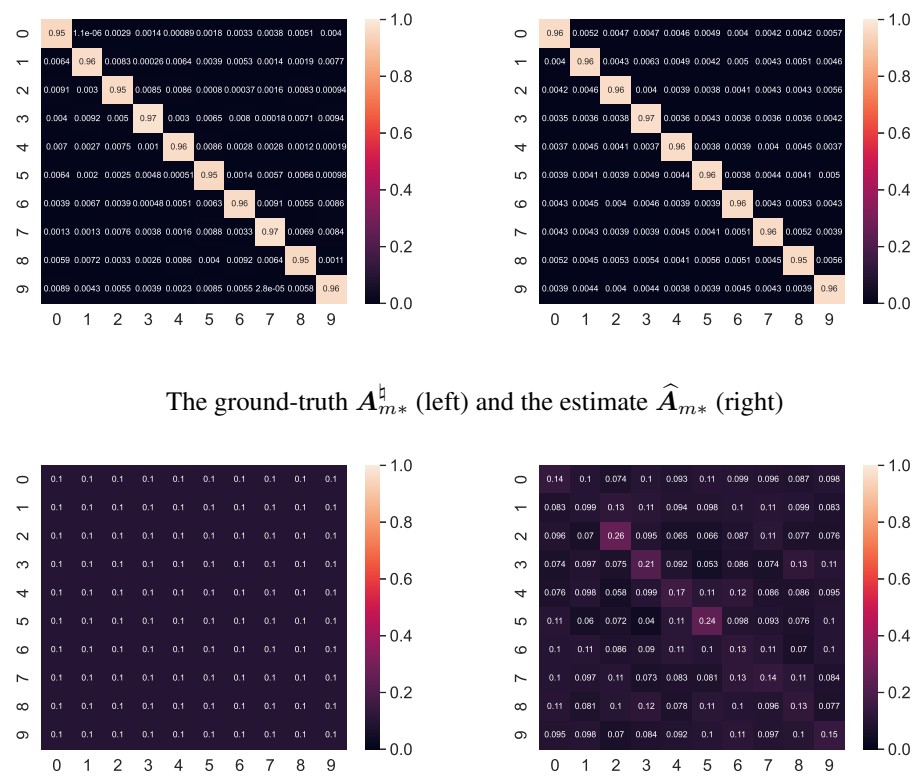

The ground-truth $\boldsymbol{A}_{m*}^{\natural}$ (left) and the estimate $\widehat{\boldsymbol{A}}_{m*}$ (right)

The ground-truth $\boldsymbol{A}_{m}^{\natural}$ (left) and the estimate $\widehat{\boldsymbol{A}}_{m}$ (right), $m \neq m^{*}$

Figure 4: The illustration of the ground-truth confusion matrices and the estimated confusion matrices by the proposed approach `GeoCrowdNet(W)` for CIFAR-10 dataset, with $M = 5$ machine annotators and $\gamma = 0.01$. The average mean squared error (MSE) of the confusion matrices is $0.102$.

For machine annotation, we train a number of machine classifiers to act as annotators. In Table 1, under Case 2, we choose $M = 5$ annotators. Specifically, Linear SVM, logistic regression, $k$-NN with $k = 5$, CNN with two convolution layers followed by a max pooling layer, a fully connected layer and a softmax layer, and a fully connected neural network (FCNN) with 1 hidden layer and 128 hidden units are trained. The CNN is trained for 5 epochs and the FCNN is trained for 10 epochs. The classifiers are intentionally not well trained, so that they can mimic error-proning annotators. For example, for the MNIST dataset, the individual label prediction accuracy of these annotators on unseen data items of size $N$ ranges from 15.88% to 82.23% (averaged over 5 random trials) and 2 annotators have less than 50% accuracy on average. Also note that our algorithm does not know which annotator is more accurate.

For Case 2, we train more annotators by changing some parameters of the above mentioned machine classifiers. Specifically, we use linear SVM, polynomial kernel-based SVM, and Gaussian SVM as different variants of SVM. For $k$-NN, we choose $k$ from $\{3, 5, 7, 10\}$. FCNN, CNN, and logistic regression are trained with different number of epochs $\{10, 15, 20, 25\}$ and with random initializations for each case. Under case 1, if an annotator is chosen to be a specialist, it is trained with more number of samples ($10, 000$ samples) with more number of epochs such that its label prediction accuracy is higher than 90%. Under this strategy, in Table 1 for the case $M = 15$, the individual label prediction accuracy of the annotators on unseen data items of size $N$ ranges from 3.15% to 95.27% (2 annotators with more than 90% accuracy) . For the case with $M = 20$, the individual label prediction accuracy of the annotators on unseen data items of size $N$ ranges from 3.24% to 95.73% (3 annotators with more than 90% accuracy).

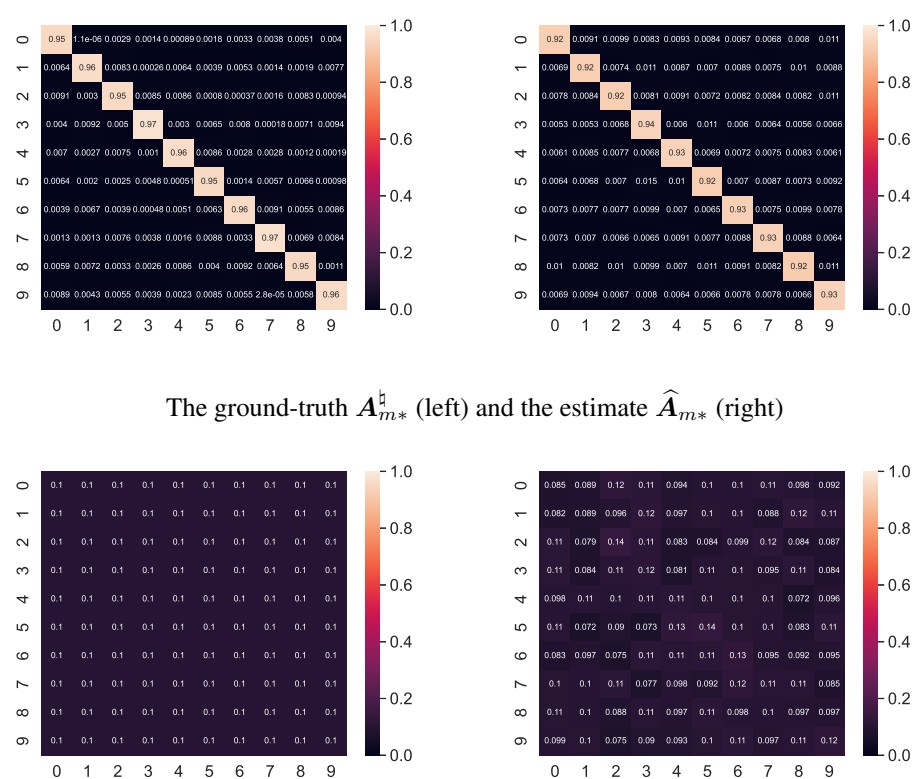

The ground-truth $\boldsymbol{A}_{m*}^{\natural}$ (left) and the estimate $\widehat{\boldsymbol{A}}_{m*}$ (right)

The ground-truth $\boldsymbol{A}_{m}^{\natural}$ (left) and the estimate $\widehat{\boldsymbol{A}}_{m}$ (right), $m \neq m^*$

Figure 5: The illustration of the ground-truth confusion matrices and the estimated confusion matrices by `GeoCrowdNet(`$\lambda = 0$`)` for CIFAR-10 dataset, with $M = 5$ machine annotators and $\gamma = 0.01$. The average mean squared error (MSE) of the confusion matrices is 0.097.

Table 4: Average test accuracy ($\pm$ std) of the proposed methods and the baselines on Fashion-MNIST under various $(N, M)$'s on Case 1; labels are produced by machine annotators; $p = 0.1$.

| Methods | $(1000, 25)$ | $(750, 30)$ |
|---------|--------------|-------------|
| GeoCrowdNet(F) | $\mathbf{82.81 \pm 2.05}$ | $\mathbf{82.40 \pm 1.51}$ |
| GeoCrowdNet(W) | $\mathbf{83.33 \pm 3.39}$ | $\mathbf{83.65 \pm 4.11}$ |
| GeoCrowdNet($\lambda = 0$) | $79.43 \pm 4.07$ | $79.07 \pm 2.91$ |
| TraceReg | $80.79 \pm 3.87$ | $81.12 \pm 4.03$ |
| CrowdLayer | $73.54 \pm 6.58$ | $75.56 \pm 8.19$ |
| MBEM | $26.19 \pm 3.45$ | $26.67 \pm 3.45$ |
| CoNAL | $63.13 \pm 8.53$ | $66.02 \pm 5.65$ |
| Max-MIG | $77.05 \pm 3.06$ | $72.59 \pm 8.38$ |
| NN-MV | $68.48 \pm 3.15$ | $71.74 \pm 2.21$ |
| NN-DSEM | $68.84 \pm 2.46$ | $73.08 \pm 4.09$ |

Additional experiment results using the MNIST and Fashion-MNIST datasets can be found in Tables 4-6. As expected, `GeoCrowdNet(W)` and `GeoCrowdNet(F)` offer the best performance in Case 1 and Case 2, respectively. .

Table 5: Average test accuracy ($\pm$ std) of the proposed methods and the baselines on MNIST under various $(N, M)$'s on Case 2; labels are produced by machine annotators; $p = 0.1$.

| Methods | $(5000, 25)$ | $(10000, 25)$ |
|---|---|---|
| GeoCrowdNet(F) | **87.26 $\pm$ 0.70** | **87.66 $\pm$ 1.23** |
| GeoCrowdNet(W) | **84.64 $\pm$ 4.04** | 86.53 $\pm$ 3.24 |
| GeoCrowdNet ($\lambda = 0$) | 80.08 $\pm$ 3.14 | 85.51 $\pm$ 2.35 |
| TraceReg | 78.06 $\pm$ 6.72 | **86.85 $\pm$ 5.45** |
| CrowdLayer | 57.39 $\pm$ 6.93 | 56.66 $\pm$ 6.34 |
| MBEM | 35.56 $\pm$ 7.92 | 21.82 $\pm$ 6.54 |
| CoNAL | 46.35 $\pm$ 6.75 | 46.86 $\pm$ 5.43 |
| Max-MIG | 83.34 $\pm$ 1.54 | 86.84 $\pm$ 2.13 |
| NN-MV | 83.46 $\pm$ 0.67 | 83.32 $\pm$ 1.34 |
| NN-DSEM | 84.37 $\pm$ 0.70 | 83.78 $\pm$ 1.11 |

Table 6: Average test accuracy ($\pm$ std) of the proposed methods and the baselines on Fashion-MNIST under various $(N, M)$'s on Case 2; labels are produced by machine annotators; $p = 0.1$.

| Methods | $(5000, 25)$ | $(10000, 25)$ |
|---|---|---|
| GeoCrowdNet(F) | **87.47 $\pm$ 0.86** | **88.39 $\pm$ 1.92** |
| GeoCrowdNet(W) | **85.80 $\pm$ 1.26** | **87.09 $\pm$ 2.71** |
| GeoCrowdNet ($\lambda = 0$) | 85.58 $\pm$ 2.23 | 85.37 $\pm$ 2.59 |
| TraceReg | 85.20 $\pm$ 2.36 | 84.02 $\pm$ 2.74 |
| CrowdLayer | 70.07 $\pm$ 7.19 | 74.92 $\pm$ 6.19 |
| MBEM | 42.15 $\pm$ 3.65 | 44.34 $\pm$ 3.48 |
| CoNAL | 65.13 $\pm$ 4.94 | 62.65 $\pm$ 7.02 |
| Max-MIG | 85. 33 $\pm$ 2.09 | 85.94 $\pm$ 1.93 |
| NN-MV | 79.85 $\pm$ 2.88 | 83.53 $\pm$ 1.53 |
| NN-DSEM | 81.34 $\pm$ 1.65 | 84.67 $\pm$ 1.15 |

### N.3 REAL DATA EXPERIMENTS - HUMAN ANNOTATORS

For experiments with annotations collected from AMT, we use the LabelMe dataset (Rodrigues et al., 2017; Russell et al., 2007) and the Music dataset (Rodrigues et al., 2014).

The LabelMe dataset (Rodrigues et al., 2017; Russell et al., 2007) is an image classification dataset consisting of 2688 images from $K = 8$ different classes, namely, highway, inside city, tall building, street, forest, coast, mountain, and open country. From the available images, 1000 images are annotated by $M = 59$ AMT workers. In total, about 2547 image annotations are obtained by the AMT workers whose labeling accuracy ranges from $0\%$ to $100\%$ with a mean accuracy of $69.2\%$. In order to enrich the training dataset, standard augmentation techniques such as rescaling, cropping, horizontal flips, etc., are employed and accordingly, the training dataset consists of $10,000$ images annotated by 59 workers; see more details in (Rodrigues & Pereira, 2018; Chu et al., 2021). The validation set consists of 500 images and the remaining 1188 images are used for testing.

The Music dataset (Rodrigues et al., 2014) consists of audio samples of 10 different genres of music such as classical, country, disco, hiphop, jazz, rock, blues, reggae, pop, and metal. The dataset has about 1000 samples of songs each having a duration of 30 seconds. About 700 samples are annotated by 44 AMT workers (overall, about 2946 annotations are observed) and the remaining 300 are allocated for testing. Out of the annotated 700 samples, we consider 595 samples for training, and the remaining 105 samples are used for validation.

For the LabelMe dataset, we employ the settings used in (Rodrigues & Pereira, 2018). The pretrained VGG-16 embeddings for the images are given as inputs to a fully connected (FC) neural network with one hidden layer having 128 hidden units and ReLU activation functions. A dropout layer with rate $50\%$ is also used, followed by a softmax layer. For the Music dataset, we choose the same FC layer and the softmax layer as employed in the LabelMe settings, but with batch normalization layers before each of these layers.

