# OpenReview forum: "Deep Learning From Crowdsourced Labels: Coupled Cross-Entropy Minimization, Identifiability, and Regularization"
_ICLR.cc/2023/Conference — ICLR 2023 poster_

### Official Review · Reviewer_MaWv · 2022-10-23

**Confidence:** 4
**Correctness:** 3
**Technical Novelty And Significance:** 3
**Empirical Novelty And Significance:** 3
**Recommendation:** 6

**Clarity, Quality, Novelty And Reproducibility:**

The presentation of this paper is easy to follow and the study is technically sound. The design and technique details of the software structure are disclosed and reproducible.

**Strength And Weaknesses:**

The presentation of this paper is easy to follow and the study is technically sound. The design and technique details of the software structure are disclosed and reproducible. In addition, the theoretical analysis presented in this paper is conducive to research on the CCEM criterion.

There are several issues:
- The experimental results in Table 1 are somewhat weak, and the authors should discuss the results in more detail, e.g. the performance of Max-MIG is competitive in most cases, and in case 1, GeoCrowdNet(W) does not show the advantages well compared with GeoCrowdNet(F) though the N is small and M is relatively large.
- Whether there is an appropriate strategy or method to make decisions about which variant (GeoCrowdNet(W) or GeoCrowdNet(F)) should be used on a specific dataset?



**Summary Of The Paper:**

This paper provides theoretical analysis to guarantee the performance of the CCEM criterion for the first time. Furthermore, to improve the identifiability of the classifiers and the workers, the authors also proposed two variants of the CCEM by introducing two different regularization terms. Experimental results on several synthetic and real-world datasets show corroborate the theoretical claims and the effectiveness of the proposed methods.

**Summary Of The Review:**

Overall, this is a valuable piece of work. The proposed methods and the theoretical analysis are conducive and attractive for researchers in the areas of CCEM in crowdsourcing. However, there are also some weaknesses in the paper. The quality of the paper could be further improved.

---

> ### Author Response · Authors · 2022-11-13
> **Response to Reviewer MaWv (Part 1)**
>
> Thank you for your comments and for the appreciation of our work. Based on your comments, we have updated our manuscript with more discussions. Please see our detailed response as follows:
>
> [**Max-MIG’s performance**]: For Max-MIG, our understanding is as follows:
>
> - According to Theorem 3.4 of (Cao et al., 2019), the Max-MIG method works when there exists a set of all-class experts as their annotators (i.e., annotators who rarely make mistakes when given data items from any class).
> - Hence, if the annotators do not contain all-class experts, but $\boldsymbol{F}^\top$ satisfies the SSC, then GeoCrowdNet(F) is expected to outperform Max-MIG, as GeoCrowdNet(F) does not have requirements on the expertise of the annotators. Note that when $N$ is large, $\boldsymbol{F}^\top$ may easily satisfy the SSC (see Theorem 4 of (Ibrahim et al. 2019)). Hence, we expect that GeoCrowdNet(F) outperforms Max-MIG when $N$ is large but the annotators are not well-trained.
>
> Note that the above was reflected in Table I (the two columns on the right) of our original submission. The results were exactly as expected.
>
> To make the above points clearer, we have also run some new experiments using the same settings in another dataset, namely, FashionMNIST. The following Table presents the results for Fashion-MNIST dataset on case 2 (i.e., all-class experts unlikely to exist). Average test accuracies over 5 random trials are reported. More details on experiment settings and results are provided in Appendix O of the revised version. We can observe that GeoCrowdNet(F) works consistently much better than Max-MIG in these challenging scenarios.
>
> |                   | $N=5000$ | $N =10000$ | $N=1000$ | $N=2000$ |
> |:-----------------:|:--------:|:----------:|:--------:|:--------:|
> |      Methods      |   $M=5$  |   $M= 5$   |  $M=25$  |  $M=25$  |
> |   GeoCrowdNet(F)  |   **80.60**  |    **83.68**   |   **78.35**  |   **85.25**  |
> |   GeoCrowdNet(W)  |   72.36  |    **74.03**   |   **78.20**  |   **83.26**  |
> | GeoCrowdNet (λ=0) |   69.31  |    73.04   |   77.59  |   81.38  |
> |      TraceReg     |   69.82  |    72.21   |   77.50  |   81.67  |
> |     CrowdLayer    |   64.73  |    63.18   |   67.79  |   70.67  |
> |        MBEM       |   37.34  |    38.62   |   37.43  |   39.23  |
> |       CoNAL       |   52.15  |    54.61   |   65.81  |   66.37  |
> |      Max-MIG      |   **73.45**  |    73.62   |   73.76  |   79.09  |
> |       NN-MV       |   61.95  |    71.64   |   71.87  |   73.06  |
> |      NN-DSEM      |   31.40  |    35.10   |   68.29  |   74.78  |
>
> We have added more discussion in the results section to highlight these points.
>
> (To be continued ...)

---

> > ### Author Response · Authors · 2022-11-13
> > **Response to Reviewer MaWv (Part 2)**
> >
> > [**Performance of GeoCrowNet(W) over GeoCrowNet(F)**]: Indeed, GeoCrowdNet(F) generally performs well in our experiments. This perhaps is because for deep neural network-based problems, $N$ is always not extremely small (otherwise there is not enough data to learn the network parameters).  The data items may be diverse enough to satisfy SSC for $\boldsymbol{F}^\top$. Note that if $\boldsymbol{F}^\top$ satisfies the SSC, GeoCrowdNet(F) guarantees model identifiability (Theorem 3(a)).
> >
> > However, GeoCrowdNet(W) could still consistently show visible margin over GeoCrowNet(F) in small-sample cases. The following Table presents the results on case 1 ($\boldsymbol{W}$ likely to have SSC) on the Fashion-MNIST data. One can  see that when $N=750$, the advantage of GeoCrowdNet(W) is most articulated. Hence, we believe that GeoCrowdNet(W) at least offers a reasonable alternative for the practitioners to use in practice, in such small sample cases.
> >
> > - #### **Case 1 ($\boldsymbol{W}$ likely to have SSC)**
> >
> > |                | $N=1000$ | $N =1000$ | $N =1000$ | $N =750$ |
> > |:--------------:|:--------:|:---------:|:---------:|:--------:|
> > |     Methods    |  $M=15$  |  $M =25$  |   $M=30$  |  $M=30$  |
> > | GeoCrowdNet(F) |   78.98  |   82.81   |   84.47   |   82.40  |
> > | GeoCrowdNet(W) |   **79.80**  |   **83.33**   |   **85.56**   |   **83.65**  |
> >
> > [**Strategy to Choose the Regularization**]: By Theorem 3(a), GeoCrowdNet(F) has identifiability when $\boldsymbol{F}^\top$ satisfies SSC, which is more likely to hold if $N$ is large; see Theorem 4 of (ibrahim et. al. 2019). By Theorem 3(b), GeoCrowdNet(W) will have identifiability guarantees if $\boldsymbol{W}$ satisfies SSC. $\boldsymbol{W}$ is more likely to satisfy the SSC when it has more diverse rows. In other words, this happens when there are a large number of annotators whose skills are diverse, $\boldsymbol{W}$ is more likely to satisfy SSC; again, see Theorem 4 of (ibrahim et. al. 2019). Under such cases, GeoCrowdNet(W) is likely to work better.
> >
> > Hence, our **rule of thumb** is as follows: When $M$ is large (e.g., $M>30$) but $N$ is relatively small (e.g., $N<1000$), using  GeoCrowdNet(W) is recommended. However, when $N$ is large, using GeoCrowdNet(F) is recommended. In the revised version, we have clearly articulated this strategy.

---

> ### Author Response · Authors · 2022-11-17
> **Gentle Reminder**
>
> Dear reviewer,
>
> We are wondering if you could take a look at our response and the revised manuscript. We hope our response resolves your concerns. We are happy to discuss if you have any follow-up questions.
>
> Thank you again for your time and efforts.

---

> > ### Comment · Reviewer_MaWv · 2022-11-20
> > **Rebuttal read**
> >
> > Thanks the author's response and futher experiments. However, I would like to remain the score in 6.

---

### Official Review · Reviewer_v39Y · 2022-10-24

**Confidence:** 3
**Correctness:** 4
**Technical Novelty And Significance:** 3
**Empirical Novelty And Significance:** 3
**Recommendation:** 6

**Clarity, Quality, Novelty And Reproducibility:**

Clarity: this paper is clearly written.

Quality: this paper provide strong theoretical proofs of their proposed method.

Novelty: this paper is designed on top of (Rodrigues & Pereira, 2018) and provides better regularization terms. This is novel.

Reproducibility: the author has provided their code for re-implementing their experiments.

**Strength And Weaknesses:**

Strength: This paper relaxed the assumption conditional independence among the annotators in crowdsourcing tasks with coupled cross entropy minimization criterion. The newly designed regularization terms based on the relaxed assumption consider 1) when the annotation data is large (voting matrix is not sparse), no expert/dominate is also ok, can set the classifier-related regularization terms; 2) when there are more annotators, can set the voting matrix as regularzation term. This paper transfers an intuitive impression "the more annotators/annotations and the stronger classifiers, the better the performance of the crowdsourcing task" in a mathematical way.

Weakness/problems:
- The author should provide a more detailed analysis of how and when to select the regularization term, with varying N and M. Table 1 has only 6 combinations, it's really hard to make an accurate conclusion of how to choose the regularization term.
 - When N and M are large, are the performances of GeoCrowdNet(M) and GeoCrowdNet(F) come close to majority voting? Does it still works when the voting marix is sparse?
 - In Table 1, case 2, N = 5000, why TraceReg perform much worse than GeoCrowdNet(λ = 0)? TraceReg has trace constraints compared with GeoCrowdNet(λ = 0).


**Summary Of The Paper:**

This paper designed new regularization terms of coupled cross entropy minimization criterion in crowdsourcing tasks. It relaxed the assumption of conditional independence among the annotators.

**Summary Of The Review:**

This paper relaxed the assumptions of current coupled cross entropy minimization criterion based crowdsourcing tasks. More experiments needed to be provided to illustrate how to choose the regularization terms.

---

> ### Author Response · Authors · 2022-11-13
> **Response to Reviewer v39Y (Part 1)**
>
> Thank you for reviewing our work and for appreciating the novelty and the strength of theoretical results. Based on your comments, we have updated our manuscript with more experiment results and discussions.
>
> [**Summary of the paper**] We hope to mention that in addition to “designing new regularization” for CCEM, another major contribution is ***understanding*** the plain-vanilla CCEM criterion’s identifiability under realistic conditions. Some new findings are interesting, e.g., that the CCEM provably works without assuming that the annotators are conditionally independent.
>
> [**Choosing Regularization**]: To see how to choose the regularization, let us recall our findings:
>
> - By Theorem 3(a), GeoCrowdNet(F) has identifiability when $\boldsymbol{F}^\top$ satisfies SSC, which is more likely to hold if $N$ is large.
> - By Theorem 3(b), GeoCrowdNet(W) will have identifiability guarantees if $\boldsymbol{W}$ satisfies SSC. $\boldsymbol{W}$ is more likely to satisfy the SSC when it has more diverse rows. Hence, when there are many annotators with diverse skills, GeoCrowdNet(W) is expected to work with performance assurances.
>
> Based on the above findings, our **rule of thumb** is as follows: When $M$ is large but $N$ is relatively small (e.g., $N<1000$), using  GeoCrowdNet(W) is recommended. However, when $N$ is large, using GeoCrowdNet(F) is recommended. This is reflected in the results presented in Table I of the original submission for the MNIST dataset. We have also considered another dataset, namely Fashion-MNIST, to further verify this strategy. The results are provided in the following table. Average test accuracies over 5 random trials are reported.
>
>
> - #### **Case 1 ($\boldsymbol{W}$ likely to have SSC)**
>
> |                | $N=1000$ | $N =1000$ | $N =1000$ | $N =750$ |
> |:--------------:|:--------:|:---------:|:---------:|:--------:|
> |     Methods    |  $M=15$  |  $M =25$  |   $M=30$  |  $M=30$  |
> | GeoCrowdNet(F) |   78.98  |   82.81   |   84.47   |   82.40  |
> | GeoCrowdNet(W) |   **79.80**  |   **83.33**   |   **85.56**   |   **83.65**  |
>
>  - #### **Case 2 ($\boldsymbol{W}$ unlikely to have SSC)**
>
> |                | $N=5000$ | $N =10000$ | $N=1000$ | $N=2000$ |
> |:--------------:|:--------:|:----------:|:--------:|:--------:|
> |     Methods    |   $M=5$  |   $M= 5$   |  $M=25$  |  $M=25$  |
> | GeoCrowdNet(F) |   **80.60**  |    **83.68**   |   **78.35**  |   **85.25**  |
> | GeoCrowdNet(W) |   72.36  |    74.03   |   78.20  |   83.26  |
>
> In the revised version, we have made this rule of thumb clearer.
>
> (To be continued ...)

---

> > ### Author Response · Authors · 2022-11-13
> > **Response to Reviewer v39Y (Part 2)**
> >
> > [**Settings with Large $N$ and Large $M$**]:  Under our setting, when $N$ and $M$ are large, the “voting matrix” (our understanding the voting matrix tabulates who labeled which data item, which is a annotator-by-item matrix of size $M \times N$) can still be quite sparse, as long as every sample is only labeled by a small number of annotators. To be specific, as every annotator labels a sample with probability $p$, the “voting matrix” only has $p \times 100%$ percent of entries that are observed. In our experiments, $p= 0.1$, which means that the voting matrix is always quite sparse in our experiments.
> >
> > To show some empirical evidence for large $N$ and large $M$ cases, we have run experiments using the same settings as in Table I. We also report the results from the Fashion-MNIST (F-MNIST) dataset as well. The following Table presents a snapshot of the results for case 2. Average test accuracies over 5 random trials are reported. More details on experiment settings and results are provided in Appendix O of the revised version. It is observed that when both $N$ and $M$ are large, GeoCrowdNet(F) and GeoCrowdNet(W) work better compared to NN-MV (neural network classifier trained using labels produced by majority voting).
> >
> > |                   |  MNIST       |            | F-MNIST    |           |
> > |:-----------------:|:------------:|:----------:|:----------:|:---------:|
> > |                   |   $N=5000$   |  $N=10000$ |  $N=5000$  | $N=10000$ |
> > |      Methods      |    $M=25$    |   $M=30$   |   $M=25$   |   $M=30$  |
> > |   GeoCrowdNet(F)  |     **87.26**    |    **87.66**   |    **87.47**   |   **88.39**   |
> > |   GeoCrowdNet(W)  |     **84.64**    |    **86.53**   |    **85.80**   |   **87.09**   |
> > | GeoCrowdNet (λ=0) |     80.08    |    85.51   |    85.58   |   85.37   |
> > |       NN-MV       |     83.46    |    83.32   |    79.85   |   83.53   |
> >
> > [**TraceReg’s Performance in Table I**]: According to Theorem 1 in (Tanno et al., 2019), the TraceReg method is guaranteed to work when the average of the confusion matrices $\boldsymbol{A}_m$’s is diagonally dominant. This would happen if a big proportion of the annotators have diagonally dominant confusion matrices, which means they are relatively close to expert annotators. However, GeoCrowdNet($\lambda=0$) does not need such a requirement for model identifiability (see our Theorem 1). In case 2 of Table I, there are no all-class experts present. Hence, it is not surprising that GeoCrowdNet($\lambda=0$) works better than TraceReg.

---

> > > ### Comment · Reviewer_v39Y · 2022-11-18
> > > **response**
> > >
> > > Thanks the author's response and futher experiments, my concerns have been addressed. I would increase my score to 6.

---

> ### Author Response · Authors · 2022-11-17
> **Gentle Reminder**
>
> Dear reviewer,
>
> We are wondering if you could take a look at our response and the revised manuscript. We are happy to follow-up if you have any further questions. We hope our response resolves your concerns and you could reconsider your score.
>
> Thank you again for your time and efforts.

---

### Official Review · Reviewer_WQN3 · 2022-10-26

**Confidence:** 4
**Correctness:** 3
**Technical Novelty And Significance:** 2
**Empirical Novelty And Significance:** 3
**Recommendation:** 5

**Clarity, Quality, Novelty And Reproducibility:**

Quality
=====================
The paper is logically organised; however, it is not easy to follow. Comprehensive experiments were conducted.

Novelty
========================
Marginal the improvement by comparing to the existing method seems marginal.

Reproducibility
========================
The reproducibility is high, and the necessary setting is given. However, it will be greatly appreciated if the source code can be published.

**Strength And Weaknesses:**

Strength
Overall, the paper is logically organised. The research problem is clear and formulated. Theoretically seems sound. Comprehensive experiments were conducted to show the effeteness of the proposed approaches.

Weaknesses
In session 1, too much spend on the background of the course source while the importance of identification is not mentioned.
The author claims that the existing identifiability result on CCEM was derived under restricted conditions, while the proposed method needs to hold under five different assumptions, and some of the assumptions are difficult to hold under real-world conditions.
The performance improvement seems marginal by comparing it to the existing methods such as Max-MIG.
The discussion on the experiment's result is not sufficient. Especially the performance difference between Max-MIG needs to be discussed.

**Summary Of The Paper:**

In this paper, the author proposed two different regularised versions of CCEM to enhance the identifiability of the method by introducing geometric conditions. The comprehensive experiment is conducted to demonstrate the effectiveness of the proposed approach where GeoCrowdNet(F) performs better when more annotated items available, where GeoCrowdNet(W) shows advantages when class specialists are available.

**Summary Of The Review:**

The weakness outweighed the strengths in the current version.

---

> ### Author Response · Authors · 2022-11-13
> **Response to Reviewer WQN3 (Part 1)**
>
> Thank you for your comments and for appreciating theoretical soundness and comprehensive experiments presented in our work. Based on your comments, we have updated our manuscript and the changes are highlighted in “blue”. Please see our point-to-point response as follows:
>
> [**Importance of Identifiability**]: Under the model $\boldsymbol{p}_n^{(m)}= \boldsymbol{A}_m \boldsymbol{f}(\boldsymbol{x}_n)$, the identifiability of $\boldsymbol{A}_m$ and $\boldsymbol{f}$ are important for us to find the true characterizations for annotator $m$ and the ground-truth label predictor $\boldsymbol{f}$. Note that the model  $\boldsymbol{p}_n^{(m)}= \boldsymbol{A}_m \boldsymbol{f}(\boldsymbol{x}_n)$ is not unique (or, not identifiable) in general, as $\boldsymbol{A}_m \boldsymbol{Q}$ and $\boldsymbol{Q}^{-1} \boldsymbol{f}(\boldsymbol{x}_n)$  (where $\boldsymbol{Q}$ is any nonsingular matrix) give rise to the same product  $\boldsymbol{p}_n^{(m)}$. But $\boldsymbol{A}_m \boldsymbol{Q}$ cannot reflect the true characteristics of annotator $m$, nor can $\boldsymbol{Q}^{-1} \boldsymbol{f}(\boldsymbol{x}_n)$ give us the correct label for $\boldsymbol{x}_n$. Hence, establishing identifiability of $\boldsymbol{A}_m$ and $\boldsymbol{f}$ can:
>
> 1. Help us correctly characterize the annotators using $\boldsymbol{A}_m$
> 2. Make correct label prediction using $\boldsymbol{f}(\cdot)$.
>
> We have added the above importance of identifiability in the revised version.
>
> [**Assumptions of Theorem 1**]:  Here, we respond to the comment
> *“The author claims that the existing identifiability result on CCEM was derived under restricted conditions, while the proposed method needs to hold under five different assumptions, and some of the assumptions are difficult to hold under real-world conditions.”*
>
> First, we hope to clarify that Theorem 1 is not for our “proposed method”, as CCEM is a widely used criterion in the literature, not proposed by us.  Theorem 1 is for ***understanding*** the CCEM criterion, which is currently lacking; i.e., the existing limited identifiability results on CCEM and its variants are not satisfactory.
>
> Second, as the reviewer mentioned, we do have 5 assumptions for our identifiability result, **but none of them is more restrictive than those in existing results**, e.g., those from (Tanno et al., 2019; Cao et al., 2019; Khetan et al., 2018). To be specific, (Tanno et al., 2019) requires that the average of $\boldsymbol{A}_m$ over the annotators is diagonally dominant, which holds when a big proportion of the annotators are “expert” annotators. In (Cao et al., 2019), there is an assumption requiring the existence of all-class expert annotators. In (Khetan et al., 2018), the analysis is based on the assumption that the annotators’ decisions are conditionally independent.  We showed that in fact the CCEM criterion does not need these restrictive conditions on the annotators. Our assumption on annotators is more relaxed (Assumption 4). We also showed that they need not have to be conditionally independent.
>
> Third, the reason why we have Assumptions 1-5 is because we tackle finite-sample analysis, which naturally needs more regularity conditions relative to the infinite-sample analysis, to make the analysis mathematically viable. For example, the works in (Tanno et al., 2019; Cao et al., 2019) looked only into the infinite-sample case.  But most of the 5 conditions (especially Assumptions 1,2,3) are standard tools used for finite sample analysis. Assumption 5 is also widely used in the noisy label learning literature (Xia et al., 2019; Li et al., 2021).
>
> Last, we would be happy to discuss with the reviewer on the last sentence, i.e., “some of the assumptions are difficult to hold under real-world conditions”. If the reviewer could point out which condition is considered difficult to hold, we will follow up to discuss.
>
> We hope the above discussion and clarification would make the significance of Theorem 1 more clearly articulated.
>
> (To be continued ...)

---

> > ### Author Response · Authors · 2022-11-13
> > **Response to Reviewer WQN3 (Part 2)**
> >
> > [**Comparison with Max-MIG**]: We respond to the comment *“The performance improvement seems marginal by comparing it to the existing methods such as Max-MIG. The discussion on the experiment's result is not sufficient. Especially the performance difference between Max-MIG needs to be discussed”* here.
> >
> > We hope to clarify that our experiments are not solely for demonstrating our edge over other methods like Max-MIG, but to validate our identifiability result (Theorem 1). In other words, we hoped to use the experiments to verify our understanding of the CCEM criterion and the designed regularizations. There are several points that we hope to verify:
> >
> > 1. CCEM works better under the existence of class specialists + anchor samples (both are more likely to hold under large $M$ and large $N$)
> > 2. GeoCrowdNet(F) is supposed to work without any class specialist with $\boldsymbol{F}^\top$ satisfying SSC, which is more likely to hold when $N$ is large.
> > 3. GeoCrowdNet(W) is supposed to work better when $\boldsymbol{W}$ satisfies SSC, which is likely to hold when class specialists exist or when $M$ is large. Simply put, GeoCrowdNet(W) is expected to work better than GeoCrowdNet(F) when $M$ is large and $N$ is relatively small (e.g., $N<1000$).
> >
> > Hence, in our experiments, there are some cases (e.g.,Table I, Case 1) under test where our method seems to be not much improved upon the baselines such as Max-MIG, but those are **expected**. Such results are presented to reflect our analyses and understanding.
> >
> > To explain, the Max-MIG method works when there exists a set of all-class experts as their annotators (i.e., annotators who rarely make mistakes when given data items from any class), according to their identifiability result from Theorem 3.4 of (Cao et al., 2019).  This condition has a better chance to hold when one has more annotators (larger $M$). Hence, our understanding is that Max-MIG has a bigger risk of not performing very well when $M$ is small. But our method GeoCrowdNet(F) does not need experts and has ensured identifiability when $\boldsymbol{F}^\top$ satisfies the SSC (which is likely to hold when $N$ gets larger). **Therefore, according to our analysis, under the case where $N$ is large and no experts are present, GeoCrowdNet(F) is expected to outperform Max-MIG. This was exactly shown in Table I under case 2**. However, when $M$ is large and when there exist all-class experts, Max-MIG is likely to be more competitive, as shown in Table I under case 1. Both cases validated our understanding and theorems in our work.
> >
> > According to the above understanding/discussion, one can expect that in the challenging case where none of the annotators are experts and $N$ is large, GeoCrowdNet(F) will get a larger margin over Max-MIG. To solidify this understanding, we present more results here. The following Table presents a snapshot of the results from the Fashion-MNIST dataset using the same settings as used in Table I for case 2 (unlikely to have all-class experts). Average test accuracies over 5 random trials are reported. More details and results are provided in Appendix O of the revised version. One can observe that GeoCrowdNet(F) works much better than Max-MIG in these challenging scenarios.
> > |                   | $N=5000$ | $N =10000$ | $N=1000$ | $N=2000$ |
> > |:-----------------:|:--------:|:----------:|:--------:|:--------:|
> > |       Method      |   $M=5$  |    $M=5$   |  $M=25$  |  $M=25$  |
> > |   GeoCrowdNet(F)  |   **80.60**  |    **83.68**   |   **78.35**  |   **85.25**  |
> > |   GeoCrowdNet(W)  |   72.36  |    **74.03**   |   **78.20**  |   **83.26**  |
> > | GeoCrowdNet (λ=0) |   69.31  |    73.04   |   77.59  |   81.38  |
> > |      Max-MIG      |   **73.45**  |    73.62   |   73.76  |   79.09  |
> >
> > We did not present more such cases because we hoped to use the experiments to validate our understanding and theorems, instead of just showing edge over the baselines. In the revised version, we will clarify these points, especially to make the purpose of each of our experiments clearer.
> >
> >
> > [**Reproducibility, Source Code**]: We have already provided the source code as supplementary material in our original submission.

---

> ### Author Response · Authors · 2022-11-17
> **Gentle Reminder**
>
> Dear reviewer,
>
> We are wondering if you could take a look at our response and the revised manuscript and see if there are things that need follow-up discussion. If our response resolves your concerns, we hope you could reconsider your score.
>
> Thank you again for your time and feedbacks.

---

> ### Author Response · Authors · 2022-12-07
> **Gentle Reminder**
>
> Dear Reviewer,
>
> This is a gentle reminder: we have not yet heard from you.  We are wondering if you have further questions or concerns regarding our response and the updated manuscript. We hope our reply addressed your concerns and you could reconsider your score.
>
> Thanks again for your feedback.

---

### Decision · Program_Chairs · 2023-01-20

**Decision:**

Accept: poster

**Justification For Why Not Higher Score:**

Assumptions are hard to verify and understand in practice
Over the two proposed E2E strategies, only one seem to perform best in practice (the F-regularized).
At the end of the paper, we do not know if the main goal was presenting the CCEM (highlited in eq 6a) theory or the importance of a new regularization term in crowdsourced losses (or both, but most of the paper seem to be on the second part).
Background methods could have been presented more deeply in appendix and less in the main paper to leave room for more on the 1st part.
Annotators are simulated with a confusion matrix, and the model estimates such a matrix. Even though it is classical to do this to obtain a comparison of estimated confusion matrices, this puts aside methods not based on such a tool.
Writting could be polished.


**Justification For Why Not Lower Score:**

The work Builds on Rodrigues et. al (2018) to improve results with new regularization terms.
Theoretical garantees over the model proposed are of interest, and based on confusion matrices (easy to interpret parameters) and it leads to competitive methods.
Code is provided




**Metareview: Summary, Strengths And Weaknesses:**

This paper deals with learning with crowdsourcing.
It introduces a new metric (CCEM) with two different types of regularization (depending on the number of annotators) with competitive performance on test accuracy in simulated datasets and good performance in real datasets.
It builds on Rodrigues et. al (2018) to improve results with new regularization terms.


**Note From Pc:**

if the above contains the word "oral" or "spotlight" please see: "oral" presentation means -> notable-top-5% and "spotlight" means -> notable-top-25%. As stated in our emails, we are disassociating presentation type from AC recommendations

**Summary Of Ac-Reviewer Meeting:**

No meeting, the reviewers did not bother interacting during the discussion phase.